# Full counting statistics for interacting trapped fermions

**Naftali R. Smith[1,2], Pierre Le Doussal[3],**
**Satya N. Majumdar[3] and Grégory Schehr[4⋆]**

**1** Laboratoire de Physique de l'École Normale Supérieure, CNRS, ENS & Université PSL, Sorbonne Université, Université de Paris, 75005 Paris, France
**2** Department of Solar Energy and Environmental Physics, Blaustein Institutes for Desert Research, Ben-Gurion University of the Negev, Sede Boqer Campus, 8499000, Israel
**3** Université Paris-Saclay, CNRS, LPTMS, 91405, Orsay, France
**4** Sorbonne Université, Laboratoire de Physique Théorique et Hautes Energies, CNRS UMR 7589, 4 Place Jussieu, 75252 Paris Cedex 05, France

⋆ schehr@lpthe.jussieu.fr

## Abstract

We study $N$ spinless fermions in their ground state confined by an external potential in one dimension with long range interactions of the general Calogero-Sutherland type. For some choices of the potential this system maps to standard random matrix ensembles for general values of the Dyson index $\beta$. In the fermion model $\beta$ controls the strength of the interaction, $\beta = 2$ corresponding to the noninteracting case. We study the quantum fluctuations of the number of fermions $\mathcal{N}_\mathcal{D}$ in a domain $\mathcal{D}$ of macroscopic size in the bulk of the Fermi gas. We predict that for general $\beta$ the variance of $\mathcal{N}_\mathcal{D}$ grows as $A_\beta \log N + B_\beta$ for $N \gg 1$ and we obtain a formula for $A_\beta$ and $B_\beta$. This is based on an explicit calculation for $\beta \in \{1, 2, 4\}$ and on a conjecture that we formulate for general $\beta$. This conjecture further allows us to obtain a universal formula for the higher cumulants of $\mathcal{N}_\mathcal{D}$. Our results for the variance in the microscopic regime are found to be consistent with the predictions of the Luttinger liquid theory with parameter $K = 2/\beta$, and allow to go beyond. In addition we present families of interacting fermion models in one dimension which, in their ground states, can be mapped onto random matrix models. We obtain the mean fermion density for these models for general interaction parameter $\beta$. In some cases the fermion density exhibits interesting transitions, for example we obtain a noninteracting fermion formulation of the Gross-Witten-Wadia model.



# 1 Introduction

## 1.1 Overview

The full counting statistics (FCS), which measures the fluctuations of the number of particles $\mathcal{N}_{\mathcal{D}}$ inside a domain $\mathcal{D}$ has been studied extensively in the context of shot noise [1], quantum transport [2], quantum dots [4,5], non-equilibrium Luttinger liquids [3] as well as in quantum spin chains and fermionic chains [6–11]. The FCS is particularly important for noninteracting fermions because of its connection to the entanglement entropy [12–16]. For free fermions, in the absence of external potential and at zero temperature it is well known that both the variance of $\mathcal{N}_{\mathcal{D}}$ and the entropy grow as $\sim R^{d-1} \log R$ with the typical size $R$ of the domain $\mathcal{D}$ in space dimension $d$ [18–23].

Recently these results have been extended for noninteracting fermions in the presence of a confining potential. This is important for applications e.g. to cold atoms experiments

[26–29] where the fermions are in traps of tunable shapes [30, 31]. In a confining potential, the Fermi gas is supported over a finite domain. Its mean density is inhomogeneous and can be calculated using the well known local density approximation (LDA) [24, 25]. To compute quantum correlations, in particular at the edge of the Fermi gas, where the density vanishes and the LDA method fails, more elaborate methods have been developed [32–34].

In $d = 1$, one can exploit the fact that for specific potentials, the problem at zero temperature can be mapped to standard random matrix ensembles, for which powerful mathematical tools are available. For instance, for $N$ noninteracting spinless fermions in a harmonic well, described by the single particle Hamiltonian $H = \frac{p^2}{2} + V(x)$ with $V(x) = \frac{1}{2}x^2$, the quantum joint probability distribution function (PDF) for the positions $\vec{x} = \{x_i\}_{i=1,\dots,N}$ of the fermions takes the form

$$|\Psi_0(\vec{x})|^2 \propto \prod_{i<j} |x_i - x_j|^\beta e^{-\sum_{i=1}^N x_i^2} , \tag{1}$$

with $\beta = 2$ and $\Psi_0(\vec{x})$ denotes the ground state many body wave function. Remarkably, Eq. (1), specialised to $\beta = 2$, coincides with the joint PDF of the eigenvalues $\lambda_i$ of a random matrix belonging to the Gaussian unitary ensemble (GUE). The two problems thus map into each other with the identification $\lambda_i = x_i$. As a result, for large $N$, the mean fermion density, i.e., the quantum average $\rho(x) = \langle \sum_i \delta(x - x_i) \rangle$, takes the Wigner semi-circle form $\rho(x) \simeq \rho^{\mathrm{bulk}}(x) = \frac{1}{\pi}\sqrt{2N - x^2}$ in the bulk, i.e., for $x \in [x_e^-, x_e^+]$ and vanishes beyond the edges $x_e^\pm \simeq \pm\sqrt{2N}$. To discuss the FCS within an interval $[a, b]$ at large $N$, i.e., the fluctuations of $\mathcal{N}_{[a,b]}$, one needs to distinguish two natural length scales: the microscopic scale given by the interparticle distance $\sim 1/\sqrt{N}$, and the macroscopic scale of order $x_e^+ - x_e^- \sim \sqrt{N}$. It is well known since Dyson and Mehta [35–38] that for an interval of microscopic size the variance for the GUE is given by [6, 39–46]

$$\mathrm{Var}\,\mathcal{N}_{[a,b]} \simeq \frac{1}{\pi^2}\left[\log\left(\sqrt{2N - a^2}\,|b - a|\right) + c_2\right] , \tag{2}$$

for $\sqrt{N}|b - a| = O(1) \gg 1$, with $c_2 = \gamma_E + 1 + \log 2$, where $\gamma_E$ is Euler's constant. This result is obtained from the celebrated sine kernel which describes the eigenvalue correlations in the GUE at microscopic scales. The formula (2) thus carries to the fermions in the harmonic potential. The FCS for an interval of macroscopic size has been studied more recently. For the harmonic well, some results for the variance in that regime were obtained using the connection to the GUE, both in mathematics[1] [49–51], and in physics using a Coulomb gas method [43, 44].

The mapping to random matrix theory (RMT) holds however only for a few specific potentials. For instance the so-called Wishart-Laguerre ensemble is related to the potential $V(x) = \frac{x^2}{2} + \frac{\gamma^2 - \frac{1}{4}}{x^2}$ for $x > 0$. For an arbitrary smooth $V(x)$, not necessarily related to RMT, we have recently obtained a general formula [52] for the variance $\mathrm{Var}\,\mathcal{N}_{[a,b]}$ for a macroscopic interval $[a, b]$ in the bulk. The method used combines determinantal point processes with semi-classical (WKB) approaches. In the special cases related to RMT, the formula recovers the available exact results [53]. It is also in general agreement with recent approaches relying on inhomogeneous bosonization [54–56], although our method allows for more precise and controled results.

One can also ask about higher cumulants of $\mathcal{N}_{[a,b]}$, i.e., beyond the variance given in (2), both for microscopic and macroscopic interval $[a, b]$. In the absence of potential, i.e., for free fermions, there exist results for the higher cumulants which are obtained using the sine-kernel [6, 42, 46]. A natural conjecture, which we put forward in [52] is that these higher cumulants are determined solely from fluctuations on microscopic scales. Consequently (i)

---

[1]See e.g. pages 2-4 in [47, 48].

they are independent of the size of the interval (within the bulk) and (ii) they are universal, i.e., independent of the precise shape of the potential (assumed to be smooth). This conjecture was used to obtain a prediction for the entanglement entropy for noninteracting fermions in a potential in [52].

An outstanding question is the study of the FCS for interacting particles [57]. It is of current interest for cold atom experiments, which have recently measured particle number fluctuations in the $1d$ Bose gas [58, 59]. On the theory side however there are only a few results, even for integrable models. For instance, for the delta Bose gas (Lieb-Liniger) model, an exact formula was derived using the Bethe ansatz for the FCS in the limit of a very small interval [60, 61]. Results for larger intervals were also obtained, but are only valid in the weak interaction/high temperature regime [62]. For interacting fermions, numerical results were obtained for the Hubbard model [63]. In the context of spin chains, several exact formulae for the FCS were obtained, e.g for the XXZ spin chain [64, 65], for the transverse field Ising model [10] and for the Haldane-Shastry chain [9].

In view of our previous works on the FCS of noninteracting trapped fermions it is thus natural to look for extensions which include interactions. A promising direction is to explore further the connection between random matrix theory for a general value of the Dyson index $\beta$, and trapped fermions in $1d$ in the presence of two-body interactions of the Calogero-Sutherland type [66–69, 71]. The simplest example corresponds to the Gaussian $\beta$ ensemble (G$\beta$E) which contains the GUE for $\beta = 2$, as well as the other standard ensembles, the GOE for $\beta = 1$ and the GSE for $\beta = 4$ [41, 71]. For general $\beta$ they can be constructed from random tridiagonal matrices [72]. The joint PDF of their eigenvalues $\lambda_i$ is given by (1) with the substitution $x_i \equiv \sqrt{\frac{\beta}{2}} \lambda_i$. The important observation is that Eq. (1) is also the quantum joint PDF, $|\Psi_0(\vec{x})|^2$, of the positions $x_i$ of $N$ fermions in the ground state of the following $N$ body Hamiltonian

$$\mathcal{H}_N = \sum_{i=1}^{N} \left( \frac{p_i^2}{2} + \frac{x_i^2}{2} \right) + \sum_{1 \le i < j \le N} \frac{\beta(\beta - 2)}{4(x_i - x_j)^2} . \tag{3}$$

It thus describes fermions which interact through either repulsive ($\beta > 2$) or attractive ($1 \le \beta < 2$) long range $1/x^2$ interaction. As we discuss below there are several other examples of two-body interactions and trapping potentials for which a similar connection exists. Note that there are also lattice versions of these models, e.g. Haldane-Shastry spin chains corresponding to a discretized version of the circular $\beta$ ensemble (C$\beta$E) with $\beta = 4$, for which FCS results exist [9].

In this paper we will use the relations between interacting fermions and RMT for general $\beta$, to obtain precise predictions for the FCS for various examples of trapped fermions in the presence of interactions. In the rest of this section we first present the main models that we will study, we explain the main idea of the method and we present the main results.

## 1.2 Models and mappings

Let us now describe the class of models which we study in this paper. In this section we focus on models related to RMT, while further extensions will be discussed below. Here we consider $N$ spinless fermions trapped in an external potential $V(x)$ and with two-body interactions parameterized by a symmetric function $W(x, y) = W(y, x)$. The Hamiltonian is of the general form (we use units such that $m = \hbar = 1$)

$$\mathcal{H}_N = \sum_{i=1}^{N} \left[ \frac{p_i^2}{2} + V(x_i) \right] + \sum_{i<j} W(x_i, x_j) . \tag{4}$$

Table 1: The mappings between (i) models of interacting trapped fermions studied here and (ii) the standard random matrix ensembles. The variable $x$ denotes the positions of the fermions, $\lambda$ the eigenvalues of the RMT ensemble, and the mapping $\lambda(x)$ is displayed in the last column. The first three columns denote respectively the domain for the fermions, the external potential $V(x)$, and their interaction $W(x, y)$, defined in (3). Note that in the second line periodic boundary conditions are to be understood for the fermionic system. The next two columns indicate the RMT ensemble and the matrix potential $V_0(\lambda)$, see Eq. (6). Here $\beta$ is the Dyson index which varies continuously and corresponds to noninteracting fermions for $\beta = 2$.

| Fermions' domain | Fermion potential $V(x)$ | Fermion interaction $W(x, y)$ | RMT ensemble | Matrix potential $V_0(\lambda)$ | Map $\lambda(x)$ |
|---|---|---|---|---|---|
| $x \in \mathbb{R}$ | $x^2/2$ | $\frac{\beta(\beta-2)}{4(x-y)^2}$ | G$\beta$E | $\beta\lambda^2/2$ | $\lambda = \sqrt{\frac{2}{\beta}}x$ |
| $x \in [0, L]$ | $0$ | $\left(\frac{2\pi}{L}\right)^2 \frac{\beta(\beta-2)}{16\sin^2\frac{\pi(x-y)}{L}}$ | C$\beta$E | $0$ | $\lambda = e^{ix\frac{2\pi}{L}}$ |
| $x \in \mathbb{R}^+$ | $\frac{x^2}{2} + \frac{\gamma^2-\frac{1}{4}}{2x^2}$ | $\frac{\beta(\beta-2)}{4}\left[\frac{1}{(x-y)^2} + \frac{1}{(x+y)^2}\right]$ | WL$\beta$E | $\frac{\beta}{2}\lambda - \gamma\log\lambda$ | $\lambda = \frac{2}{\beta}x^2$ |
| $x \in [0, \pi]$ | $\frac{1}{8}\left(\frac{\gamma_1^2-\frac{1}{4}}{\sin^2\frac{x}{2}} + \frac{\gamma_2^2-\frac{1}{4}}{\cos^2\frac{x}{2}}\right)$ | $\frac{\beta(\beta-2)}{16}\left(\frac{1}{\sin^2\frac{x-y}{2}} + \frac{1}{\sin^2\frac{x+y}{2}}\right)$ | J$\beta$E | $\log\frac{1}{\lambda^{\gamma_1}(1-\lambda)^{\gamma_2}}$ | $\lambda = \frac{1-\cos x}{2}$ |

In this paper we study specific choices for $V(x)$ and $W(x, y)$ such that the joint PDF of the positions of the fermions in the ground state can be written in the form

$$|\Psi_0(\vec{x})|^2 = e^{-U(\vec{x})}, \; U(\vec{x}) = \sum_i v(x_i) + \sum_{i<j} w(x_i, x_j), \tag{5}$$

with $w$ a symmetric function $w(x, y) = w(y, x)$. One example of such models corresponds to Eqs. (3), with $V(x) = \frac{x^2}{2}$ and $W(x, y) = \frac{\beta(\beta-2)}{4(x-y)^2}$, and (1), with $v(x) = x^2$ and $w(x, y) = -\beta\log|x - y|$. This is one instance of a more general class of fermion models that can be mapped onto a random matrix ensemble (in that case G$\beta$E).

Let us briefly review the ensembles of interest. We denote $\lambda_i$, $i = 1, \dots, N$ the eigenvalues of a random matrix in an ensemble such that the joint PDF can be written as

$$P(\vec{\lambda}) = \frac{e^{-F(\vec{\lambda})}}{Z_N}, \; F(\vec{\lambda}) = \sum_{i=1}^N V_0(\lambda_i) - \beta\sum_{i<j}\log\left|\lambda_i - \lambda_j\right|, \tag{6}$$

where $\vec{\lambda} = \{\lambda_i\}_{i=1,\dots,N}$ and $Z_N$ is a normalisation constant. Here $\beta$ is the Dyson index and $V_0$ the matrix potential (not to be confused with the fermion potential $V$). For instance the Gaussian-beta ensemble (G$\beta$E) corresponds to the case where the eigenvalues are on the real axis, $\lambda_i \in \mathbb{R}$, with $V_0(\lambda) = \frac{\beta}{2}\lambda^2$. It contains the Gaussian unitary, orthogonal and symplectic ensemble for $\beta = 2, 1, 4$ respectively. The circular-beta ensemble (C$\beta$E) corresponds to the case where the eigenvalues are on the unit circle in the complex plane, with $V_0(\lambda) = 0$, and includes the circular unitary ensemble (CUE) for $\beta = 2$. The Wishart-Laguerre-beta ensemble (WL$\beta$E) corresponds to $\lambda_i \in \mathbb{R}^+$ and $V_0(\lambda) = \frac{\beta}{2}\lambda - \gamma\log\lambda$. The Jacobi-beta ensemble (J$\beta$E) corresponds to $\lambda_i \in [0, 1]$ and $V_0(\lambda) = -\gamma_1\log\lambda - \gamma_2\log(1-\lambda)$. In all these cases and for any $\beta$, $Z_N$ has an explicit expression as a Selberg integral [71], and the ensembles can be mapped onto certain tridiagonal matrices [72]. These ensembles are recapitulated in the Table 1.

The general idea behind the mapping between fermions and RMT is that, upon some map which we denote $\lambda(x)$, i.e., $\lambda_i = \lambda(x_i)$, one can identify the joint PDF (6) with the quan-

tum joint PDF (5) corresponding to the many-body ground state of the fermion system with Hamiltonian (4). Taking into account the Jacobian of the map, the correspondence reads

$$v(x) = V_0(\lambda(x)) - \log|\lambda'(x)|, \tag{7}$$

$$w(x, x') = -\beta \log|\lambda(x) - \lambda(x')|. \tag{8}$$

The simplest case is the mapping $\lambda(x) = \sqrt{\frac{2}{\beta}} x$ from the G$\beta$E to the fermions on the real axis described by the model (3). For the C$\beta$E ensemble the map is $\lambda(x) = e^{ix\frac{2\pi}{L}}$, where the fermions live on the periodic ring, $x_j \in [0, L]$. In that case it maps onto the Sutherland model, without any external potential $V(x) = 0$, see second line of the Table 1. For the WL$\beta$E ensemble the map is $\lambda(x) = \frac{2}{\beta} x^2$ and the fermions live on $\mathbb{R}^+$, $x_i > 0$, with potential sum of harmonic and $1/x^2$ wall, and $1/x^2$ type interactions as given in the third line of the Table 1. Finally for the J$\beta$E ensemble the map is $\lambda(x) = \frac{1}{2}(1 - \cos x)$ and the fermions live in a box $x_i \in [0, \pi]$ with potential and interactions given in the fourth line of the Table 1. For $\gamma_1 = \gamma_2 = 1/2$ the potential is a hard box with Dirichlet boundary conditions. For $\beta = 2$ the fermions are noninteracting in all four cases, and their positions $x_i$ in the ground state form a determinantal point process. Note that the above mappings are valid for any $N$.

To summarize, our main strategy behind the mapping between the ground state of trapped fermions with two-body interactions $W$ and the joint distribution of the eigenvalues of a matrix model consists of the following three steps.

- We consider the Hamiltonian $\mathcal{H}_N$ in (4) which has only one and two-body potentials, $V$ and $W$ respectively. We then write the many-body ground state wave function in any given ordered sector, e.g. $x_1 < \cdots < x_N$, as $\Psi_0(\vec{x}) \sim e^{-U(\vec{x})/2}$, where $U(\vec{x})$ is of the form (5) consisting only of one-body and two-body terms.

- We next substitute this wave function $\Psi_0(\vec{x}) \sim e^{-U(\vec{x})/2}$ in the Schrödinger equation $\mathcal{H}_N \Psi_0 = E_0 \Psi_0$ (in an ordered sector). The main condition is that this equation is satisfied for some value of the ground state energy $E_0$, i.e., that no three-body interaction is generated upon applying the kinetic operator. This condition selects some special families of potentials $V$ and interactions $W$. In the absence of potential this approach dates back to Sutherland and Calogero [69, 70]. This is a standard although tedious calculation, recalled in Appendix A, allowing also to determine $E_0$ for each model. The fact that $\Psi_0$ is indeed the ground state is ensured by the additional condition that $\Psi_0(\vec{x})$ vanishes only at $x_i = x_j$ for $i \neq j$, but not elsewhere,[2] see also [66, 67].

- Finally, we identify the quantum probability, given by $|\Psi_0(\vec{x})|^2$, as the joint PDF of the eigenvalues $\lambda_1, \ldots, \lambda_N$ of a random matrix, under a map $\lambda_i = \lambda(x_i)$, and we show how to construct this map explicitly for several examples. This last step allows us to identify new connections between interacting (and noninteracting) fermions and random matrix models, see e.g. Section 6.

**Mean density**. The simplest observable to compute is the average density $\rho(x)$ of the fermions

$$\rho(x) = \left\langle \sum_{i=1}^{N} \delta(x - x_i) \right\rangle, \tag{9}$$

---

[2]Besides, the wave function must be well behaved around $x_i = x_j$. If it vanishes slower than square root, e.g. $\Psi_0(\ldots x_i, \ldots, x_j, \ldots) \sim |x_i - x_j|^\nu$, $\nu < 1/2$ (as $x_i \to x_j$), then the expectation value of the kinetic energy diverges. It is easier to see this by considering a single-particle wave function whose behavior around the origin is $\psi(x) \sim |x|^\nu$, and then the expectation value of the kinetic energy is $\sim -\int \psi(x) \psi''(x) dx = \int [\psi'(x)]^2 dx = \infty$. For $\beta < 1$, it is this condition that prevents the mapping from RMT to fermions.

where $\langle\dots\rangle$ denotes expectation values with respect to the ground state. In particular one can ask how the interactions modify this density as compared to the noninteracting case $W = 0$. In the large $N$ limit and in the absence of interactions, the density in the bulk reads $\rho(x) \simeq \frac{1}{\pi}\sqrt{2(\mu - V(x))_+}$ as given by LDA or semi-classical methods (we denote everywhere $(x)_+ = \max(x, 0)$). Note that the LDA works only for noninteracting fermions and in the bulk [34]. Here $\mu$ denotes the Fermi energy which is determined by the normalization condition $\int dx\, \rho(x) = N$ (e.g., $\mu \simeq N$ for the harmonic oscillator (HO) considered above). Note that for some integrable systems, the LDA may be improved as in Ref. [105] to include interactions. We will not explore this route here.

For the models in Table 1, the noninteracting case corresponds to $\beta = 2$. To obtain the density for arbitrary $\beta$ one can interpret the PDF (5), or equivalently (6), as the Boltzmann distribution for a gas of classical particles at unit temperature, with energy $U(\vec{x})$, or equivalently $F(\vec{\lambda})$. Using (7) and (8), in both cases, the interaction between these particles is logarithmic which corresponds to the $2d$ Coulomb interaction. In the large $N$ limit and in the presence of a confining potential, the equilibrium density is obtained by minimizing the corresponding energy. This Coulomb gas (CG) method has been widely used in the context of RMT. Rewriting (6) as $F(\vec{\lambda}) = \frac{\beta}{2}\left[\sum_{i=1}^{N}\frac{2V_0(\lambda_i)}{\beta} - 2\sum_{i<j}\log|\lambda_i - \lambda_j|\right]$, one immediately sees that the Coulomb gas result for a general $\beta$ coincides with that of a gas with $\beta = 2$ and a matrix potential $2V_0(\lambda)/\beta$. Using the known results for the average eigenvalue density, defined as $\sigma(\lambda) = \frac{1}{N}\sum_i\langle\delta(\lambda - \lambda_i)\rangle$, we can write, respectively for the G$\beta$E and WL$\beta$E

$$\sigma(\lambda) = \frac{1}{\sqrt{N}}\sigma_W\left(\frac{\lambda}{\sqrt{N}}\right), \quad \sigma_W(z) = \frac{\sqrt{(2 - z^2)_+}}{\pi}, \tag{10}$$

$$\sigma(\lambda) = \frac{1}{N}\sigma_{MP}\left(\frac{\lambda}{N}\right), \quad \sigma_{MP}(z) = \frac{1}{2\pi}\sqrt{\left(\frac{4 - z}{z}\right)_+}. \tag{11}$$

The subscripts 'W' and 'MP' stand for Wigner (semi-circle) and Marcenko-Pastur, respectively. Using the mapping to the fermions with

$$\rho(x) = N\lambda'(x)\sigma(\lambda(x)), \tag{12}$$

and $\lambda(x) = \sqrt{\frac{2}{\beta}}x$ and $\lambda(x) = \frac{2}{\beta}x^2$ for the G$\beta$E and WL$\beta$E respectively, we obtain the fermion density for the models in the first and third line of the Table 1 as

$$\rho(x) \simeq \frac{2}{\pi\beta}\sqrt{(N\beta - x^2)_+}\ , \quad V(x) = \frac{x^2}{2}, \tag{13}$$

$$\rho(x) \simeq \frac{2\,\theta(x)}{\pi\beta}\sqrt{(2N\beta - x^2)_+}\ , \quad V(x) = \frac{x^2}{2} + \frac{\gamma^2 - \frac{1}{4}}{2x^2},$$

where $\theta(x)$ is the Heavisde function. The positions of the two edges are thus $x = x_e^\pm \simeq \pm\sqrt{\beta N}$ in the first case, while $x_e^- \simeq 0$ and $x_e^+ \simeq \sqrt{2\beta N}$ in the second one. For $\beta = 2$ it agrees with the LDA result, and it shows that the Fermi gas expands for $\beta > 2$ and shrinks for $\beta < 2$, as compared to the noninteracting case $\beta = 2$, while retaining a semi-circular shape. In the case of the box, corresponding to the J$\beta$E, the density is uniform in the large $N$ limit. Note that in the large $N$ limit, with $\gamma = O(1)$, the $1/x^2$ part of the potential in the third and fourth models in the Table 1 do not affect the bulk density (for a different scaling see below). They become important only in the region close to the wall (see below).

## 1.3 Outline and main results

In this paper we study the statistics of $\mathcal{N}_{\mathcal{I}}^{(\beta)}$, i.e., the number of fermions in an interval $\mathcal{I}$, for the models in the Table 1 for any $\beta \geq 1$, and, in a second stage, for a larger class of models.

In parallel to the applications to fermions we also obtain new results in the corresponding random matrix ensembles, with a slightly larger domain of validity, i.e., for any $\beta > 0$.

In Section 2 we study the variance of $\mathcal{N}_{[a,b]}^{(\beta)}$ for an interval $\mathcal{I} = [a,b]$ of macroscopic size in the bulk for $\beta = 1, 2, 4$ in the large $N$ limit, and then propose an extension to any $\beta$. In all these cases the variance grows logarithmically with $N$ for large $N$, and we obtain the amplitude of the logarithm together with the $O(1)$ correction term which has a non trivial dependence on the two edges $a, b$ on macroscopic scales. In the noninteracting case $\beta = 2$ there exists a formula, recalled here in Eq. (24) for the variance $\mathrm{Var}\mathcal{N}_{[a,b]}^{(\beta=2)}$ for a general potential $V(x)$. For the harmonic potential $V(x) = \frac{x^2}{2}$, which corresponds to G$\beta$E, we extend this formula to $\beta = 1, 2, 4$, and it reads

$$
\begin{aligned}
\frac{\beta \pi^2}{2} \mathrm{Var}\mathcal{N}_{[a,b]} \;=\; & \log N + \frac{3}{4} \log\big[\big(1 - \tilde{a}^2\big)\big(1 - \tilde{b}^2\big)\big] \\
+ \; & \log\left| \frac{4|\tilde{a} - \tilde{b}|}{1 - \tilde{a}\tilde{b} + \sqrt{(1 - \tilde{a}^2)(1 - \tilde{b}^2)}} \right| + c_\beta + o(1),
\end{aligned}
\tag{14}
$$

where $\tilde{a} = a/\sqrt{\beta N}$ and $\tilde{b} = b/\sqrt{\beta N}$, $|\tilde{a}|, |\tilde{b}| < 1$, where $\pm\sqrt{\beta N}$ are the positions of the two edges, as can be seen in Eq. (13). For $\beta = 1, 2, 4$ the constant $c_\beta$ takes the values

$$
c_1 = \log 2 + \gamma_E + 1 - \frac{\pi^2}{8}, \quad c_2 = \log 2 + \gamma_E + 1,
\tag{15}
$$

$$
c_4 = 2\log 2 + \gamma_E + 1 + \frac{\pi^2}{8}.
\tag{16}
$$

Here we argue that formula (14) extends to the model (3) of interacting fermions with general $\beta$ in the harmonic potential. Using related works [74, 75] (see discussion below) we propose the following expression as a series representation for $c_\beta$

$$
c_\beta = \gamma_E + \log \beta + \sum_{q=1}^{\infty} \left[ \frac{2}{\beta} \psi^{(1)}\left(\frac{2q}{\beta}\right) - \frac{1}{q} \right],
\tag{17}
$$

where here and below $\psi^{(k)}(z) = \frac{d^{k+1}}{dz^{k+1}} \log \Gamma(z)$ is the polygamma function. We have checked numerically the predictions (14), (17) for the variance (see Fig. 4).

In fact, going beyond the harmonic potential, our more general prediction for the models in Table 1 reads at large $N$ and in the bulk[3]

$$
\frac{\beta \pi^2}{2} \mathrm{Var}\mathcal{N}_{[a,b]}^{(\beta)} - c_\beta = \pi^2 \mathrm{Var}\mathcal{N}_{[a',b']}^{(\beta=2)} - c_2 + o(1),
\tag{18}
$$

where $a' = a\sqrt{2/\beta}$ and $b' = b\sqrt{2/\beta}$ for the models in line 1 and 3 in the Table 1 and $a' = a$ and $b' = b$ for the other two models (on a circle and in a box). On the right hand side of Eq. (18), $\mathrm{Var}\mathcal{N}_{[a,b]}^{(\beta=2)}$ is the variance for noninteracting fermions (i.e., for $\beta = 2$) in the presence of a potential $V(x)$ indicated in the Table 1 and given by the general formula (24) (which takes simpler forms for the models in the Table 1). The constant $c_\beta$ is independent of the model, and (18) also holds on microscopic scales in the limit of large interval (as in Eq. (2)). Finally, in Section 2 we also analyze the case of a "semi-infinite" interval, i.e., $[a, +\infty)$ for the G$\beta$E, $[0, b]$ for WL$\beta$E and J$\beta$E, for which we have a similar prediction. One consequence of our

---

[3]For the WL$\beta$E, $\gamma$ on both sides of Eq. (18) is different. Including the $\gamma$ dependence explicitly, the equation should read $\frac{\beta \pi^2}{2} \mathrm{Var}\mathcal{N}_{[a,b]}^{(\beta,\gamma)} - c_\beta = \pi^2 \mathrm{Var}\mathcal{N}_{[a',b']}^{(\beta=2,\gamma')} - c_2 + o(1)$ where $\gamma' = 2\gamma/\beta$, so that $\tilde{\gamma} = 2\gamma/(N\beta)$ is the same on both sides of the equation.

prediction (18) is that in the microscopic limit ($|a-b|$ small compared to the size of the Fermi gas) one has[4]

$$\text{Var}\,\mathcal{N}_{[a,b]} \simeq \frac{2}{\beta\pi^2}\left[\log\left(k_F(a)|b-a|\right)+c_\beta\right],\tag{19}$$

for $k_F(a)|b-a|=O(1)\gg 1$.

In Section 3 we study the higher cumulants of $\mathcal{N}_{[a,b]}$. We present the following conjecture for interacting fermions. Consider an interval $[a,b]$ inside the bulk. For the models displayed in Table 1 and for any $\beta$, the cumulants of $\mathcal{N}_{[a,b]}^{(\beta)}$ of order 3 and higher are determined solely from the microscopic scales. This implies that these higher cumulants are identical to those of the C$\beta$E. The cumulants for the C$\beta$E have been given in [74], using yet another conjecture about extended Fisher-Hartwig asymptotics for C$\beta$E, formulated in [75]. We will thus use these formulae and obtain here the full counting statistics for a larger class of interacting fermion models. These cumulants for general $\beta$ admit the following series representations

$$\left\langle\left(\mathcal{N}_{[a,b]}^{(\beta)}\right)^{2p}\right\rangle^c = \frac{2}{(2\beta\pi^2)^p}\tilde{C}_{2p}^{(\beta)},\tag{20}$$

$$\tilde{C}_{2p}^{(\beta)} = (-2)^{p+1}\frac{1}{\beta^p}\sum_{q=1}^{\infty}\psi^{(2p-1)}\left(\frac{2q}{\beta}\right),\tag{21}$$

for arbitrary integer $p>1$, while the odd cumulants vanish. For $\beta\in\{1,2,4\}$ the explicit evaluation for the fourth cumulant gives

$$\tilde{C}_4^{(\beta=2)} = -12\zeta(3),\quad \tilde{C}_4^{(\beta=1)} = \frac{\pi^4}{4}-24\zeta(3),\quad \tilde{C}_4^{(\beta=4)} = -24\zeta(3)-\frac{\pi^4}{4},\tag{22}$$

where $\zeta(z)$ is the Riemann-zeta function. The conjecture extends naturally to the case of an interval with only one point in the bulk (i.e., for a "semi-infinite" interval). It is a natural extension of the conjecture previously formulated in Ref. [52] for noninteracting fermions $\beta=2$ and recalled in the introduction. One can check that formula (20) reduces to Eq. (23) in [52] in the case $\beta=2$.

This conjecture can be checked in a few cases, with impressive agreement. For instance in Section 4 we study the limit from the bulk to the edge for any $\beta$. In the case $\beta\in\{1,2,4\}$ we can compare with the results of Bothner and Buckingham [77] obtained by Riemann-Hilbert methods. We find that it agrees in a quite non-trivial way.

In Section 5 we discuss the approaches to interacting fermions using bosonisation in terms of the Luttinger liquid. As we explain, the Luttinger parameter is given here by $K=2/\beta$ for the models in Table 1.

Finally in Section 6 we present a more general class of interacting fermion models which have a ground state wave function of the one- and two-body form (5). Some of these models still map onto random matrices, with however a more general matrix potential $V_0$. We study in detail the example of fermions on the circle in the presence of an external periodic potential which, in the noninteracting case $\beta=2$ turns out to be related to the so-called Gross-Witten-Wadia model in high energy physics [78,79]. The density in this model exhibits an interesting transition, and we show that the LDA formula in that case reproduces the well known results obtained in Refs. [78,79] from the Coulomb gas method. As discussed there we expect that our results for the counting statistics extend to this more general class of models.

---

[4]In a recent work [76], the leading logarithmic term in Eq. (19) was proven rigorously.

## 2 Number variance

### 2.1 Previous results for noninteracting fermions ($\beta = 2$) in an external potential

In a recent work [52] we have calculated the variance of the number of fermions in a domain $\mathcal{D}$ in $d = 1$, for noninteracting fermions in their ground state in a general potential $V(x)$. In this case the positions of the fermions form a determinantal point process. This means that the $n$-point correlation function can be written as a $n \times n$ determinant built from the so-called kernel $K_\mu(x, y)$. As a result the variance can be computed from the following formula [41, 71]

$$\text{Var}\mathcal{N}_{\mathcal{D}} = \int_{x \in \mathcal{D}} \int_{y \in \bar{\mathcal{D}}} dx dy K_\mu(x, y)^2 , \tag{23}$$

in terms of the kernel. By plugging the large-$N$ asymptotic form of the kernel (given by the WKB expansion of the eigenstates of the single-particle Hamiltonian) into (23) we obtained the leading- and subleading-order terms in the number variance, which are generically of order $O(\log N)$ and $O(1)$ respectively. Consider a confining potential, such that the bulk density $\rho(x) = k_F(x)/\pi$, where $k_F(x) = \sqrt{2(\mu - V(x))}$ is the local Fermi wave vector, has a single support $[x_e^-, x_e^+]$. For an interval $[a, b]$ in the bulk with $|a - b| \gg 1/k_F(a)$, we obtained that for $N \gg 1$ (i.e., $\mu \gg 1$) the variance is given by [52]

$$(2\pi^2)\text{Var}\mathcal{N}_{[a,b]} = 2 \log \left( 2k_F(a)k_F(b) \int_{x^-}^{x^+} \frac{dz}{\pi k_F(z)} \right)$$
$$+ \log \left( \frac{\sin^2 \frac{\theta_a - \theta_b}{2}}{\sin^2 \frac{\theta_a + \theta_b}{2}} |\sin \theta_a \sin \theta_b| \right) + 2c_2 + o(1) , \tag{24}$$

$$\text{where} \quad \theta_x = \pi \frac{\int_{x^-}^{x} dz/k_F(z)}{\int_{x^-}^{x^+} dz/k_F(z)}, \quad \begin{cases} \theta_{x^-} = 0 \\ \theta_{x^+} = \pi \end{cases} , \tag{25}$$

and $c_2$ is given in (15). For a semi-infinite interval, and for any $a$ in the bulk, the variance reads [52]

$$\text{Var}\mathcal{N}_{[a,+\infty)} \simeq \frac{1}{2\pi^2} \left( \log \frac{2k_F(a)^2 \sin \theta_a}{d\mu/dN} + c_2 \right). \tag{26}$$

For the harmonic potential $V(x) = \frac{1}{2}x^2$ this gives the explicit expression for the variance for the semi-infinite interval [52]

$$\text{Var}\mathcal{N}_{[a,+\infty)} = \text{Var}\mathcal{N}_{(-\infty,a]} = \frac{1}{2\pi^2} \left[ \log \mu + \frac{3}{2} \log(1 - \tilde{a}^2) + c_2 + 2\log 2 + o(1) \right], \tag{27}$$

where $\tilde{a} = \frac{a}{\sqrt{2\mu}}$. For an interval in the bulk, Eqs. (24) and (25) lead to the formula given in (14) with $\beta = 2$. Note that one can eliminate the Fermi energy $\mu$ in all above formula and express all quantities as a function of the number of fermions $N$ using the relation

$$N = \int dx \rho(x) \simeq \frac{1}{\pi} \int dx \sqrt{2(\mu - V(x))_+} , \tag{28}$$

valid at large $N$. Since the Fermi energy does not have a direct meaning for interacting fermions, it is indeed more natural to use $N$, in order to study the dependence in $\beta$, *at fixed $N$*, as we do below. Note that in the interacting case, the zero-temperature chemical potential can be obtained in the large $N$ limit as $\mu = \partial_N E_0$ where $E_0 = E_0(N, \beta)$ is the ground state energy (which, in the models studied here can be calculated exactly, see Appendix A). For $\beta = 2$ this definition of $\mu$ coincides with the Fermi energy.

We will now compute the number variance for interacting fermions, $\beta \neq 2$. For this purpose we recall the definition of a more general observable, the covariance function.

## 2.2 Two-point covariance function

It is useful to define the two point covariance function $C(x, y)$ which gives the covariance between the numbers of particles in infinitesimal intervals around two distinct points $x$ and $y$:

$$\text{Cov}\left(\mathcal{N}_{[x,x+dx]}, \mathcal{N}_{[y,y+dy]}\right) = C(x, y)\, dx\, dy. \tag{29}$$

Using the linearity of the covariance, one immediately obtains, for any two nonintersecting domains $\mathcal{D}_1$ and $\mathcal{D}_2$

$$\text{Cov}\left(\mathcal{N}_{\mathcal{D}_1}, \mathcal{N}_{\mathcal{D}_2}\right) = \int_{x \in \mathcal{D}_1} \int_{y \in \mathcal{D}_2} C(x, y)\, dx\, dy. \tag{30}$$

Using that $N = \mathcal{N}_{\mathcal{D}} + \mathcal{N}_{\bar{\mathcal{D}}}$, where $N$ is the total number of fermions, and where $\bar{\mathcal{D}}$ is the complement of $\mathcal{D}$, we obtain a convenient expression for the number variance in a domain:

$$\text{Var}\mathcal{N}_{\mathcal{D}} = -\int_{x \in \mathcal{D}} \int_{y \in \bar{\mathcal{D}}} C(x, y)\, dx\, dy. \tag{31}$$

For noninteracting fermions, the determinantal structure can be used in order to express the covariance function in terms of the kernel, $C(x, y) = -K_{\mu}(x, y)^2$ and one recovers the formula in (23).

## 2.3 Number variance for interacting fermions in a harmonic trap

Let us now discuss the case of interacting fermions described by the model (3), which corresponds to random matrices in the G$\beta$E. For interacting fermions, the positions of the fermions do not form a determinantal point process, however for $\beta = 1, 4$ they exhibit a Pfaffian structure which allows for (complicated) exact expressions for $C(x, y)$ for any finite $N$ (see e.g. [41, 71, 80, 81]). Here, to calculate the variance for $\beta = 1, 2, 4$, we will only need their large $N$ asymptotics. We will first recall their expressions separately for the case of microscopic scales $|x - y| = O(1/k_F(x))$ and macroscopic scales $|x - y| \gg 1/k_F(x)$. Indeed, when computing the integral in (31), as we do below, there are contributions from both regimes of scales.

(i) *Microscopic scales*. For $x$ and $y$ in the bulk with $x - y$ microscopic, $C(x, y)$ can be obtained from the Eqs. (18)-(20) in [82] using the mapping from the Gaussian ensembles to the fermions in the harmonic potential (the same formula also holds, see Eqs. (18) and (19) in [82], for the circular ensembles, i.e., for fermions on the circle)

$$C(x, y) \simeq -[\rho(x)]^2 Y_{2\beta}(\rho(x)|x - y|), \tag{32}$$

$$Y_{21}(r) = (s(r))^2 - \text{Js}(r)\,\text{Ds}(r), \tag{33}$$

$$Y_{22}(r) = (s(r))^2, \tag{34}$$

$$Y_{24}(r) = (s(2r))^2 - \text{Is}(2r)\,\text{Ds}(2r), \tag{35}$$

where $\rho(x)$ is the fermion density given in (13), and

$$s(r) = \frac{\sin(\pi r)}{\pi r}, \quad \text{Ds}(r) = \frac{ds}{dr} = \frac{\pi r \cos(\pi r) - \sin(\pi r)}{\pi r^2}, \tag{36}$$

$$\text{Is}(r) = \int_0^r s(r')\, dr' = \frac{\text{Si}(\pi r)}{\pi}, \quad \text{Js}(r) = \text{Is}(r) - \frac{\text{sgn}(r)}{2}, \tag{37}$$

where $\text{sgn}(r)$ is the sign function and $\text{Si}(z) = \int_0^z \frac{\sin t}{t}\, dt$ is the sine integral. Note that for $\beta = 2$ the positions of the fermions form a determinantal process, with an associated kernel given in the bulk by the function $s(r)$ which is the well-known sine-kernel.

(ii) *Macroscopic scales.* For $x$ and $y$ well separated in the bulk, and for the G$\beta$E for arbitrary $\beta$ the covariance function $C(x, y)$ is also known in the large $N$ limit [83–86]. Under the RMT to fermion mapping it leads to

$$C(x, y) \simeq -\frac{1 - \frac{xy}{\beta N}}{\beta \pi^2 (x - y)^2 \left(1 - \frac{x^2}{\beta N}\right)^{1/2} \left(1 - \frac{y^2}{\beta N}\right)^{1/2}}, \tag{38}$$

up to rapidly oscillating terms that average out to zero when integrating over macroscopic domains. For noninteracting fermions $\beta = 2$, Eq. (38) was also derived directly from the fermion model, see the Supp. Mat. of [52] (for a numerical check of this formula see [87]). Using a Coulomb gas method Eq. (38) was extended to arbitrary *matrix* potentials and $\beta$ in [84]. One can check that the above formula match between microscopic and macroscopic scales, i.e., that the limit $r \gg 1$ in (32) agrees with the limit $|x - y| \ll \sqrt{\beta N}$ in (38).

The double integral (31) can then be calculated in the large-$N$ limit. The calculation is performed in the Appendix C. First one can approximate $C(x, y) \simeq 0$ if either $x$ or $y$ are not in the bulk. Plugging in $C(x, y)$ from (32) for $x$ near $y$, and (38) for $x$ far from $y$ (this procedure works because there is a joint regime where both of the approximate expressions for $C(x, y)$ are valid) one finds the result for the variance given in (14) for $\beta \in \{1, 2, 4\}$. For instance, for a finite interval $[-a, a]$ centered around the origin and contained in the bulk, $\tilde{a} = a/\sqrt{\beta N} < 1$, Eq. (14) simplifies into

$$\frac{\beta \pi^2}{2} \text{Var}\left(\mathcal{N}_{[-a,a]}\right) \simeq \log\left[4N\tilde{a}\left(1 - \tilde{a}^2\right)^{3/2}\right] + c_\beta. \tag{39}$$

The leading term agrees with a Coulomb gas calculation for general $\beta$, [43, 44]. In the limit $|\tilde{b} - \tilde{a}| \ll 1$, Eq. (14) matches with the microscopic result (2).

We also obtain the result for a semi-infinite interval whose edge is in the bulk, $|\tilde{a}| < 1$ as

$$\beta \pi^2 \text{Var}\left(\mathcal{N}_{[a,\infty)}\right) \simeq \log N + \frac{3}{2}\log\left(1 - \tilde{a}^2\right) + 2\log 2 + c_\beta. \tag{40}$$

In both formulae (39) and (40) the constants $c_\beta$ for $\beta = 1, 2, 4$ are related to the so-called Dyson-Mehta constants and given in (15).

For $\beta = 1$ and $\beta = 4$, we have checked numerically the predictions given in (39) and (40) for the fermion model using the correspondence in the first line of the Table 1. We have performed exact diagonalizations of the G$\beta$E with $\beta = 1, 4$ in Figs. 1 and 2 respectively (the case $\beta = 2$ has been tested numerically in [52]). The convergence at large $N$ appears to be slower as $\beta$ is increased, which is known to occur quite generally, see e.g. Fig. 3.2 in [88].

## 2.4 Conjecture for general $\beta$ and results for other potentials

In the previous section, we have calculated the number variance for interacting fermions in the harmonic potential for $\beta = 1, 2, 4$. We now conjecture that the formula (14), (39) and (40) hold for general values of $\beta$, with an a priori unknown $\beta$-dependent constant $c_\beta$. The rationale behind this conjecture is that (i) the expression (38) for $C(x, y)$ on macroscopic scale is valid for arbitrary $\beta$, a result which comes naturally from the Coulomb gas calculations [43, 44, 84] (ii) the above calculations for $\beta = 1, 2, 4$ show that the $\beta$-dependence of the constant part, $c_\beta$, is determined from microscopic scales only. Hence we expect that it is independent of the RMT ensemble in the Table 1. As we argue below, this constant is given for general $\beta$ by formula (17), which we will justify in Section 3 based on previous works on the C$\beta$E.

The conjecture can be expressed as follows. For an arbitrary interval $[a, b]$ in the bulk and for the fermion models listed in Table 1, there is a relation between the variance for an

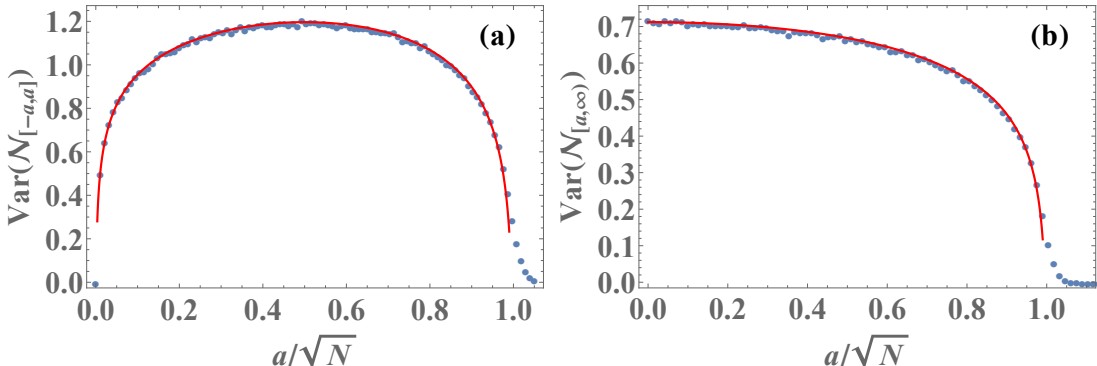

Figure 1: Variance of the number of fermions for finite intervals centered around the origin (a) and semi-infinite intervals (b) for the model in the first line of Table 1 (with quadratic potential) associated to the GOE ($\beta = 1$). The blue markers are the empirical variance computed over $5 \times 10^4$ simulated GOE matrices with $N = 100$, and the red lines are our predictions (with $\beta = 1$) (39) in (a), and (40) in (b).

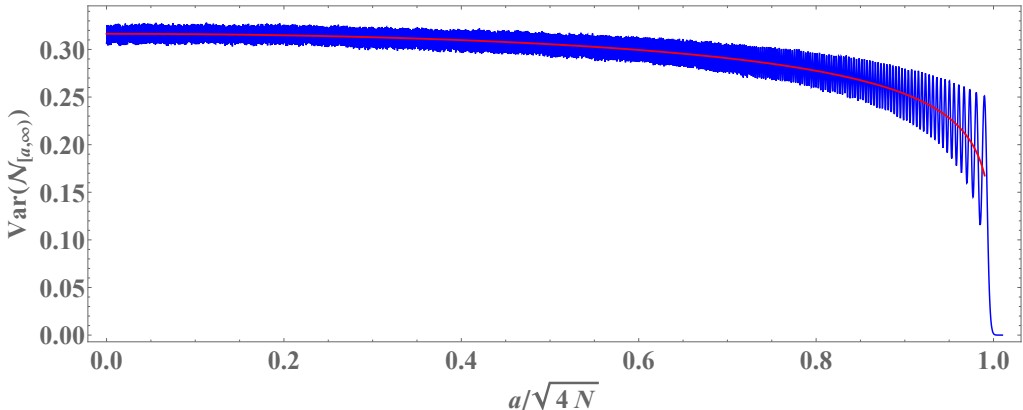

Figure 2: Variance of the number of fermions in a semi-infinite interval for the model in the first line of Table 1 associated to the GSE ($\beta = 4$). The blue line is the empirical variance computed over $2 \times 10^5$ simulated GSE matrices with $N = 1000$, and the red line is our prediction (with $\beta = 4$) (40).

arbitrary $\beta \geq 1$ and the variance for $\beta = 2$ (up to a possible rescaling of lengths). This relation is given in (18) above.

Let us now present a few results which follow from this conjecture. For fermions on the circle with $L = 2\pi$ this leads to

$$\beta \pi^2 \mathrm{Var} \mathcal{N}_{[a,b]} = 2 \log N + \log \left( \sin^2 \frac{b-a}{2} \right) + 2c_\beta . \tag{41}$$

In the microscopic limit, this formula agrees with (19) with $k_F = \pi \rho = N/2$.

Consider now interacting fermions related to the WL$\beta$E in the potential (line 3 in Table 1)

$$V(x) = \frac{\gamma^2 - \frac{1}{4}}{2x^2} + \frac{1}{2} x^2 . \tag{42}$$

In the introduction, we have discussed this model when the parameter $\gamma = O(1)$ in which case the $1/x^2$ hard wall potential does not affect the bulk properties for $x > 0$, such as the density given in Eq. (13). Another interesting limit amounts to scale the parameter $\gamma \sim \mu \sim N$ in which case the effect of the $1/x^2$ potential is to open a gap in the density of fermions

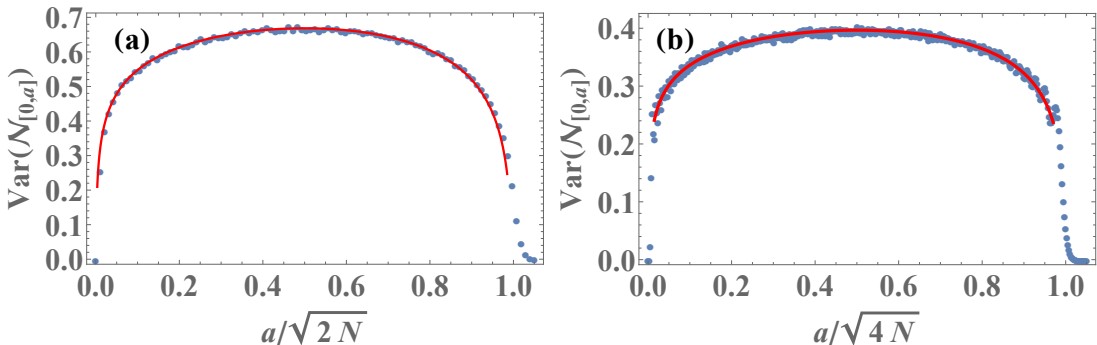

Figure 3: Variance of the number of particles in the interval $[0, a]$ (or equivalently, $[a, \infty)$) for the model in the third line of Table 1 associated to the WL$\beta$E with $\gamma = 2$, and $\beta = 1$ (a) and $\beta = 2$ (b). The blue markers are the empirical variance computed over $5 \times 10^4$ simulated WL$\beta$E matrices with $N = 100$, and the red lines are our theoretical prediction (48).

near the origin. Let us recall that the associated Wishart-Laguerre (WL) matrix potential is $V_0(\lambda) = \frac{\beta}{2}\lambda - \gamma \log \lambda$. By a similar argument as in Eq. (10) and below, one can absorb the $\beta$ dependence in the product $\frac{2}{\beta}V_0(\lambda)$, by rescaling $\gamma$. As a result the eigenvalue density $\sigma(\lambda; \gamma)$ for the WL$\beta$E takes the scaling form at large $N$ (as obtained e.g. from the Coulomb gas method)

$$\sigma(\lambda; \gamma) \simeq \frac{1}{N} \sigma_{WL}\left(\frac{\lambda}{N}; \frac{2\gamma}{\beta N}\right), \tag{43}$$

where

$$\sigma_{WL}(z; c) = \frac{\sqrt{(z - \zeta_-)(\zeta_+ - z)}}{2\pi z}, \quad \zeta_\pm = (1 \pm \sqrt{1+c})^2. \tag{44}$$

The normalization condition reads $\int_{\zeta_-}^{\zeta_+} dz \, \sigma_{WL}(z; c) = 1$, where the two scaled edges of the support $\zeta_\pm$ depend on the parameter $c$. Using the mapping to the fermions, with $\lambda = \frac{2}{\beta}x^2$ we obtain the fermion density [see (10)] as

$$\rho(x) \simeq \frac{4x}{\beta} \sigma_{WL}\left(\frac{2x^2}{\beta N}; \frac{2\gamma}{\beta N}\right). \tag{45}$$

For $\beta = 2$, one can check that this result coincides with the prediction from the LDA in the bulk as expected, i.e.,

$$\rho(x) \simeq \frac{1}{\pi}\sqrt{2(\mu - V(x))}, \tag{46}$$

together with the relation between $\mu$ and $N$, which reads $\mu = 2N + \gamma + \frac{1}{2} \simeq 2N + \gamma$ in the large $N$ limit, with $\gamma = O(N)$ considered here. The prediction (45) allows to obtain the density for interacting fermions for general $\beta$ in the potential (42). It is interesting to note in the above result that the gap in the fermion density near the origin remains non-zero for any value of the interaction parameter $\beta = O(1)$. In the limit $\gamma/N \to 0$ one recovers the result in (13).

One can now use our main conjecture (18) and its analog

$$\beta \pi^2 \text{Var} \mathcal{N}_{[0,a]}^{(\beta)} - c_\beta = 2\pi^2 \text{Var} \mathcal{N}_{[0,a]}^{(\beta=2)} - c_2 + o(1), \tag{47}$$

for semi-infinite intervals, and the result that we obtained in [52] for the case $\beta = 2$, to predict the variance of the number of fermions in an interval for general $\beta$ for the potential (42). For

the interval $[0,a]$ with $a$ in the bulk, this leads to

$$\beta \pi^2 \mathrm{Var} \mathcal{N}_{[0,a]} \simeq \log(8N) + c_\beta + \log\left( \tilde{a} \sqrt{1 + \frac{\tilde{\gamma}}{2}} \frac{\left(1 - \frac{\tilde{a}^2}{1+\frac{\tilde{\gamma}}{2}} - \frac{\tilde{\gamma}^2}{8\tilde{a}^2(2+\tilde{\gamma})}\right)^{3/2}}{\left(1 - \frac{\tilde{\gamma}^2}{(2+\tilde{\gamma})^2}\right)^{1/2}} \right), \qquad (48)$$

where $\tilde{\gamma} = 2\gamma/(N\beta)$ and $\tilde{a} = \frac{a}{\sqrt{2\beta N}}$ (which is the position of the edge at $\tilde{\gamma} = 0$). The theoretical prediction (48) is compared with numerical simulations of Wishart matrices in Fig. 3 with $\gamma = 2$ and $\beta \in \{1,2\}$, with excellent agreement. A formula analogous to (48) for a general interval $[a,b]$ with $a, b$ in the bulk is given in Appendix D, where we also give some formula for the Jacobi box potential (line 4 in Table 1) as well as the details of the derivation of (48).

## 3 Higher cumulants

We now study the higher cumulants (larger than 2) of the number of fermions in an interval for the interacting fermion models displayed in Table 1. In our previous work for noninteracting fermions in [52] we had conjectured, and checked with available rigorous results for several potentials, that the higher cumulants are determined solely from microscopic scale. Hence they are independent of the potential in the large $N$ limit. Here we will go one step further and conjecture that this remains true in the interacting case for general $\beta$. Although the numerical values of these cumulants depend non trivially on $\beta$, i.e., on the interaction strength, they are insensitive to the details of an external smooth potential. Indeed these cumulants are determined at microscopic scales where the $1/x^2$ interactions dominate over the local variations of the potential. Consequently we can conjecture that the higher cumulants are the same as for fermions on a circle without a potential, i.e for the C$\beta$E.

It turns out that the cumulants for the C$\beta$E were recently predicted in Ref. [74] in a different context. Consider the periodic model in the second line of Table 1 where $x$ is the coordinate along a circle of perimeter $L = 2\pi$. The result of [74] gives the FCS generating function for $\mathcal{N}_{[a,b]}$, i.e., the number of fermions with positions $x_i \in [a,b]$ as[5]

$$\log\left\langle e^{2\pi\sqrt{\frac{\beta}{2}}t(\mathcal{N}_{[a,b]} - \langle\mathcal{N}_{[a,b]}\rangle)} \right\rangle = 2t^2 \log N + t^2 \log\left(4\sin^2 \frac{|b-a|}{2}\right) + 2\log|A_\beta(t)|^2, \qquad (49)$$

up to terms that vanish in the large $N$ limit. Here $t$ is a parameter,[6] and for $\beta = 2s/r$, with $s, r$ integers mutually prime

$$A_\beta(t) = r^{-t^2/2} \prod_{\nu=0}^{r-1} \prod_{p=0}^{s-1} \frac{G\left(1 - \frac{p}{s} + \frac{\nu + it\sqrt{\frac{2}{\beta}}}{r}\right)}{G\left(1 - \frac{p}{s} + \frac{\nu}{r}\right)}, \qquad (50)$$

where $G(z)$ is the Barnes function [89]. This formula is based on yet another conjecture made in [75] (see formula (3.22)-(3.23) there[7]).

---

[5]Note the misprint in formula (21) in [74], both in arXiv and published versions.

[6]Note that Eq. (49) is valid for real $t$ but can be extended for complex $t$ in the neighborhood of $t = 0$ with the replacement $|A_\beta(t)|^2 \to A_\beta(t)A_\beta(-t)$.

[7]The result (49), (50) is obtained from Eq. (3.22)-(3.23) in [75] in the case $R = 2$ with $b_1 = -b_2 = \sqrt{2}\beta i b$ and $a_1 = a_2 = 0$ (see notations there). Note the misprint $G^2 \to G$ in formula (3.21) in [75] (both in ArXiv and published versions). For $V(x) \neq 0$ the exact energy levels are not always doubly degenerate, e.g. a delta impurity at $x = 0$ breaks the degeneracy between odd (sinus) and even (cosinus) wavefunctions. Note that the ground state is never degenerate. However this effect goes beyond the semi-classical approximation.

From (49) expanding on both sides in powers of $t$ one finds that the cumulants $\langle \mathcal{N}_{[a,b]}^k \rangle^c$ of order $k > 2$ take the form

$$\left\langle \mathcal{N}_{[a,b]}^k \right\rangle^c = \frac{2}{\left( \pi \sqrt{2\beta} \right)^k} \tilde{C}_k^{(\beta)} + o(1) , \tag{51}$$

where the coefficients $\tilde{C}_k^{(\beta)}$ are defined for $k \geq 2$ as

$$\tilde{C}_k^{(\beta)} = \frac{d^k}{dt^k} \bigg|_{t=0} \log \left( A_\beta(t) A_\beta(-t) \right) . \tag{52}$$

It is obvious from this formula that $\tilde{C}_{2p+1}^{(\beta)} = 0$ hence all the odd cumulants vanish, i.e., $\langle \mathcal{N}_{[a,b]}^{2p+1} \rangle^c = 0$. We thus focus now on the even cumulants. Although the coefficients $\tilde{C}_k^{(\beta)}$ are defined here for rational values of $\beta$, it is possible to obtain expressions for these coefficient for any real $\beta$, as an explicitly continuous function of $\beta$. This is achieved using the fact that any real $\beta$ can be reached by a sequence $\beta = 2s_n/r_n$ of arbitrary large $s_n, r_n$ and performing an asymptotic analysis (see details in [74]). The result for $k = 2p$ with $p \geq 2$ can be written in terms of the following double series

$$\tilde{C}_{2p}^{(\beta)} = (-1)^{p+1} 2 (2p-1)! \sum_{v=0}^{\infty} \sum_{q=1}^{\infty} \frac{1}{\left( v\sqrt{\frac{\beta}{2}} + q\sqrt{\frac{2}{\beta}} \right)^{2p}} . \tag{53}$$

In addition, one of the sums (either over $v$ or over $q$) can be carried out, leading to two equivalent "dual" expressions

$$\begin{aligned} \tilde{C}_{2p}^{(\beta)} &= (-2)^{1-p} \beta^p \sum_{v=0}^{\infty} \psi^{(2p-1)} \left( 1 + \frac{\beta v}{2} \right) \\ &= (-2)^{p+1} \frac{1}{\beta^p} \sum_{q=1}^{\infty} \psi^{(2p-1)} \left( \frac{2q}{\beta} \right) , \end{aligned} \tag{54}$$

where we recall that $\psi^{(q)}(x) = \frac{d^{q+1}}{dx^{q+1}} \log \Gamma(x)$ is the polygamma function. The above series are convergent for $p \geq 2$, since at large $x$ one has $\psi^{(2p-1)}(z) \simeq \frac{(2p-2)!}{z^{2p-1}}$. The asymptotics for small and large $\beta$ can be obtained from either of the dual series in (54) and can be found in [74]. For the classical values $\beta \in \{1, 2, 4\}$ these series can be performed explicitly, e.g. see formula (22) for the fourth cumulant. Note that the formula (53) transforms simply under the "duality" $\beta \to 4/\beta$. This duality was studied in [90].

From our conjecture, the formula (51) for the cumulants of the fermion model on the circle (i.e., the C$\beta$E) is thus predicted to hold for all the fermion models in Table 1, with no modification. Indeed the rescaling of lengths is unimportant here since the values of these cumulants are independent of the size of the intervals (assumed here to be macroscopic in the bulk). In the case of a semi-infinite interval, e.g. $[a, +\infty)$ for the quadratic potential, the result is divided by a factor of 2.

We can now return to the question of the $O(1)$ term in the second cumulant (the variance) as discussed in the previous section. In particular the above predictions based on the C$\beta$E allow to obtain explicitly the universal constant $c_\beta$ which enters in all the formulae for the variance of the fermion models considered here. Comparing the formula (41) with the $O(t^2)$ term in (49) one finds that the relation (18) holds, together with $c_\beta = \log 2 + \frac{1}{2} \tilde{C}_2^{(\beta)}$, where

$\tilde{C}_2^{(\beta)}$ is given in (52). Using the analysis of $\tilde{C}_2^{(\beta)}$ in [74], the constant $c_\beta$ can be written in several alternative forms, either as a convergent double series

$$c_\beta = \log 2 + \gamma_E + \sum_{\nu=0}^{+\infty}\left[\sum_{q=1}^{+\infty}\frac{\beta/2}{\left(\nu\frac{\beta}{2}+q\right)^2} - \frac{1}{1+\nu}\right], \tag{55}$$

or, performing one of the sums, as a convergent simple series as given in the Introduction, see (17), or as the dual series

$$c_\beta = \log 2 + \gamma_E + \sum_{\nu=0}^{+\infty}\left[\frac{\beta}{2}\psi^{(1)}\left(1+\frac{\beta\,\nu}{2}\right) - \frac{1}{1+\nu}\right]. \tag{56}$$

We have tested the prediction for $c_\beta$ (17) numerically, together with the prediction for the variance (40), see Fig. 4. Using the correspondence in Table (1) we have diagonalized G$\beta$E matrices generated using the Dimitriu-Edelman tridiagonal matrices [72] for various sizes $N$. A rather large even-odd finite $N$ effect is observed, however the average value over consecutive $N$'s is very close to the predictions. Note that in Fig. 4 we tested these predictions also in the range $0 < \beta < 1$, which is only relevant for RMT (or log-gases) but not for the fermion models studied in the rest of this paper (since in the latter models $\beta \geq 1$). It would be interesting to test our predictions for the higher cumulants too. This is more computationally demanding, but nevertheless the fourth cumulant was tested numerically in [73] for the GUE.

## 4 FCS near the edge and matching with the bulk

Until now we have discussed the counting statistics in the large $N$ limit for an interval which has at least one point inside the bulk. Let us consider a general smooth potential $V(x)$ such that the Fermi gas has two edges $x \simeq x_e^{\pm}$, where the LDA density vanishes. There is a region near these edges, of width denoted $w_N$, where it is known that the quantum fluctuations are enhanced, and that the counting statistics are different from the bulk. While this region has been studied in the noninteracting case ($\beta = 2$), there are only a few recent results for the counting statistics in the edge region for the interacting case. They were obtained in the context of RMT, specifically for the G$\beta$E for $\beta = 1, 4$. This corresponds to interacting fermions in a quadratic potential. In Ref. [77] the FCS (i.e., all the cumulants) have been obtained, in the outer edge region, i.e., for an interval $[a, +\infty)$ where $\frac{x_+ - a}{w_N}$ is $O(1)$ but large (i.e., in the crossover region from the edge to the bulk). Inside the edge region, there are recent results about the second cumulant for general linear statistics for $\beta = 1, 4$ [92]. In this section we discuss how these results compare with our conjecture for the cumulants for general $\beta$. We start by recalling the noninteracting case and the matching between the bulk and the edge regions.

### 4.1 Noninteracting case $\beta = 2$

In the case of noninteracting fermions the positions $x_i$ form a determinantal point process based on the kernel $K_\mu(x, y)$ discussed in Section 2. For any smooth confining potential $V(x)$, as discussed in [34], for $x, y$ near the right edge $x^+$ (and similarly for $x^-$) the kernel takes the universal scaling form $K_\mu(x, y) \simeq \frac{1}{w_N}K_{\text{Ai}}\left(\frac{x-x^+}{w_N}, \frac{y-x^+}{w_N}\right)$ where $w_N$ is the width of the edge region $w_N = \left[2V'\left(x^+\right)\right]^{-1/3}$ for noninteracting fermions. Here $K_{\text{Ai}}$ is the Airy kernel given by

$$K_{\text{Ai}}(x, y) = \frac{\text{Ai}(x)\text{Ai}'(y) - \text{Ai}'(x)\text{Ai}(y)}{x - y}. \tag{57}$$

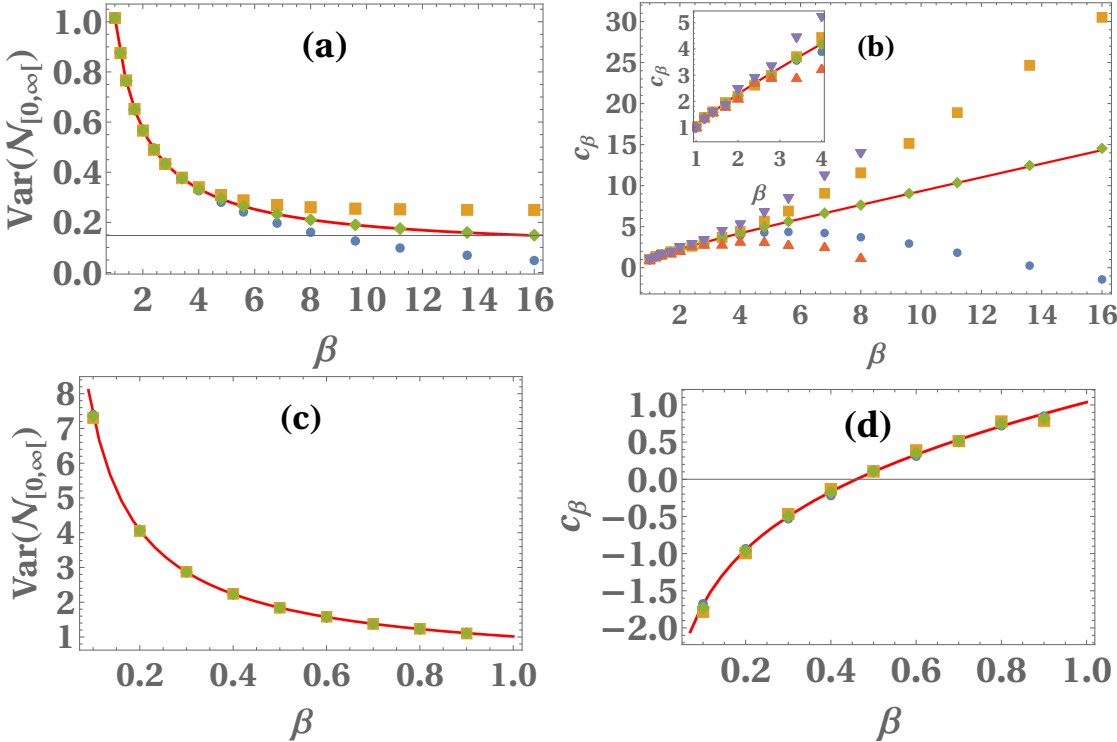

Figure 4: (a) Number variance for a semi-infinite interval $[0,\infty)$ for the harmonic oscillator (3) as a function of $\beta \geq 1$. The squares and circles correspond to numerical simulations of G$\beta$E with $N = 2001$ and $N = 2000$ respectively (with $10^5$ simulations for each plot marker), and their average is indicated by the diamonds. The red line is the conjecture given by (17) and (40) (with $\tilde{a} = 0$). The reason for averaging over two consecutive values of $N$ is because it appears that there is a parity effect, which comes from subleading corrections which we do not calculate here (for $\beta = 2$ these corrections are known and indeed depend on the parity of $N$ [91]). (b) Markers: Numerically computed $\beta \pi^2 \text{Var}(\mathcal{N}_{[0,\infty)}) - \log(4N)$. According to our conjecture (40), this should converge to the constant $c_\beta$ in the large-$N$ limit. The squares, circles, and diamonds are based on the same data as in (a). The triangles and upside-down triangles correspond to $10^5$ simulations with $N = 100$ and $N = 101$, respectively. Red line: our conjecture (17). One observes a (rather) slow convergence of the numerical results to our conjecture as $N$ is increased. In (c) and (d), analogous results are plotted for $0 < \beta < 1$ (and $N = 2000, 2001$), again with good agreement between the numerical results and our conjecture. These results are only relevant for RMT (or log-gases) but not for the fermion models studied in the rest of this paper (since in the latter models $\beta \geq 1$).

Using this scaling form one obtains the number variance for any interval in the edge region in terms of the Airy kernel as was done in [43, 44] for the case of the harmonic oscillator/GUE,

$$\text{Var}\mathcal{N}_{[a,+\infty)} = \int_a^{+\infty} dx \int_{-\infty}^a dy\, K_\mu^2(x,y) \simeq \frac{1}{2}\mathcal{V}_2(\hat{a}), \tag{58}$$

$$\mathcal{V}_2(\hat{a}) := 2\int_{\hat{a}}^{+\infty} du \int_{-\infty}^{\hat{a}} dv K_{\text{Ai}}^2(u,v)\,, \quad \hat{a} = \frac{a - x^+}{w_N}\,,$$

where the scaling function $\mathcal{V}_2(\hat{a})$, defined in [43, 44], is universal, i.e., independent of the potential $V(x)$, in terms of the scaling variable $\hat{a}$.

One interesting question is the matching of the number variance as $a$ in (58) moves from the bulk to the edge. In [52] it was shown that the asymptotic behavior for $a \to x^+$ coming from the bulk reads

$$\text{Var}\mathcal{N}_{[a,+\infty)} = \frac{1}{2\pi^2}\left[\frac{3}{2}\log(-\hat{a}) + c_2 + 2\log 2\right] + o(1),\tag{59}$$

in terms of the edge scaling variable $\hat{a}$ defined in (58). The result (59) matches exactly with the formula (58) in the limit $\hat{a} \to -\infty$, which corresponds to the crossover from the edge to the bulk. The comparison between the two results is performed in Appendix F. This crossover was also obtained in the case of the harmonic potential (i.e., for the GUE) in [77] (see also [93]). In fact in Ref. [77] the FCS, i.e., the higher cumulants were also given for $\beta = 2$ in this crossover regime. As we discuss below and in Appendix E, this result matches perfectly our predictions for the higher cumulants in the bulk given in [52].

## 4.2 Interacting case

Let us start with the case of the harmonic potential $V(x) = \frac{1}{2}x^2$. From the RMT-fermion correspondence in the first line of Table 1, i.e., $\lambda(x) = \sqrt{2/\beta}\,x$, one finds that for general $\beta$ the right edge is at position $x^+ = \sqrt{\beta N}$ and the width of the edge region is

$$w_N = \frac{\sqrt{\beta}}{2}N^{-1/6}.\tag{60}$$

It is known in RMT that the edge properties of the G$\beta$E are described by the Airy$_\beta$ point process denoted $a_i^\beta$, which implies that the fermion positions in the edge region and for $N \to +\infty$ can be written as

$$x_i = \sqrt{\frac{\beta}{2}}\lambda_i, \quad \lambda_i \simeq \sqrt{2N} + \frac{a_i^\beta}{\sqrt{2}N^{1/6}}.\tag{61}$$

The statistics of the $a_i^\beta$ is described by the so-called stochastic Airy operator [94–96]. It is known that the largest eigenvalue (i.e., the rightmost fermion) is described by the $\beta$- Tracy-Widom distribution $\text{Prob}(\max_i a_i^\beta < s) = F_\beta(s)$ which depends continuously on $\beta$. Explicit expressions in terms of Fredholm determinants or solutions to Painlevé equations are known for $\beta \in \{1, 2, 4\}$ [97, 98] (see also [77, 99, 100]). From known results for the mean density $\sigma(\lambda)$ of eigenvalues of G$\beta$E at the edge for $\beta \in \{1, 2, 4\}$ [101, 102] we obtain that the mean density for the fermion model takes the scaling form in the large $N$ limit

$$\rho(x) = N\sqrt{\frac{2}{\beta}}\sigma\left(\sqrt{\frac{2}{\beta}}x\right) \simeq \frac{1}{w_N}\sigma_\beta^e\left(\frac{x - \sqrt{\beta N}}{w_N}\right),\tag{62}$$

where the scaling functions are

$$\sigma_\beta^e(\xi) = \begin{cases} \sigma_2^e(\xi) + \frac{1}{2}\text{Ai}(\xi)\left[1 - \int_\xi^\infty \text{Ai}(t)\,dt\right], & \beta = 1, \\ \sigma_2^e(\xi) = \text{Ai}(\xi)^2 - \xi\text{Ai}'(\xi)^2, & \beta = 2, \\ \frac{1}{\sqrt{\kappa}}\left[\sigma_2^e(\kappa\xi) - \frac{1}{2}\text{Ai}(\kappa\xi)\int_{\kappa\xi}^\infty \text{Ai}(t)\,dt\right], & \beta = 4, \end{cases}\tag{63}$$

with $\kappa = 2^{2/3}$. The (smooth) linear statistics for general $\beta$ was studied in [103] in the limit towards the bulk. For the FCS for general $\beta$ however explicit formulae are still lacking in the edge region.

Concerning the variance of the particle number, the scaling form (58) can be extended to any $\beta$, see [43, 44]

$$\text{Var}\mathcal{N}_{[a,\infty)} \simeq \frac{1}{2}\mathcal{V}_\beta\left(\frac{a - x^+}{w_N}\right).\tag{64}$$

However the explicit form for general $\beta$ is unknown at present. The asymptotic behavior in the limit towards the bulk, $\hat{a} \to -\infty$ can also be extracted from the results in [77], see Appendix F for details. This leads to the asymptotic behaviors for $\beta \in \{1, 2, 4\}$

$$\mathcal{V}_\beta(\hat{a}) \simeq 2 \frac{\frac{3}{2}\log(-\hat{a}) + c_\beta + 2\log 2}{\beta \pi^2}, \qquad -\hat{a} \gg 1, \tag{65}$$

which matches with the bulk result (40) that we obtained above. The leading order (logarithmic) term in (65) was conjectured in [43] for any $\beta$ based on the expected matching with the bulk. From our conjecture in Section 2.4 we can now predict that (65) holds for general $\beta$, with the constant $c_\beta$ given in (17).

Concerning the cumulants of the particle number of order three and higher, these have been obtained in the limit towards the bulk ($\hat{a} \to -\infty$) for $\beta \in \{1, 2, 4\}$ in Ref. [77]. In Appendix E we have verified that our prediction for the higher cumulants (20) for general $\beta$ match perfectly with the results from Ref. [77] for $\beta \in \{1, 2, 4\}$. This provides a non trivial check of our conjecture, and involves not so well known identities among Barnes functions.

The above results were obtained for fermions in the harmonic potential, associated to the G$\beta$E. In RMT it is known that there is a universality at the soft edge, hence we expect the same behavior to hold for the fermion model associated to the WL$\beta$E (line 3 in Table 1) at its soft edge (i.e., near $x_e^+ = \sqrt{2\beta N}$). On the other hand, for that model near $x_e^- = 0$ (and for the Jacobi box) the universality is called the hard-edge and described by the Bessel stochastic operator [104].

As in the case $\beta = 2$ it is reasonable to conjecture that for general $\beta$ the above results extend to any smooth confining potential, e.g. $V(x) \sim |x|^p$ with $p > 0$, near the edge, the functions $\mathcal{V}_\beta$ being thus universal. The simple picture is that $V(x)$ near the edge can be approximated by a linear potential in all cases. One can also expect that the higher cumulants will also take a universal scaling form, being a non-trivial function of $\hat{a}$.

Note that the effect of short range interactions at the edge was discussed in Ref. [105] and found to be subdominant within the model studied there. It was also noticed there that the case of $1/x^2$ interactions lead to a new universality class for general $\beta$.

# 5 Bosonisation and Luttinger liquid

It is well known that interacting fermions in one dimension can be described by the effective theory of the Luttinger liquid (LL) based on the bosonisation method, for a review see e.g. [106]. It provides a hydrodynamic description which in its simplest form is valid in the absence of an external potential. At equilibrium, and for spinless fermions, it depends only on two parameters, the mean density $\rho_0$ and the dimensionless Luttinger parameter $K$, with $K = 1$ for noninteracting fermions, $K < 1$ for repulsive interactions and $K > 1$ for attractive interactions. The dynamics also depends on the "sound velocity" $v_F$ (Fermi velocity for free fermions). This is based on the description in terms of the phase field $\varphi(x)$ defined such that $\rho(x) = -\frac{1}{\pi}\partial_x \varphi(x)$, which at large scale is described by a Gaussian theory. This theory can be extended in the presence of a potential which varies very slowly on scales of the order of the inter-particle distance $1/\rho(x)$. Many recent studies have addressed the case of inhomogeneous bosonisation where $v_F$, $K$ and $\rho_0$ may become slowly varying functions of the position $x$ [54–56] (see also [107–109]).

Let us recall that for a LL the density correlations are given by

$$\langle \rho(x)\rho(0) \rangle \simeq \rho_0^2 \left[ 1 - \frac{2K}{(2\pi\rho_0 x)^2} + \sum_{m=1}^{+\infty} A_m (\rho_0|x|)^{-2Km^2} \cos(2\pi m \rho_0 x) \right], \tag{66}$$

while the correlation function of the fermionic field (which is the analogue of the kernel in the case of noninteracting fermions) reads

$$\langle \Psi^\dagger(x)\Psi(0)\rangle \simeq \rho_0 \sum_{m=0}^{+\infty} C_m (\rho_0 |x|)^{-\frac{1}{2K} - 2K(m+\frac{1}{2})^2} \sin\left(2\pi\left(m + \frac{1}{2}\right)\rho_0 x\right). \qquad (67)$$

These formulae (66) and (67) are valid for $\rho_0 x \gtrsim 1$. Here $A_1$ in (66) and $C_0$ in (67) represent the leading behaviors at large $\rho_0 x$, while the terms $A_m$, $m \geq 2$ and $C_m$, $m \geq 1$ represent the contributions of higher harmonics (often neglected in LL studies). For noninteracting (free) fermions, $K = 1$, $C_0 = \frac{1}{\pi}$ and all $C_m = 0$ for $m \geq 1$, and the expression in (67) becomes exact. In this case, this is precisely the sine kernel $\sin(\pi\rho_0 x)/(\pi x)$. In the presence of interactions we see that the correlation function of the fermionic field in (67) in the ground state now decays at large $x$ as

$$\langle \Psi^\dagger(x)\Psi(0)\rangle_0 \sim \frac{\sin(\pi\rho_0 x)}{|x|^\eta}, \quad \eta = \frac{1}{2}\left(K + K^{-1}\right), \qquad (68)$$

with a non-universal prefactor.

Consider now the model of interacting fermions on the circle (second line in Table 1). One can predict that it corresponds to a Luttinger liquid with parameter $K = 2/\beta$. Indeed the density correlations were calculated for the C$\beta$E in Ref. [110] and one can check that formula 4.11 there agrees with the prediction of the LL theory (66), with the choice $K = 2/\beta$, up to subleading terms (which for each harmonic decreases faster by a factor $1/|x|$). As mentioned in proposition 13.2.4, p. 604 of [71] (see also [111, 112]) this asymptotics was established for $\beta$ an even integer. However the value of $K$ can be inferred already from the second term in (66), i.e., from the coefficient of the long range decay $\sim 1/x^2$, which in fact can be obtained by electrostatic arguments and linear response theory from the Coulomb gas representation [84, 113]. Note that the identification $K = 2/\beta$ was also noted in [105] (see Appendix there) and in Ref. [114] where an approximate formula for more general power-law interactions was also obtained. We thus expect that all the universal properties of the Luttinger liquid with this value of the parameter will hold for the model in second line in Table 1 for fermions on the circle. For instance the variance of the number of fermions, $\mathcal{N}_{[a,b]} = \frac{1}{\pi}(\varphi(a) - \varphi(b))$, can be computed from the correlator of the phase field given in [106] (see also [115]) as

$$\mathrm{Var}\,\mathcal{N}_{[a,b]} \simeq \frac{2}{\pi^2} \int_{-\infty}^{\infty} \frac{d\omega}{2\pi} \int_{-k_F}^{k_F} \frac{dq}{2\pi} \frac{\pi K[1 - \cos q(b-a)]}{\frac{\omega^2}{v_F} + v_F q^2} \simeq \frac{K}{\pi^2}\left[\log\left(k_F|a-b|\right) + \gamma_E\right], \quad (69)$$

where $|a - b| \gg 1/k_F$, where in the general interacting case $k_F$ is defined as $\pi\rho_0$ and is unrelated to $v_F$ (while $\hbar k_F = v_F/m$ in the noninteracting case). Substituting $K = 2/\beta$ in (69) this agrees with the leading logarithmic term in our prediction (19), although it is not accurate enough to predict the $O(1)$ term.

In the presence of both interactions and an external potential, an inhomogeneous bosonization approach was recently developed which aims to calculate correlations in the bulk using conformal field theory methods [54–56]. The potential induces a spatial dependence of the LL parameters, $K(x)$, $v_F(x)$ and $\rho_0(x)$. In the case of free fermions this approach can be compared with the exact results of Ref. [52] that we presented in Section 2.1. The correspondence appears to be, from Eq. (20) of [54] that $z(x, 0) \sim \theta(x) \sim \int^x \frac{dx}{v_F(x)}$, where $z(x, y)$ is defined there and $\theta(x)$ was defined in (25). It would be interesting to extend these results to the present models with $K = 2/\beta$. Since it is natural to assume, because of the long range nature of the interactions, that $K = 2/\beta$ is independent of $x$, a similar description will hold for more general potentials than the ones considered here.

Table 2: The mappings between (i) models of interacting trapped fermions studied in section 6 (with details in Appendix A) and (ii) random matrix ensembles. The table is in the same format as Table 1. In the first line, GWW refers to the Gross-Witten-Wadia model who studied it for $\beta = 2$ [78,79]. The acronym SW$\beta$E refers to the Stieltjes-Wigert ensemble [71].

| Fermions' domain | Fermion potential $V(x)$ | Fermion interaction $W(x,y)$ | RMT ensemble | Matrix potential $V_0(\lambda)$ | Map $\lambda(x)$ |
|---|---|---|---|---|---|
| $x \in [0, 2\pi]$ | Eq. (70) | $\frac{1}{16} \frac{\beta(\beta-2)}{\sin^2 \frac{x-y}{2}}$ | GWW | $-\frac{g(\lambda+\lambda^*)}{2}$ | $\lambda = e^{ix}$ |
| $x \in \mathbb{R}^+$ | Eq. (81) | $\frac{\beta(\beta-2)}{4}\left[\frac{1}{(x-y)^2} + \frac{1}{(x+y)^2}\right]$ | Half line | Eq. (82) | $\lambda = \frac{2}{\beta}x^2$ |
| $x \in [0, \pi]$ | Eq. (119) | $\frac{\beta(\beta-2)}{16}\left(\frac{1}{\sin^2 \frac{x-y}{2}} + \frac{1}{\sin^2 \frac{x+y}{2}}\right)$ | Box | Eq. (83) | $\lambda = \frac{1-\cos x}{2}$ |
| $x \in \mathbb{R}$ | Eq. (85) | $\frac{\beta(\beta-2)}{16\sinh^2 \frac{x-y}{2}}$ | "Hyperbolic" | Eq. (86) | $\lambda = e^x$ |
| $x \in \mathbb{R}^+$ | Eq. (136) | Eq. (137) | "Hyperbolic" on half line | Eq. (139) | $\lambda = \cosh(px)$ |
| $x \in \mathbb{R}$ | $\frac{1}{2}a^2x^2$ | $\frac{\beta(\beta-2)}{16\sinh^2 \frac{x-y}{2}} - \frac{a\beta(x-y)}{4}\coth \frac{x-y}{2}$ | SW$\beta$E | $a \log^2 \lambda$ | Eq. (87) |
| $x \in \mathbb{R}$ | $\frac{-1}{\cosh^2(x)}$ | $\frac{\beta(\beta-2)}{16}\left(\frac{1}{\sinh^2 \frac{x-y}{2}} - \frac{1}{\cosh^2 \frac{x+y}{2}}\right)$ | Cauchy | Eq. (145) | $\lambda = \sinh(px)$ |
| $x \in \mathbb{R}$ | Eq. (157) | Eq. (156) | Quartic matrix potential | $c_2\lambda^2 + c_4\lambda^4$ | $\lambda = x$ |

# 6 More general models

So far we have focused on the models in Table 1, with ground state wave functions of the form (5) involving one and two-body factors and are related to random matrix models of the form (6). There is in fact a larger class of models for which the ground state wavefunctions are still of the form (5). The study of such models was initiated by Sutherland and Calogero [68,70,116] and extended in Refs. [117–119]. We recall how models with this property are constructed using a slightly more general approach, and give a list of corresponding Hamiltonians in the Appendix A. In addition we show that some of these models are related to other interesting random matrix models. Some of them were studied in the RMT literature. The models presented in this section are summarized in Table 2.

*Fermions on the circle*. The first interesting extension corresponds to fermions on the circle, i.e $x \in [0, 2\pi]$ with periodic boundary conditions, in the presence of an external periodic potential. It generalizes the models of the second line of the Table 1 which are related to C$\beta$E (in the absence of potential). It corresponds to the Hamiltonian (4) with the external potential $V(x)$ and two-body interaction $W(x,y)$ given by

$$V(x) = \frac{g}{4}\left(1 + \frac{N-1}{2}\beta\right)\cos x - \frac{g^2}{8}\cos^2 x \,, \tag{70}$$

$$W(x,y) = \frac{1}{16}\frac{\beta(\beta-2)}{\sin^2 \frac{x-y}{2}} \,. \tag{71}$$

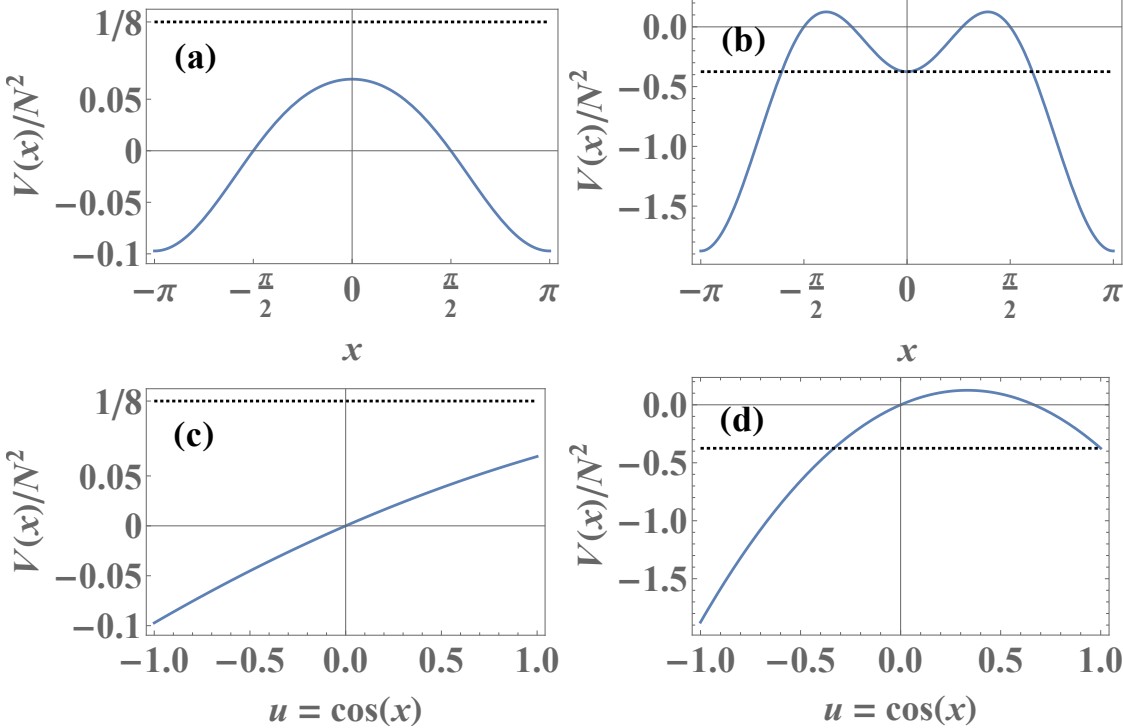

Figure 5: The potential (73) for $\tilde{g} = 1/3$ [(a) and (c)] and $\tilde{g} = 3$ [(b) and (d)]. $V$ is plotted as a function of $x$ in (a) and (b), and as a function of $u = \cos x$ in (c) and (d). As is seen in (a) and (b), for weak coupling $\tilde{g} < 1$, $V(x)$ has a single maximum, but for strong coupling $\tilde{g} > 1$ there are two degenerate maxima. The dotted lines correspond to the Fermi energy $\mu$.

The ground state wave function is of the form (5) for any $N$, with $v(x) = g \cos x$ and $w(x, y) = -\beta \log|\sin\frac{x-y}{2}|$ and the ground state energy $E_0$ is given in (116). The quantum probability is (up to a normalization)

$$|\Psi_0(\vec{x})|^2 \propto \prod_{i<j} \left|\sin\frac{x_i - x_j}{2}\right|^\beta e^{g \sum_i \cos x_i} . \tag{72}$$

Interestingly, for $\beta = 2$ this probability is identical to the one studied at large $N$ by Gross and Witten and independently by Wadia in the context of lattice gauge theories [78, 79], and later on in combinatorics [120]. The weight (72) corresponds to the joint PDF of the eigenvalues $\lambda_j = e^{ix_j}$ of a matrix model with a probability measure on the $N \times N$ unitary matrices $\propto e^{\frac{g}{2} \text{Tr}(U + U^\dagger)} dU$. Remarkably, in the present context this corresponds to noninteracting fermions. The mapping is summarized in the first line of Table 2.

As we now show, this mapping allows us to shortcut the Coulomb gas method used in Refs. [78, 79] and obtain the eigenvalue density in a simpler way. For $\beta = 2$ the external potential for the fermions reads

$$V(x) = \frac{N^2 \tilde{g}^2}{8}\left(\frac{2}{\tilde{g}} u - u^2\right), \quad u = \cos x , \tag{73}$$

where we have defined $\tilde{g} = g/N$ which is the important parameter that is kept of $O(1)$ in the large $N$ limit. In Fig. 5 we show a plot of $V(x)$ for various values of $\tilde{g}$. Since there is no interaction $W = 0$ for $\beta = 2$ in the large $N$ limit the fermion density can be obtained from the

LDA formula

$$\tilde{\rho}(x) = \frac{\sqrt{2}}{N\pi}\sqrt{(\mu - V(x))_+}\,, \tag{74}$$

where $\mu$ is the Fermi energy and is determined by the normalization condition $\int_0^{2\pi} dx\, \tilde{\rho}(x) = 1$. From (73) and Fig. 5 we see that there are two cases depending on $\tilde{g}$. For $\tilde{g} < 1$ there is a single maximum $V_{\max}$ of the potential $V(x)$ which is attained for $u = \cos x = 1$, i.e., $x = 0$, with $V_{\max} = \frac{N^2}{8}(2\tilde{g} - \tilde{g}^2)$. For $\tilde{g} > 1$ there are two degenerate maxima of the potential $V(x)$ which are attained for $u = \cos x = 1/\tilde{g}$, of values $V_{\max} = \frac{N^2}{8}$. As we now discuss this change of behavior of $V(x)$ results in two distinct phases: for $\tilde{g} < 1$ (weak coupling) the Fermi energy is above the maximum of the potential and the density is everywhere positive. For $\tilde{g} > 1$ (strong coupling) the Fermi energy is below the maximum and the density has a restricted support.

Since the Fermi energy $\mu$ is itself determined by the normalization condition this is a non trivial transition. In the weak coupling phase one finds, as we show below, that

$$\mu = \frac{N^2}{8} > V_{\max} = \frac{N^2}{8}(2\tilde{g} - \tilde{g}^2)\,, \quad \tilde{g} < 1\,. \tag{75}$$

For that particular value, one sees that the expression inside the square root in (74) becomes a perfect square leading to

$$\tilde{\rho}(x) = \frac{1}{2\pi}(1 - \tilde{g}\cos x)\,, \quad \tilde{g} < 1\,, \tag{76}$$

which is automatically normalized to unity, with a support $x \in [0, 2\pi]$. Hence (75) is the correct value for $\mu$. Since the density must be positive, this solution is acceptable only for $\tilde{g} < 1$.

For $\tilde{g} > 1$ we note that the potential $V(x)$ has a local minimum at $x = 0$ of value $V_{\min} = \frac{N^2}{8}(2\tilde{g} - \tilde{g}^2)$. In this strong coupling phase, one finds

$$\mu = \frac{1}{8}\left(2gN - g^2\right) = V_{\min}\,, \quad \tilde{g} > 1\,. \tag{77}$$

As can be seen on the Fig. 5 this value of $\mu$ is such that the system has a single support in all phases. For this value of $\mu$ we obtain from (74)

$$\tilde{\rho}(x) = \frac{\tilde{g}}{\pi}\left|\sin\left(\frac{x}{2}\right)\right|\sqrt{\left(\frac{1}{\tilde{g}} - \cos^2\left(\frac{x}{2}\right)\right)_+}\,, \quad \tilde{g} > 1\,, \tag{78}$$

which has now a restricted support $[x_-, x_+]$ where the edges are $x_- = 2\arccos\sqrt{\frac{1}{\tilde{g}}}$ and $x_+ = 2\pi - x_-$. One can check the normalization

$$\int_0^{2\pi} dx\, \tilde{\rho}(x) = \frac{2\tilde{g}}{\pi}\int_{-\sqrt{\frac{1}{\tilde{g}}}}^{\sqrt{\frac{1}{\tilde{g}}}} du\sqrt{\frac{1}{\tilde{g}} - u^2} = 1\,, \tag{79}$$

which shows that (77) is the correct value of $\mu$. For $\tilde{g} \to 1^+$ one has $x_- \simeq 2\sqrt{\tilde{g} - 1}$, and for $\tilde{g} \to +\infty$ one has $x_+ \to \pi^-$ (all fermions are around $x = \pi$). For $\tilde{g} = 1$ the formulae (76) and (78) become identical.

The phase transition in the density in the above formula recovers the results obtained in [78, 79] by a different method, upon the identification $\tilde{g} = 2/\lambda$ (and $x \to x + \pi$) from the notations of [78]. In these papers the partition function (i.e., the normalization amplitude of the probability measure in (72)) was computed and shown to exhibit a third order phase transition at $\tilde{g} = 1$ (for a recent review, see [100]). In the fermion system this transition at

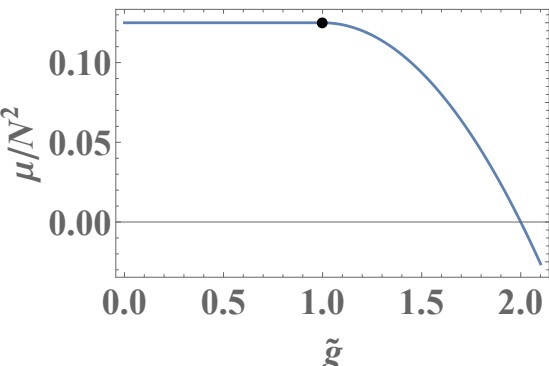

Figure 6: The fermi energy $\mu$ as a function of $\tilde{g} = g/N$, see Eq. (80). The black dot corresponds to the point $\tilde{g} = 1$ where the freezing transition occurs.

$\tilde{g} = 1$ can be seen as a freezing transition for the Fermi energy as a function of the coupling strength $\tilde{g}$ (see Figs. 5 and 6)

$$\mu = \begin{cases} \frac{N^2}{8}, & \tilde{g} < 1 \quad \text{(weak coupling)} \\ \frac{N^2}{8}\left(2\tilde{g} - \tilde{g}^2\right), & \tilde{g} > 1 \quad \text{(strong coupling)} \end{cases}. \tag{80}$$

This transition coincides with the opening of a gap in the bulk in the fermion density. A similar transition has been recently studied by us for fermions in an inverted parabolic potential [121], and the correlation kernel at the transition was explicitly computed. However, in the present situation the critical behavior is expected to be different. Indeed the shape of the potential at criticality around $x = 0$ is here $V(x) - 1 \sim -x^4$ while it is $V(x) = -x^2$ in [121].

Finally note that the model for $\beta \neq 2$, i.e., for interacting fermions in (70) is also of interest. Using CG arguments, its density $\tilde{\rho}_{\beta,\tilde{g}}(x)$ is obtained from the density for $\beta = 2$ by a simple rescaling, i.e., $\tilde{\rho}_{\beta,\tilde{g}}(x) = \tilde{\rho}_{2,\frac{2}{\beta}\tilde{g}}(x)$ found above. The transition then occurs for $\tilde{g} = \beta/2$. The correlation functions are expected, however, to depend on $\beta$ and remain to be explored.

*Fermions on the half-line.* The second interesting extension generalizes the models on the third line of Table 1 which are related to the WL$\beta$E. The interaction potential $W$ is the same as in Table 1, but the potential $V$ is more general

$$V(x) = 2c_1^2 x^6 + 2c_0 c_1 x^4 + \frac{c_2(c_2 + 2)}{8x^2} + \left(\frac{c_0^2}{2} + c_1(c_2 - 3) - \beta(N-1)2c_1\right)x^2. \tag{81}$$

The ground state wavefunction is of the form (5) with $v(x) = c_0 x^2 + c_1 x^4 + c_2 \log x$. It corresponds to a matrix model of the form (7) upon the map $\lambda(x) = \frac{2}{\beta}x^2$ with a matrix potential

$$V_0(\lambda) = c_0 \frac{\beta}{2}\lambda + c_1 \frac{\beta^2}{4}\lambda^2 + \left(\frac{c_2}{2} + 1\right)\log\lambda. \tag{82}$$

The mapping is summarized in the second line of Table 2.

*Fermions in a box.* The next extension generalizes the models on the fourth line of the Table 1 which are related to the J$\beta$E. The interaction potential $W(x, y)$ is the same as in Table 1, but the potential $V(x)$ contains additional $\cos x$ and $\cos 2x$ terms, see formula (119). The ground state wave function has the form (5) with $v(x) = c_1 \log \sin \frac{x}{2} + c_2 \log \cos \frac{x}{2} + c_3 \cos x$. It corresponds to a matrix model of the form (7) upon the map $\lambda(x) = \frac{1}{2}(1 - \cos x)$ and matrix potential

$$V_0(\lambda) = \frac{c_1 + 1}{2}\log\lambda + \frac{c_2 + 1}{2}\log(1-\lambda) - 2c_3\lambda. \tag{83}$$

The mapping for this model is summarized in the third line of Table 2. For $c_3 = 0$ it recovers the J$\beta$E. For $c_1 = -1$ and $c_3 \neq 0$, this matrix model was studied in [147–149] and its density was calculated using the Coulomb gas method. We show in Appendix B that this result agrees with the LDA.

Finally there are some models not related to Table 1.

*Hyperbolic models.* The simplest example are fermions on the real line with the two-body interaction potential

$$W(x,y) = \frac{\beta(\beta - 2)}{16 \sinh^2 \frac{x-y}{2}} \, . \tag{84}$$

Its ground state wave function is of the form (5) with a two-body term $w(x,y) = -\beta \log|\sinh \frac{1}{2}(x-y)|$. For normalizability of (5) one needs a confining potential. The most general family consistent with (84) is

$$V(x) = \frac{1}{8} c_1^2 e^{2x} + \frac{1}{8} c_2^2 e^{-2x} - \frac{\beta(N-1)}{8} (c_1 e^x + c_2 e^{-x}) + \frac{1}{4} c_1 (c_0 - 1) e^x - \frac{1}{4} c_2 (c_0 + 1) e^{-x}, \tag{85}$$

a potential of the (generalized) Morse type. The one-body term in the ground state wave function is then $v(x) = c_0 x + c_1 e^x + c_2 e^{-x}$. This model corresponds to a matrix model under the map $\lambda(x) = e^x$ with matrix potential

$$V_0(\lambda) = c_1 \lambda + c_2 \lambda^{-1} + \left[ 1 + \frac{\beta}{2}(N-1) + c_0 \right] \log \lambda \, . \tag{86}$$

The mapping for this model is summarized in the fourth line of Table 2. In the case $c_2 = 0$ of the Morse potential this relation to the Wishart model was also obtained in [122]. For $c_2 \neq 0$ the calculation of the mean density $\sigma(\lambda)$ was performed using Coulomb gas methods for some values of the parameters in [123]. We show that this result agrees with the LDA in Appendix B. Note that this matrix model was also studied in various contexts in Refs. [124, 125].

Another interesting example in this class of hyperbolic models corresponds to a ground state wave function of the form form (5) with $v(x) = ax^2$ and $w(x,y) = -\beta \log|\sinh \frac{1}{2}(x-y)|$. In that case there is an additional repulsive two-body interaction $\delta W$ on top of the interaction (84), of the form $\delta W(x,y) \propto -a\beta(x-y)\coth \frac{1}{2}(x-y)$, which never vanishes for any value of $\beta$ (the model is always interacting). This model is related to the Stieltjes-Wigert $\beta$-ensemble (SW$\beta$E) [118, 126–129] which was studied in the context of Chern-Simons theory in high energy physics [130] and of non-intersecting Brownian bridges [129, 131, 132]. The correspondence is through the map (see the discussion below Eq. (125) in Appendix A)

$$\lambda(x) = e^{x + \frac{1}{2a}\left(1 + \frac{\beta}{2}(N-1)\right)}, \tag{87}$$

and the matrix potential reads

$$V_0(\lambda) = a \log^2 \lambda = \frac{\beta}{2} \tilde{a} \log^2 \lambda \, , \tag{88}$$

where $\tilde{a} = 2a/\beta$. The mapping for this model is summarized in the sixth line of Table 2. For this matrix model the joint PDF of the eigenvalues is determinantal for $\beta = 2$, since the model becomes bi-orthogonal [133–135]. In the limit of large $N$, scaling $a = O(N)$, the eigenvalue density $\sigma(\lambda)$ is known [129, 130, 132]

$$\sigma(\lambda) = \frac{1}{\pi u \lambda} \arctan \frac{\sqrt{4e^u \lambda - (1+\lambda)^2}}{1+\lambda} \, , \tag{89}$$

where $u = N/(2\tilde{a}) = O(1)$. From the CG arguments this density is in fact independent of $\beta$. Its support is $\lambda \in [z_-, z_+]$ where $z_\pm = -z \pm \sqrt{z^2 - 1}$ and $z = 1 - 2e^u$. Hence we obtain the fermion density for the associated quantum model for any $\beta$ as $\rho(x) = N e^{u/2} e^x \sigma(e^{u/2} e^x)$.

Finally, there are two more hyperbolic models for fermions, one on the positive half axis which corresponds to the fifth line in Table 2, and the second one, which maps to the Cauchy random matrix ensemble, and corresponds to the seventh line in Table 2. These models are described in the Appendix A.

Let us close this section by indicating yet another family of quantum models where the interaction $W(x, y)$ is a sum of a harmonic attraction $\propto (x - y)^2$ and of the inverse square interaction $\frac{\beta(\beta-2)}{4(x-y)^2}$. The first case is in an external potential $V(x) \sim x^2$. The second is related to a quartic matrix model $V_0(\lambda) = c_2 \lambda^2 + c_4 \lambda^4$ and corresponds to a fermion model with a polynomial potential with terms $x^2, x^4, x^6$. These models are described in the Appendix A.

To relate to the main focus of the paper, i.e., the counting statistics, let us point out that many models presented in this Section are noninteracting for $\beta = 2$. In that case, the methods of [52] summarized in the Section 2.1 can be applied to obtain the variance of the number of fermions in an interval. Upon scaling properly the parameters of the model with $\beta$ one can relate the variance of the interacting model to the one for $\beta = 2$ by similar relations as in (18), with the same constants (17). Our conjecture for the higher cumulants should also apply.

# 7 Discussion and conclusion

In summary, we calculated the counting statistics for several models of $N \gg 1$ interacting spinless fermions in their ground state in one dimension confined by an external potential, see Tables 1 and 2. The interactions are of the general Calogero-Sutherland type, and depend on the parameter $\beta$, where $\beta = 2$ corresponds to the noninteracting case. We have emphasized the connections to random matrix ensembles, where $\beta$ is the Dyson index. We found that the variance of the number of fermions in a macroscopic interval $[a, b]$ in the bulk of the Fermi gas grows with $N$ as $A_\beta \log N + B_\beta + o(1)$. We obtained explicit formulae for $A_\beta$ and $B_\beta$, which depend on $a, b$, on the type of interaction and on the shape of the confining potential. These results were obtained by explicit calculations for $\beta \in \{1, 2, 4\}$ and from a conjecture that we formulated for general $\beta$. This conjecture extends to the higher-order cumulants of the distribution of $\mathcal{N}_{[a,b]}$. They are $O(1)$ at $N \gg 1$ and are predicted here to be given by (51). Remarkably, this result is universal: it does not depend on the confining potential. This is because the conjecture states that the short scales determine the $O(1)$ part of the fluctuations of the particle number. We have obtained a few "smoking gun" tests for this conjecture. First we have shown that it matches in a highly nontrivial way, near the edge of the Fermi gas, with recent results from the mathematics literature [77]. Second we have shown that our analytical predictions are in very good agreement with our numerical simulations. In addition we have shown that the leading term $A_\beta$ is in agreement with the predictions from the Luttinger liquid theory with parameter $K = 2/\beta$.

Finally, we formulated a general approach for obtaining mappings between interacting fermion models in one dimension in their ground state and random matrix models (or, more generally, models of classical interacting particles confined by an external potential in thermal equilibrium). We applied this approach and found several such mappings. In particular we found a surprising mapping of the famous RMT Gross-Witten-Wadia model from high energy physics onto noninteracting fermions in an external potential on a circle. The simple application of the LDA allows to obtain the mean fermion density in that case, and recovers results known for this model obtained by more involved Coulomb gas methods. In turn, we have shown that these Coulomb gas methods can be used to study interacting fermions in a trapping potential. We exploited these mappings to obtain the mean fermion density for these models for general interaction parameter $\beta$ by relating them to the noninteracting case $\beta = 2$. Similarly, we argue that the counting statistics in these models can be calculated by relating

them to the noninteracting case.

Our results hold also for Dyson indices $0 < \beta < 1$ which, although meaningless in the fermion systems on which we focused here (since for fermions $\beta \geq 1$), are meaningful for the RMT ensembles. The scaling limit $\beta \sim 1/N$ has generated much interest recently, and it would be interesting to study the counting statistics in this limit [136–140].

Among the connections unveiled in this paper, e.g., with the models in Table 2, many interesting questions remain to be explored. In particular one may wonder whether the universality of the higher cumulants of the fermion number can be extended to more general interacting models, and whether one can derive formula for the variance in more general potentials. In particular, it would be interesting to test this universality when perturbing the interaction term away from the Calogero-Sutherland type studied here.

For noninteracting fermions, the counting statistics is connected to the bipartite entanglement entropy (EE) of the subsystem $\mathcal{D}$ with its complement $\overline{\mathcal{D}}$ [12–15]. Given the results of the present work, it would be interesting to search for similar (perhaps approximate) connections for interacting fermions in order to calculate the EE.

Finally, it would be interesting to extend our approach to higher dimensions. In particular, there is a known mapping between noninteracting fermions in a 2d rotating harmonic trap and random matrices of the complex Ginibre ensemble [16,17]. It remains a challenge to extend this mapping to more general cases.

## Acknowledgements

PLD thanks Y. V. Fyodorov for an earlier collaboration on related topics. We thank A. Borodin and P. J. Forrester for useful correspondence. We thank D. S. Dean and C. Salomon for interesting discussions. We thank T. Bothner for useful comments on the manuscript. We thank M. Beau for pointing out the recent references [141, 142] about ground-states in Calogero-type models and their extensions in higher dimensions. NRS acknowledges support from the Yad Hanadiv fund (Rothschild fellowship). This research was supported by ANR grant ANR-17-CE30-0027-01 RaMaTraF.

## A   Interacting fermion models with ground state of the form (5) and mappings to RMT

In this Appendix we recall the construction of quantum Hamiltonians with two-body interactions in one dimension (4), whose ground state wave function has itself a two-body form as in Eq. (5). This question was pionneered by Calogero [70] (following Sutherland [68,116]) and extended in Refs. [117–119, 141, 142]. In some cases these models are also fully integrable (i.e., their full eigenspectrum is known), see e.g. [69, 71, 143, 144]. Here we also discuss the construction of the ground state in the light of the connections to random matrix ensembles. In particular we perform a search for models using the map $\lambda(x)$ which relates RMT to fermions.

### A.1   Schrödinger equation and general conditions for two-body-only interaction

Consider the following unnormalized wave function, $\Psi_0(\vec{x}) = e^{-U(\vec{x})/2}$ (defined up to a sign in an ordered sector), where $U(\vec{x}) = \sum_i v(x_i) + \sum_{i<j} w(x_i, x_j)$ has the two-body form (5). A necessary condition for it to be the ground state of the two-body Hamiltonian $\mathcal{H}_N$ in (4) with energy $E_0$ is that $\mathcal{H}_N \Psi_0(\vec{x}) = E_0 \Psi_0(\vec{x})$. Substituting and multiplying by $e^{U(\vec{x})/2}$ on both sides

one gets

$$\sum_i V(x_i) + \sum_{i<j} W\left(x_i, x_j\right) - E_0 = e^{U(\vec{x})/2} \frac{1}{2} \sum_i \partial_{x_i}^2 e^{-U(\vec{x})/2} = -\frac{1}{4} \sum_i U_{ii}'' + \frac{1}{8} \sum_i (U_i')^2 \quad (90)$$

$$= -\frac{1}{4}\left[\sum_i v''(x_i) + \sum_{i\neq j} w_{20}(x_i, x_j)\right] + \frac{1}{8}\sum_i \left[v'(x_i) + \sum_{j\neq i} w_{10}(x_i, x_j)\right]^2$$

$$= T_1 + T_2 + T_2' + T_3\,, \quad (91)$$

where $T_n$ denotes a term which is naively $n$ body. Here, and in the following, we use the notation $U_i' = \partial_{x_i} U(\vec{x})$ and similarly $U_{ii}'' = \partial_{x_i}^2 U(\vec{x})$. We recall that $w(x, y)$ is a symmetric function and denote by subscripts the order of its partial derivatives. These terms are

$$T_1 = \sum_i V^{(1)}(x_i)\,, \quad V^{(1)}(x) = \frac{1}{8} v'(x)^2 - \frac{1}{4} v''(x)\,, \quad (92)$$

$$T_2 = \sum_{i<j} W^{(1)}(x_i, x_j)\,, \quad W^{(1)}(x, y) = W^{(1,1)}(x, y) + W^{(1,2)}(x, y)\,, \quad (93)$$

$$W^{(1,1)}(x, y) = -\frac{1}{4}\left[w_{20}(x, y) + w_{02}(x, y)\right]\,, \quad (94)$$

$$W^{(1,2)}(x, y) = \frac{1}{8}\left[w_{10}(x, y)^2 + w_{10}(y, x)^2\right]\,, \quad (95)$$

$$T_2' = \sum_{i<j} W^{(2)}(x_i, x_j)\,, \quad W^{(2)}(x, y) = \frac{1}{8}(v'(x)w_{10}(x, y) + v'(y)w_{10}(y, x))\,, \quad (96)$$

$$T_3 = \frac{1}{8} \sum_{j\neq i, k\neq i, j\neq k} w_{10}(x_i, x_j)w_{10}(x_i, x_k)$$

$$= \frac{1}{8} \sum_{i<j<k} \sum_{\tau\in S_3} w_{10}(x_{\tau(i)}, x_{\tau(j)})w_{10}(x_{\tau(i)}, x_{\tau(k)})\,, \quad (97)$$

where we have splitted the term $\frac{1}{8}\sum_i \sum_{j\neq i} w_{10}(x_i, x_j)\sum_{k\neq i} w_{10}(x_i, x_k)$ into the term $j = k$ (in $T_2$) and $j \neq k$ (in $T_3$). In the cases that we will study, these terms will drastically simplify and turn out to be constants (and sometimes zero). The ground state energy $E_0$ will be determined, as a result.

To obtain a two-body Hamiltonian we must thus impose the condition that the three-body interactions are absent. This amounts to a condition on $w(x, y)$ so that $T_3$ can be written as two-body term, or a one-body or a constant. We will search for solutions to this condition in two possible forms $w(x, y) = w(x - y)$ and $w(x, y) = -\beta \log|\lambda(x) - \lambda(y)|$. Asking that $T_3$ is a constant, or one-body, then allows for a systematic search. This leads to a set of quantum model with two-body interactions $W(x, y) = (W^{(1)} + W^{(2)})|_{2\text{body}}$, with a specific family of interactions $T_2|_{2\text{body}}$ which vanish for $\beta = 2$, while $T_2'|_{2\text{body}}$ depends on $v(x)$. For some specific choices of $v(x)$ which we identify $T_2'|_{2\text{body}} = 0$, which lead to simpler quantum models in an external potential which become noninteracting for $\beta = 2$.

In the next section we make the list of the models which are obtained by this method, and in the following section we explain how one searches for these models.

Remark: The term $T_1$ has the form of potentials from supersymmetric quantum mechanics. More generally the above equation (90) is equivalent to

$$H - E_0 = \frac{1}{2}\sum_i \left(-\partial_{x_i} + \frac{U_i'}{2}\right)\left(\partial_{x_i} + \frac{U_i'}{2}\right)\,. \quad (98)$$

For repulsive interactions $\beta > 2$, the mappings described here are expected to hold for bosons too. For bosons, it is the repulsive interaction that causes the many-body wave function $\Psi_0$ to vanish at $x_i = x_j$ for $i \neq j$.

## A.2 Families of models

We consider here different kinds of models. Some are defined on the real line (or the half-line) and require a confining potential $v(x)$ in order for $\Psi_0$ to be normalized. The others are called "periodic" models, and defined either on the circle or an interval, in which case $v(x)$ may be chosen to be zero. We recall that when there is a mapping $x \mapsto \lambda(x)$ between the fermions models with potential $v(x)$ and a matrix model (6) with a matrix potential $V_0(\lambda)$, the relation between the two potentials reads

$$v(x) = V_0(\lambda(x)) - \log|\lambda'(x)| . \tag{99}$$

Note that $w$ and $v$ in (5) are defined up to an irrelevant additive constant which can be absorbed into the normalisation of $\Psi_0$. Depending on $v(x)$, one may also extract a one-body part from $W(x,y)$ and add it to $V(x)$, and extract constant parts from $W, V$ and add them to $-E_0$, where $E_0$ below denotes the ground state energy.

**Logarithmic models**. In this class, the first set of models is, for $x, y$ on the real axis

$$
\begin{aligned}
w(x,y) &= -\beta \log|x-y|, \quad \lambda(x) = x, \quad T_3 = 0, \tag{100} \\
W(x,y) &= \frac{\beta(\beta-2)}{4(x-y)^2} - \frac{\beta}{4}\frac{v'(x)-v'(y)}{x-y}, \quad V(x) = \frac{1}{8}v'(x)^2 - \frac{1}{4}v''(x), \quad E_0 = 0 . \tag{101}
\end{aligned}
$$

In this set of models the only normalizable choice of $v(x)$ which does not contribute to the two-body interaction $W$ is $v(x) = ax^2$. It corresponds to the quantum model[8]

$$V(x) = \frac{a^2}{2}x^2, \quad W(x,y) = \frac{\beta(\beta-2)}{4(x-y)^2}, \quad E_0 = \frac{\beta a}{4}N(N-1) + \frac{Na}{2}. \tag{102}$$

This corresponds to the **G$\beta$E**, for which the canonical choice, given in the text, is $a = 1$, $\lambda(x) = \sqrt{\frac{2}{\beta}}x$ and $V_0(\lambda) = \beta\lambda^2/2$. For the noninteracting case $\beta = 2$ (setting $a = 1$) one recovers that $E_0$ is the sum of the energies of the single-particle states up to the Fermi energy, $E_0 = \sum_{n=0}^{N-1}(n + \frac{1}{2})$.

The second set of models is, for $x, y$ on the positive real axis[9]

$$
\begin{aligned}
w(x,y) &= -\beta \log|x^2-y^2|, \quad \lambda(x) = x^2, \quad T_3 = 0, \tag{103} \\
W(x,y) &= \frac{\beta(\beta-2)}{4}\left(\frac{1}{(x-y)^2} + \frac{1}{(x+y)^2}\right) - \frac{\beta}{2}\frac{xv'(x)-yv'(y)}{x^2-y^2}, \\
V(x) &= \frac{1}{8}v'(x)^2 - \frac{1}{4}v''(x), \quad E_0 = 0 .
\end{aligned}
$$

In this set of models the only normalizable choice of $v(x)$ which does not contribute to the

---

[8]In Eqs. (102), (104), (105), (106), (114)-(116), (119)-(121), (128), (130), (136)-(138), (142)-(144), (154)-(158) constant terms from $W(x,y)$ and $V(x)$ were absorbed into $-E_0$, and/or one-body terms from $W(x,y)$ were absorbed into $V(x)$.

[9]It is possible to define the model on the real axis, however $\Psi_0(\vec{x})$, although still an eigenfunction, is not the ground state, since it also vanishes at points $x_i = -x_j$ (in addition to $x_i = x_j$).

two-body interaction is $v(x) = c_0 x^2 + c_1 x^4 + c_2 \log x$ which corresponds to the quantum model

$$V(x) = 2c_1^2 x^6 + 2c_0 c_1 x^4 + \left(\frac{c_0^2}{2} + c_1 (c_2 - 3) - \beta(N-1)2c_1\right) x^2 + \frac{c_2 (c_2 + 2)}{8x^2}, \quad (104)$$

$$W(x,y) = \frac{\beta(\beta - 2)}{4} \left(\frac{1}{(x-y)^2} + \frac{1}{(x+y)^2}\right), \quad (105)$$

$$E_0 = \beta c_0 \frac{N(N-1)}{2} + \frac{1}{2} c_0 (1 - c_2)N. \quad (106)$$

This contains the case of the **WL$\beta$E** with the canonical choice, given in the text

$$c_0 = 1, \quad c_1 = 0, \quad c_2 = -(1 + 2\gamma), \quad \lambda(x) = \frac{2}{\beta} x^2, \quad V_0(\lambda) = \frac{\beta}{2}\lambda - \gamma \log \lambda, \quad (107)$$

which leads to

$$V(x) = \frac{x^2}{2} + \frac{\gamma^2 - \frac{1}{4}}{2x^2}, \quad W(x,y) = \frac{\beta(\beta-2)}{4}\left(\frac{1}{(x-y)^2} + \frac{1}{(x+y)^2}\right), \quad (108)$$

$$E_0 = \frac{\beta}{2} N(N-1) + (\gamma + 1)N. \quad (109)$$

For $\beta = 2$ one recovers the energy $E_0 = \sum_{n=0}^{N-1}(2n + 1 + \gamma)$. However there is a larger class of potentials which correspond to matrix models with matrix potentials:

$$V_0(\lambda) = c_0 \frac{\beta}{2}\lambda + c_1 \frac{\beta^2}{4}\lambda^2 + \frac{c_2 + 1}{2}\log\lambda.$$

**Periodic models.** In this class, the first set of models is defined on the circle with $px \in [0, 2\pi[$

$$w(x,y) = -\beta \log|\sin\frac{p}{2}(x-y)|, \quad T_3 = -\frac{1}{8}\frac{N(N-1)(N-2)}{3}\beta^2\frac{p^2}{4}, \quad (110)$$

$$W(x,y) = \frac{\beta(\beta-2)p^2}{16\sin^2\frac{p}{2}(x-y)} - \frac{\beta p}{8}(v'(x) - v'(y))\cot\frac{p}{2}(x-y), \quad (111)$$

$$V(x) = \frac{1}{8}v'(x)^2 - \frac{1}{4}v''(x), \quad (112)$$

$$E_0 = \frac{\beta^2 p^2}{32}\left(N(N-1) + \frac{N(N-1)(N-2)}{3}\right) = \frac{\beta^2 p^2}{32}\frac{N(N-1)(N+1)}{3}. \quad (113)$$

It contains the **C$\beta$E** which is obtained for $v(x) = 0$. The canonical choice given in the text is $p = 1$. One can check that for $\beta = 2$, the ground state energy is exactly equal to the sum of the energies of the single-particle states, e.g. for $N$ odd one has $E_0 = 2\sum_{k=0}^{\frac{N-1}{2}} k^2 = \frac{N(N-1)(N+1)}{24}$.

In this set the only choice of $v(x)$ which does not generate a two-body interaction is $v(x) = b\cos(px)$ (up to translations on the circle), which leads to the quantum model on the circle

$$V(x) = b\frac{p^2}{4}\left(1 + \frac{N-1}{2}\beta\right)\cos(px) - \frac{1}{8}b^2 p^2 \cos^2(px), \quad (114)$$

$$W(x,y) = \frac{p^2}{16}\frac{\beta(\beta-2)}{\sin^2\frac{p}{2}(x-y)}, \quad (115)$$

$$E_0 = \frac{\beta^2 p^2}{16}\frac{N(N-1)}{2} + \frac{1}{8}\frac{N(N-1)(N-2)}{3}\beta^2\frac{p^2}{4} - N\frac{b^2 p^2}{8}. \quad (116)$$

For $\beta = 2$ this is the Gross-Witten-Wadia model discussed in the text.

The second set of models is defined for $px \in [0, \pi]$ and corresponds to the choice $w(x, y) = -\beta \log |\cos px - \cos py|$, which is equivalent to the choice

$$w(x, y) = -\beta \log \left| \sin \frac{p}{2}(x - y) \right| \left| \sin \frac{p}{2}(x + y) \right| , \quad \lambda(x) = \frac{1}{2}(1 - \cos(px)) = \sin^2 \frac{px}{2} ,$$

$$T_3 = -\frac{1}{8} \frac{N(N-1)(N-2)}{3} \beta^2 p^2 ,$$

$$W(x, y) = \frac{\beta(\beta - 2)p^2}{16} \left( \frac{1}{\sin^2 \frac{p(x-y)}{2}} + \frac{1}{\sin^2 \frac{p(x+y)}{2}} \right) + \frac{\beta p \left( \sin(px)v'(x) - \sin(py)v'(y) \right)}{4(\cos(px) - \cos(py))} ,$$

$$\tag{117}$$

$$E_0 = \frac{\beta^2 p^2}{8} \frac{N(N-1)}{2} + \frac{1}{8} \frac{N(N-1)(N-2)}{3} \beta^2 p^2 . \tag{118}$$

In this set the only choice of $v(x)$ which does not contribute to the two-body interaction is $v(x) = c_1 \log \sin \frac{px}{2} + c_2 \log \cos \frac{px}{2} + c_3 \cos px$, which leads to the quantum model on the circle

$$V(x) = \frac{c_3 p^2}{8}(2 - c_1 - c_2 + 2\beta(N-1))\cos(px) - \frac{c_3^2 p^2}{16}\cos(2px)$$

$$+ \frac{p^2 c_1(2 + c_1)}{32 \sin^2 \frac{px}{2}} + \frac{p^2 c_2(2 + c_2)}{32 \cos^2 \frac{px}{2}} , \tag{119}$$

$$W(x, y) = \frac{\beta(\beta - 2)p^2}{16} \left( \frac{1}{\sin^2 \frac{p(x-y)}{2}} + \frac{1}{\sin^2 \frac{p(x+y)}{2}} \right) , \tag{120}$$

$$E_0 = \frac{\beta^2 p^2}{8} \frac{N(N-1)}{2}(1 - c_1 - c_2) + \frac{1}{8} \frac{N(N-1)(N-2)}{3} \beta^2 p^2$$

$$+ \frac{p^2 N}{32} \left[ (c_1 + c_2)^2 + 2c_3(2c_1 - 2c_2 - c_3) \right] . \tag{121}$$

In this set of models, choosing $c_3 = 0$, $c_1 = -(2\gamma_1 + 1)$, $c_2 = -(2\gamma_2 + 1)$ we obtain the Jacobi box potential which corresponds to the **J$\beta$E**. Let us set $p = 1$, i.e., $L = \pi$ for the box, and define the map $\lambda(x) = \frac{1}{2}(1 - \cos x) = \sin^2 \frac{x}{2}$ and $1 - \lambda(x) = \frac{1}{2}(1 + \cos x) = \cos^2 \frac{x}{2}$. The matrix potential becomes $V_0(\lambda) = -\gamma_1 \log \lambda - \gamma_2 \log(1 - \lambda)$ hence (for $0 < x < \pi$)

$$v(x) = V_0(\lambda(x)) - \log |\lambda'(x)| = -\left( \gamma_1 + \frac{1}{2} \right) \log \sin^2 \frac{x}{2} - \left( \gamma_2 + \frac{1}{2} \right) \log \cos^2 \frac{x}{2} . \tag{122}$$

In summary we have

$$V(x) = \frac{1}{8} \left( \frac{\gamma_1^2 - \frac{1}{4}}{\sin^2 \frac{x}{2}} + \frac{\gamma_2^2 - \frac{1}{4}}{\cos^2 \frac{x}{2}} \right), \quad W(x, y) = \frac{\beta(\beta - 2)}{16} \left( \frac{1}{\sin^2 \frac{x-y}{2}} + \frac{1}{\sin^2 \frac{x+y}{2}} \right), \tag{123}$$

$$E_0 = \frac{(\gamma_1 + \gamma_2 + 1)^2 N}{8} + \frac{\beta^2 N(N-1)}{16} + \frac{\beta N(N-1)(\gamma_1 + \gamma_2 + 1)}{8}$$

$$+ \frac{\beta^2 N(N-1)(N-2)}{24} . \tag{124}$$

For $\beta = 2$ using the single-particle energy levels $\epsilon_n = \frac{1}{2}\left( n + \frac{\gamma_1 + \gamma_2 + 1}{2} \right)^2$ one finds that $E_0 = \sum_{n=0}^{N-1} \epsilon_n = \frac{N\left[ 6N(\gamma_1 + \gamma_2) + 3(\gamma_1 + \gamma_2)^2 + 4N^2 - 1 \right]}{24}$ which coincides with the formula (124) spe-

cialised to $\beta = 2$.

**Hyperbolic models**. In this class, the first set of models is defined on the real axis

$$w(x,y) = -\beta \log \left| \sinh \frac{p}{2}(x-y) \right|, \quad T_3 = \frac{1}{8} \frac{N(N-1)(N-2)}{3} \beta^2 \frac{p^2}{4}, \tag{125}$$

$$W(x,y) = \frac{\beta(\beta-2)p^2}{16\sinh^2 \frac{p}{2}(x-y)} - \frac{\beta p}{8}(v'(x)-v'(y))\coth \frac{p}{2}(x-y),$$

$$V(x) = \frac{1}{8}v'(x)^2 - \frac{1}{4}v''(x), \tag{126}$$

$$E_0 = -\frac{\beta^2 p^2}{32}\left[N(N-1) + \frac{N(N-1)(N-2)}{3}\right] = -\frac{\beta^2 p^2}{32}\frac{N(N-1)(N+1)}{3}. \tag{127}$$

Note that this model $(w, v)$ is equivalent to the model $(\tilde{w}, \tilde{v})$ where $\tilde{w}(x,y) = -\beta \log|\lambda(x) - \lambda(y)|$ with $\lambda(x) = e^{px}$ and $\tilde{v}(x) = v(x) + \frac{\beta p}{2}(N-1)x$. It is thus equivalent to a matrix model with the matrix potential $V_0(\lambda)$ such that $\tilde{v}(x) = V_0(e^{px}) - px$.

For the choice $v(x) = ax^2$ this model is related to the Stieltjes-Wigert $\beta$ ensemble [118, 129] as discussed in the text, where we have also used the parametrization $(\tilde{w}, \tilde{v})$ to obtain Eq. (87).

In this set the only choice of $v(x)$ which does not contribute to the two-body interaction is $v(x) = c_0 x + c_1 e^{px} + c_2 e^{-px}$. This leads to the quantum model

$$V(x) = \frac{1}{8}c_1^2 p^2 e^{2px} + \frac{1}{8}c_2^2 p^2 e^{-2px} + \frac{1}{4}c_1 p(c_0-p)e^{px} - \frac{1}{4}c_2 p(c_0+p)e^{-px} \tag{128}$$

$$-\frac{\beta p^2}{8}(N-1)(c_1 e^{px} + c_2 e^{-px}),$$

$$W(x,y) = \frac{\beta(\beta-2)p^2}{16\sinh^2 \frac{p}{2}(x-y)}, \tag{129}$$

$$E_0 = \frac{N}{8}\left(2c_1 c_2 p^2 - c_0^2\right) - \frac{\beta^2 p^2}{32}\frac{N(N-1)(N+1)}{3}, \tag{130}$$

which has a potential of the (generalized) Morse type. Note that the ground state energy behaves as $\propto -N^3$, which is surprising for a repulsive interaction (say $\beta > 2$). This is because the confining potential (necessary for normalization) has a deep minimum at energy $\propto -N^2$. Note that it corresponds to a matrix model with matrix potential

$$V_0(\lambda) = c_1 \lambda + c_2 \lambda^{-1} + \left(1 + \frac{\beta}{2}(N-1) + \frac{c_0}{p}\right)\log \lambda, \quad \lambda \geq 0. \tag{131}$$

The second set of models is defined on the positive half line and corresponds to the choice $w(x,y) = -\beta \log|\cosh px - \cosh py|$, which is equivalent to the choice

$$w(x,y) = -\beta \log \left| \sinh \frac{p}{2}(x-y) \right| \left| \sinh \frac{p}{2}(x+y) \right|, \quad \lambda(x) = \cosh(px), \tag{132}$$

$$T_3 = \frac{1}{8}\frac{N(N-1)(N-2)}{3}\beta^2 p^2, \tag{133}$$

$$W(x,y) = \frac{\beta(\beta-2)p^2}{16}\left(\frac{1}{\sinh^2 \frac{p(x-y)}{2}} + \frac{1}{\sinh^2 \frac{p(x+y)}{2}}\right)$$

$$- \frac{\beta p \left(\sinh(px)v'(x) - \sinh(py)v'(y)\right)}{4(\cosh(px)-\cosh(py))}, \tag{134}$$

$$E_0 = -\frac{\beta^2 p^2}{8}\frac{N(N-1)}{2} - \frac{1}{8}\frac{N(N-1)(N-2)}{3}\beta^2 p^2. \tag{135}$$

In this set the only choice of $v(x)$ which does not contribute to a two-body interaction is $v(x) = c_1 \log \sinh \frac{px}{2} + c_2 \log \cosh \frac{px}{2} + c_3 \cosh px$, which leads to the quantum model

$$V(x) = \frac{c_3 p^2}{8} \left[ -2 + c_1 + c_2 - 2\beta(N-1) \right] \cosh(px) + \frac{c_3^2 p^2}{16} \cosh(2px)$$
$$+ \frac{p^2 c_1 (2 + c_1)}{32 \sinh^2 \frac{px}{2}} - \frac{p^2 c_2 (2 + c_2)}{32 \cosh^2 \frac{px}{2}}, \tag{136}$$

$$W(x, y) = \frac{\beta(\beta - 2)p^2}{16} \left( \frac{1}{\sinh^2 \frac{p(x-y)}{2}} + \frac{1}{\sinh^2 \frac{p(x+y)}{2}} \right), \tag{137}$$

$$E_0 = -\frac{\beta^2 p^2 N(N-1)}{16} - \frac{N(N-1)(N-2)}{24} \beta^2 p^2 + \frac{\beta p^2}{8} \frac{N(N-1)}{2} (c_1 + c_2)$$
$$- \frac{p^2 N}{32} \left[ (c_1 + c_2)^2 + 2c_3 (2c_1 - 2c_2 - c_3) \right]. \tag{138}$$

It corresponds to a matrix model with matrix potential

$$V_0(\lambda) = \frac{c_1 + 1}{2} \log(\lambda - 1) + \frac{c_2 + 1}{2} \log(\lambda + 1) + c_3 \lambda, \quad \lambda \in [1, +\infty). \tag{139}$$

The mapping for this model is summarized in the fifth line of Table 2.

Finally there is a third set of models, defined on the line or on the positive half-line and which corresponds to the choice $w(x, y) = -\beta \log |\sinh px - \sinh py|$, which is equivalent to the choice

$$w(x, y) = -\beta \log \left| \sinh \frac{p}{2}(x - y) \right| \left| \cosh \frac{p}{2}(x + y) \right|, \quad \lambda(x) = \sinh(px),$$

$$T_3 = \frac{1}{8} \frac{N(N-1)(N-2)}{3} \beta^2 p^2,$$

$$W(x, y) = \frac{\beta(\beta - 2)p^2}{16} \left( \frac{1}{\sinh^2 \frac{p(x-y)}{2}} - \frac{1}{\cosh^2 \frac{p(x+y)}{2}} \right)$$
$$+ \frac{\beta p \left( \cosh(py)v'(y) - \cosh(px)v'(x) \right)}{4(\sinh(px) - \sinh(py))}, \tag{140}$$

$$E_0 = -\frac{\beta^2 p^2}{8} \frac{N(N-1)}{2} - \frac{1}{8} \frac{N(N-1)(N-2)}{3} \beta^2 p^2. \tag{141}$$

In this set the only choice of $v(x)$ which does not contribute to the two-body interaction is $v(x) = c_1 \arctan \sinh px + c_2 \log \cosh px + c_3 \sinh px$ which leads to the quantum model

$$V(x) = \left( c_1^2 - c_2 (c_2 + 2) \right) \frac{p^2}{8 \cosh^2(px)} + \frac{p^2}{8} c_3^2 \cosh^2(px) + \frac{1}{4} c_1 (c_2 + 1) p^2 \frac{\sinh px}{\cosh^2 px}$$
$$+ \frac{p^2}{4} c_3 (c_2 - 1 - \beta(N-1)) \sinh(px), \tag{142}$$

$$W(x, y) = \frac{\beta(\beta - 2)p^2}{16} \left( \frac{1}{\sinh^2 \frac{p(x-y)}{2}} - \frac{1}{\cosh^2 \frac{p(x+y)}{2}} \right), \tag{143}$$

$$E_0 = -\frac{\beta^2 p^2}{8} \frac{N(N-1)}{2} - \frac{N(N-1)(N-2)}{24} \beta^2 p^2$$
$$+ \beta p^2 \frac{N(N-1)}{8} c_2 - \frac{N}{8} \left( c_2^2 + 2c_1 c_3 \right) p^2. \tag{144}$$

Note however that for normalizability on the whole axis one needs $c_3 = 0$. For $c_3 = 0$ the potential $V(x)$ is known as the hyperbolic Scarf potential [151]. It corresponds to a matrix

model

$$V_0(\lambda) = c_3\lambda + c_1\arctan(\lambda) + \frac{c_2+1}{2}\log(1+\lambda^2)\,. \tag{145}$$

The mapping for this model is summarized in the seventh line of Table 2. In the case $c_3 = 0$ this model is the generalized Cauchy beta ensemble (Ca$\beta$E) studied, e.g., in [152, 153]. The joint PDF of eigenvalues has the form

$$P(\vec{\lambda}) \propto \prod_j \frac{1}{(1+i\lambda_j)^{a+ib}} \frac{1}{(1-i\lambda_j)^{a-ib}} \times \prod_{i<j} |\lambda_i - \lambda_j|^\beta\,, \tag{146}$$

with $a = \frac{1+c_2}{2}$ and $b = -c_1/2$. In the case $c_1 = 0$ (i.e., $b = 0$) and $a = \frac{\beta}{2}(N-1)+1$, the Ca$\beta$E is related to the circular ensemble C$\beta$E via the stereographic projection $e^{i\theta} = \frac{1+i\lambda}{1-i\lambda}$. As a result the exact density $\sigma_N(\lambda)$ is known for any finite $N$ from the fact that it is uniform on the circle [71, 153] and it is given by

$$\sigma_N(\lambda) = \frac{1}{\pi}\frac{1}{1+\lambda^2}\,, \tag{147}$$

independently of $N$ and $\beta$. This mapping does not relate however the Schrödinger operators on the circle and the line, so it does not extend to the quantum model. In fact the potential in the fermion model associated to the case $c_1 = c_3 = 0$ is the Pöschl-Teller potential, as can be seen from (142) (setting $p = 1$)

$$V(x) = -\frac{c_2(c_2+2)}{8\cosh^2 x}\,. \tag{148}$$

For $\beta = 2$ (noninteracting fermions) the LDA approximation at large $N$ for the fermion density, is $\rho(x) = \frac{1}{\pi}\sqrt{2\mu + \frac{c_2(c_2+2)}{4\cosh^2 x}}$. It is easy to check that this formula is compatible with the exact result , which holds for $c_2 \simeq 2N$ at large $N$. This determines the value of the Fermi energy as $\frac{\mu}{N^2} \simeq 0$, which means that in the ground state, the potential well which has only a finite number of energy levels, is almost full.

**Elliptic models**. In all the above models $T_3$ was a constant. There is a more general family of models for which $T_3$ is a sum of one-body terms. They are such that $w(x,y) = -\beta\log|\lambda(x)-\lambda(y)|$ and $\lambda(x)$ solution of:

$$\lambda''(x) = B + A\lambda(x) + \frac{C}{2}\lambda(x)^2\,. \tag{149}$$

In that case $T_3 = \frac{1}{4}\sum_{i<j<k}(u(x_i)+u(x_j)+u(x_k))$ with $u(x) = \frac{\beta^2}{3}\frac{\lambda'''(x)}{\lambda'(x)}$, which leads to a potential $V(x) = \frac{1}{8}(N-1)(N-2)u(x)$. For $C = 0$ one recovers the models discussed above (and $u(x)$ is then a constant). For the general case $C \neq 0$ the solutions of (149) are of the form $\lambda(x) = a + b\,\mathcal{P}(x; g_2, g_3)$ where $a = -A/C$, $b = 12/C$ and $g_2 = -\frac{C}{6}(B - \frac{A^2}{2C})$ and $g_3$ is an arbitrary constant. Here $\mathcal{P}$ is the Weierstrass function[10] (see also next Section). This leads to $V(x) = \frac{\beta^2}{2}(N-1)(N-2)\mathcal{P}(x; g_2, g_3)$. The two-body term $T_2$ then leads to the interaction $W^{(1)}(x,y) = \frac{\beta}{8}\left[(\beta-2)\frac{\lambda'(x)^2+\lambda'(y)^2}{(\lambda(x)-\lambda(y))^2} + 2\frac{\lambda''(x)-\lambda''(y)}{\lambda(x)-\lambda(y)}\right]$ which appears to be quite complicated. We will not further study these solutions here.

---

[10]See e.g., https://dlmf.nist.gov/23.2.

**Models with quadratic interactions**. A last example is the following fermion model defined on the real axis

$$w(x, y) = a(x - y)^2 - \beta \log|x - y|, \tag{150}$$

$$W(x, y) = \left[a^2 + \frac{a^2}{2}(N - 2)\right](x - y)^2 + \frac{\beta(\beta - 2)}{4(x - y)^2}$$
$$+ [v'(x) - v'(y)]\left[\frac{a}{2}(x - y) - \frac{\beta}{4(x - y)}\right], \tag{151}$$

$$V(x) = \frac{1}{8}v'(x)^2 - \frac{1}{4}v''(x), \tag{152}$$

$$E_0 = a(1 + \beta)\frac{N(N - 1)}{2} + \frac{1}{4}a\beta N(N - 1)(N - 2). \tag{153}$$

For general $a$, one has that $T_3$ is the sum of two-body terms. There are two interesting special cases. The first one is the case where $v(x) = c_2 x^2$ which leads to

$$W(x, y) = \left[a^2 + \frac{a^2}{2}(N - 2) + a c_2\right](x - y)^2 + \frac{\beta(\beta - 2)}{4(x - y)^2}, \quad V(x) = \frac{1}{2}c_2^2 x^2, \tag{154}$$

$$E_0 = \frac{N c_2}{2} + \left[a(1 + \beta) + \frac{1}{2}\beta c_2\right]\frac{N(N - 1)}{2} + \frac{1}{4}a\beta N(N - 1)(N - 2). \tag{155}$$

Another special case is $a = 0$, for which $T_3$ is actually a constant (this model also belongs to the first class in Eq. (100)). In this case, one has $w(x, y) = -\beta \log|x - y|$ and $v(x) = c_2 x^2 + c_4 x^4$ which leads to

$$W(x, y) = \frac{\beta c_4}{2}(x - y)^2 + \frac{\beta(\beta - 2)}{4(x - y)^2}, \tag{156}$$

$$V(x) = 2c_4^2 x^6 + 2c_2 c_4 x^4 + \left(\frac{1}{2}c_2^2 - 3c_4 - \beta c_4 \frac{3}{2}(N - 1)\right)x^2, \tag{157}$$

$$E_0 = \frac{\beta c_2}{2}\frac{N(N - 1)}{2} + \frac{c_2}{2}N. \tag{158}$$

This quantum model is thus related via $\lambda(x) = x \in \mathbb{R}$, to the random matrix model with a quartic matrix potential

$$V_0(\lambda) = c_2 \lambda^2 + c_4 \lambda^4. \tag{159}$$

This model is well known in RMT [145, 146, 154, 155]. The mapping for this model is summarized in the eighth line of Table 2.

## A.3 Search for models

Let us summarize how one searches for models. One imposes the condition that the three-body interactions are absent, i.e., that $T_3$ can be written as a two-body term, or a one-body term or a constant. We first consider the case when $T_3$ in Eq. (97) is simply a constant. It means that for all $x, y, z$

$$t_3(x, y, z) := w_{10}(x, y)w_{10}(x, z) + w_{10}(y, x)w_{10}(y, z) + w_{10}(z, x)w_{10}(z, y) = \pm q^2, \tag{160}$$

in which case $T_3 = \frac{1}{4}\sum_{i < j < k}(\pm q^2) = \pm \frac{1}{8}\frac{N(N-1)(N-2)}{3}q^2$. Note that this does not involve $v(x)$ hence for now $v(x)$ is arbitrary.

A first series of model is obtained by considering $w(x, y) = w(x - y)$ with $w$ even. Inserting into (160) and taking $z, y \to x$ one sees that there is no differentiable solution at $x = 0$, i.e.,

with $w'(0) = 0$ since $w'(x)$ is an odd function. Hence one needs $w'(x)$ to diverge at $x = 0$. One thus writes $w'(z) = 1/g(z)$ with $g(z)$ an odd function. Setting $z = x + \epsilon$, one must have, regrouping the terms

$$\frac{1}{g(\epsilon)}\left[\frac{1}{g(x+\epsilon-y)} - \frac{1}{g(x-y)}\right] + \frac{1}{g(y-x)g(y-x-\epsilon)} = \pm q^2 . \qquad (161)$$

We see that the only possibility (for a smooth $g(x)$ at generic non zero $x$) is $g(\epsilon) = O(\epsilon)$, i.e., a simple pole for $w'(x)$, and taking $\epsilon \to 0$ one finds

$$\frac{1}{g(x)^2}\left[1 - \frac{g'(x)}{g'(0)}\right] = \pm q^2 . \qquad (162)$$

The solutions are $g(x) = \frac{1}{q}\tan(qxg'(0))$, $g(x) = \frac{1}{q}\tanh(qxg'(0))$, and $g(x) = g'(0)x$. Defining $g'(0) = -1/\beta$ and $q = -\frac{\beta}{2}p$ one obtains $w(x,y) = -\beta\log|\sin(\frac{p}{2}(x-y))|$ for the $(+)$ branch, $w(x,y) = -\beta\log|\sinh(\frac{p}{2}(x-y))|$ for the $(-)$ branch, and $w(x,y) = -\beta\log|x-y|$ for $q = 0$. Here $\beta$ and $p$ are now an arbitrary parameters. One checks that indeed they satisfy the condition (160) (providing not so trivial trigonometric identities). These are the models respectively (110), (125) and (100). For these models the two-body term from $T_2$ leads to the interaction potential $W^{(1)}(x,y) = \frac{1}{4}w'(x-y)^2 - \frac{1}{2}w''(x-y)$. In the presence of a potential $v(x)$ there is generically another two-body term from $T_2'$ in Eq. (96). It reads $W^{(2)}(x,y) = \frac{1}{4}(v'(x) - v'(y))w'(x-y)$ which leads to the interaction potentials in (110), (125) and (100). For each of these three models there is a unique family of exceptional potentials $v(x)$ for which $W^{(2)}(x,y)$ is either one-body or constant. To search for them one imposes the necessary condition $\partial_x\partial_y W^{(2)}(x,y) = 0$ and solves it for $y \to x$. This leads to the models (102), (114) and (128).

Interestingly, the models found so far are closely related, up to some change of variable $\lambda = \lambda(x)$, to the logarithmic interaction which appears naturally in the $\beta$ random matrix models of the form (6), with an a priori arbitrary matrix potential $V_0(\lambda)$. Thus it is natural to search for $w(x,y)$ parameterized in the form $w(x,y) = -\beta\log|\lambda(x) - \lambda(y)|$, where again, for now, $v(x)$ is arbitrary. Note that there is some redundancy, since e.g. the problem with $\lambda(x) = 1/\tilde{\lambda}(x)$ is equivalent to $w(x,y) = -\beta\log|\tilde{\lambda}(x) - \tilde{\lambda}(y)|$ and $v(x) \to v(x) + (N-1)\beta\log|\tilde{\lambda}(x)|$. One now imposes to satisfy (160) with $w_{10}(x,y) = -\beta\frac{\lambda'(x)}{\lambda(x)-\lambda(y)}$ and we restrict here to $\lambda(x)$ real. Substituting and taking successively the limits $z \to x$ and $y \to x$, one arrives at the necessary condition $\beta^2\frac{\lambda'''(x)}{\lambda'(x)} = \pm q^2$. It is convenient to parameterize these models using the parameter $p$ defined now via[11] $q^2 = \beta^2 p^2$. The general solutions are $\lambda(x) = ax + bx^2$ (for $q = 0$), $\lambda(x) = a_1\cos px + a_2\sin px$ (for the $-$ branch) and $\lambda(x) = a_1'\cosh px + a_2'\sinh px$ for the (for the $+$ branch). The case $\lambda(x) = x$ recovers (100), while $\lambda(x) = x^2$ gives the new family (103) (and upon a translation in $x$ one can always reduce to one of these cases). Similarly for the periodic model one can always choose $a_2 = 0$ by translation, which leads to (117). Finally for the hyperbolic solutions one can always reduce to either $a_1' = a_2'$, which leads again to (125), or to $a_2' = 0$ (whenever $a_1' > a_2$), which leads to (134), or to $a_1' = 0$ (whenever $a_1' < a_2$), which leads to (140). One can check that $T_3$ is indeed a constant for each of these models, again via non trivial trigonometric identities. With the chosen parameterization $w(x,y) = -\beta\log|\lambda(x) - \lambda(y)|$ the term $T_2$ leads to an interaction $W^{(1)}(x,y) = \frac{\beta}{8}((\beta-2)\frac{\lambda'(x)^2+\lambda'(y)^2}{(\lambda(x)-\lambda(y))^2} + 2\frac{\lambda''(x)-\lambda''(y)}{\lambda(x)-\lambda(y)})$, which simplifies for these models as given in (100), (103), (117), (125),(134), (140).

---

[11]This allows to state that e.g. the model (125) is equivalent to the model with $\lambda(x) = e^{px}$. Although there is then a relative factor of 4 in their respective values for $T_3$, this is compensated by the change in $T_2'$ from the change $v \to \tilde{v}$ see discussion below (125).

In each of these sets we can search for exceptional potentials $v(x)$ for which $W^{(2)}(x,y)$ in Eq. (96) is either one-body or constant, i.e.,

$$W^{(2)}(x,y) = -\frac{\beta}{4}\frac{v'(x)\lambda'(x) - v'(y)\lambda'(y)}{\lambda(x) - \lambda(y)} = -\frac{\beta}{4}(u(x) + u(y)). \tag{163}$$

This is equivalent to

$$v'(x)\lambda'(x) - v'(y)\lambda'(y) = (u(x) + u(y))(\lambda(x) - \lambda(y)). \tag{164}$$

This is possible only if the cross term on the right hand side, namely $\lambda(x)u(y) - \lambda(y)u(x)$, is a one-body term, meaning that $\partial_x\partial_y(\lambda(x)u(y) - \lambda(y)u(x)) = 0$. This implies that $u'(y)/\lambda'(y) = u'(x)/\lambda'(x)$, which implies: $u'(y)/\lambda'(y) = u'(x)/\lambda'(x) = K_1$ i.e., $u(x) = K_1\lambda(x) + K_2$. Inserting into (164) it gives the necessary condition

$$v'(x) = \frac{K_1\lambda(x)^2 + K_2\lambda(x) + K_3}{\lambda'(x)}, \quad u(x) = K_1\lambda(x) + K_2. \tag{165}$$

Using the specific forms for $\lambda(x)$ obtained above (i.e., for which $T_3$ is a constant), this relation (165) leads to the models (104), (119), (128), (136) and (142) which are the most general solutions in each case.

Next we turn to the condition that the three-body term in (160) is the sum of one-body terms, i.e., $t_3(x,y,z) = u(x) + u(y) + u(z)$. Substituting and taking successively the limits $z \to x$ and $y \to x$ one arrives at the necessary condition $\beta^2\frac{\lambda'''(x)}{\lambda'(x)} = 3u(x)$. Next we expand in $y - x$ and to second order we obtain another necessary condition

$$-3\lambda^{(3)}(x)\lambda''(x)^2 + \lambda'(x)\left(\lambda^{(3)}(x)^2 - \lambda^{(5)}(x)\lambda'(x)\right) + 3\lambda^{(4)}(x)\lambda'(x)\lambda''(x) = 0. \tag{166}$$

Multiplying this equation by $\lambda'(x)^{-4}$ it can be integrated once. Multiplying the result by $\lambda'(x)$ it can be rewritten as $\frac{d}{dx}\left[\frac{\lambda^{(3)}(x)}{\lambda'(x)}\right] = C\lambda'(x)$. Multiplying by $\lambda'(x)$ and integrating once more leads to the simple condition $\lambda''(x) = B + A\lambda(x) + \frac{C}{2}\lambda(x)^2$, i.e., the relation given above in Eq. (149), where $A, B$ and $C$ are integration constants.

We have not explored in full generality the condition that $T_3$ is a two-body term, i.e., $t_3(x,y,z)$ defined in (160) be a sum of two and one-body potentials. This condition leads to the nonlinear, nonlocal partial differential equation

$$\partial_x\partial_y\partial_z t_3(x,y,z) = w_{21}(x,y)w_{11}(x,z) + w_{11}(x,y)w_{21}(x,z) + w_{21}(y,x)w_{11}(y,z) \tag{167}$$
$$+ w_{11}(y,x)w_{21}(y,z) + w_{21}(z,x)w_{11}(z,y) + w_{11}(z,x)w_{21}(z,y) = 0,$$

whose study we leave for future research. In the case of the form $w(x,y) = w(x - y)$ this was done by Calogero [70] who found that the general solution must obey $w''(z) = a\,\mathcal{P}(z, g_2, g_3)$ where $\mathcal{P}$ is the Weierstrass $\mathcal{P}$ function[12] i.e., a solution of $\mathcal{P}'' = 6\mathcal{P}^2 - \frac{g_2}{2}$ and $(\mathcal{P}')^2 = 4\mathcal{P}^3 - g_2\mathcal{P} - g_3$. Integrating twice the general solution is thus in that case, $w(z) = -\beta\log(\sigma(z; g_2, g_3)) + bz^2$ where $\sigma(z; g_2, g_3)$ is the $\sigma$-Weierstrass function. Thanks to non trivial identities involving Weierstrass functions [70], the absence of three-body term is indeed obeyed. Note that it is natural to set $a = \beta$ since $\sigma \simeq z$ at small $z$, hence $w(z)$ is again of the logarithmic type at small $z$. The resulting quantum interaction $W(x,y)$ can be expressed as a polynomial in terms of Weierstrass functions [70].

Here we only give the simple example (150). In fact, the only solution with $w(x,y) = w(x - y)$ and $w''(0)$ finite is the quadratic form $w(x,y) = a(x - y)^2$.

---

[12]See e.g., https://dlmf.nist.gov/23.2.

## B  Mean fermion density: LDA versus Coulomb gas

In the absence of interactions, $\beta = 2$, the LDA prediction for the mean fermion density in the large $N$ limit is (with unit normalization)

$$\tilde{\rho}(x) = \frac{\sqrt{2}}{N\pi} \sqrt{\mu - V(x)}\,. \tag{168}$$

On the other hand, when there is a map $\lambda(x)$ which maps the ground state of this model to the joint PDF of the eigenvalues of a RMT ensemble of the form

$$P(\vec{\lambda}) \propto e^{-\frac{\beta}{2}\sum_i N\tilde{V}_0(\lambda_i) - \beta\sum_{i<j}\log|\lambda_i - \lambda_j|}\,, \tag{169}$$

i.e., when the matrix potential in (6) is scaled as $V_0(\lambda) = \frac{\beta}{2}N\tilde{V}_0(\lambda)$, then it is possible to use the CG method to obtain the eigenvalue density $\sigma(\lambda)$. It is given as the optimal density which minimizes the CG energy functional

$$\mathcal{E}[\sigma] = \int d\lambda\, \tilde{V}_0(\lambda)\sigma(\lambda) - \int d\lambda d\lambda'\, \sigma(\lambda)\sigma(\lambda')\log|\lambda - \lambda'|\,, \tag{170}$$

under the constraint that $\int d\lambda\, \sigma(\lambda) = 1$, which we (abusively) also denote $\sigma(\lambda)$. The connection between the two is $\tilde{\rho}(x) = \lambda'(x)\sigma(\lambda(x))$. Since the CG density is independent of $\beta$ this allows to obtain the fermion density for any $\beta$.

We have discussed this connection in the text on the example of the Gross-Witten-Wadia model. Here we give some more details for the other cases. The computationally difficult part is to determine the Fermi energy $\mu$.

**Hyperbolic model**. Consider the model (84) discussed in the text with potential $V(x)$ given in (85), which has three parameters $c_0, c_1, c_2$. We will scale them as $c_j = N\frac{\beta}{2}\tilde{c}_j$. From (86), it corresponds, in the large $N$ limit, to the matrix potential (dropping subleading terms at large $N$) $\tilde{V}_0(\lambda) = \tilde{c}_1\lambda + \tilde{c}_2\lambda^{-1} + (1 + \tilde{c}_0)\log\lambda$. In [123] the minimization equation was solved in the case $\tilde{c}_1 = 1$, $1 + \tilde{c}_0 = -1$ and it was found that

$$\sigma(\lambda) = \frac{1}{2\pi}\frac{\lambda + c}{\lambda^2}\sqrt{(\lambda - a)(b - \lambda)}\,. \tag{171}$$

In [123] the parameters $a, b, c$ are given as a function of $\mu_1 = \tilde{c}_2$. On the other hand the LDA prediction for $\beta = 2$ with $\lambda(x) = e^x$ gives in the more general case of the potential (85)

$$\begin{aligned}
\sigma(\lambda) &= \frac{dx}{d\lambda}\frac{\sqrt{2}}{N\pi}\sqrt{\mu - V(x(\lambda))} \\
&= \frac{\sqrt{2}}{\pi}\frac{1}{\lambda}\sqrt{\tilde{\mu} - \left(\frac{\tilde{c}_1^2}{8}\lambda^2 + \frac{\tilde{c}_2^2}{8}\frac{1}{\lambda^2} + \frac{\tilde{c}_1}{4}(\tilde{c}_0 - 1)\lambda - \frac{\tilde{c}_2}{4}(\tilde{c}_0 + 1)\frac{1}{\lambda}\right)}\,,
\end{aligned} \tag{172}$$

with $\mu = N^2\tilde{\mu}$. Let us recover (171) from this result. Plugging $\tilde{c}_1 = 1$, $\tilde{c}_0 = -2$ into (172) yields

$$\sigma(\lambda) = \frac{1}{2\pi}\frac{1}{\lambda^2}\sqrt{-\lambda^4 + 6\lambda^3 + 8\tilde{\mu}\lambda^2 - 2\tilde{c}_2\lambda - \tilde{c}_2^2}\,. \tag{173}$$

Now, requiring that the expression under the square root in (173) can be written in the form

$$-\lambda^4 + 6\lambda^3 + 8\tilde{\mu}\lambda^2 - 2\tilde{c}_2\lambda - \tilde{c}_2^2 = (\lambda + c)^2(\lambda - a)(b - \lambda)\,, \tag{174}$$

for some $a, b, c$, one obtains by comparing the coefficients of powers of $\lambda$ on both sides of the equation, the following relations

$$6 = a + b - 2c, \quad 8\tilde{\mu} = -ab + 2ac + 2bc - c^2, \quad -2\tilde{c}_2 = -2abc + ac^2 + bc^2, \quad \tilde{c}_2^2 = abc^2, \quad (175)$$

whose solution, in terms of $v = \sqrt{ab}$ and $u = \sqrt{a/b}$, is

$$v = 2u\frac{3u^2 - 2u + 3}{(1 - u^2)^2}, \quad \tilde{c}_2 = 2vu\frac{v - 1}{u^2 + 1} = -4u^2\frac{\left(u^2 - 6u + 1\right)\left(3u^2 - 2u + 3\right)}{(1 - u^2)^4},$$

$$c = \frac{\tilde{c}_2}{v} = -2u\frac{u^2 - 6u + 1}{(1 - u^2)^2}, \quad (176)$$

in agreement with [123], and the (rescaled) fermi energy

$$\tilde{\mu} = \frac{1}{8}\left(-v^2 + 2\tilde{c}_2\frac{u^2 + 1}{u} - \frac{\tilde{c}_2^2}{v^2}\right), \quad (177)$$

which can be expressed in terms of $\tilde{c}_2$ alone using the equations in (176). In particular, the agreement with [123] ensures that $\sigma(\lambda)$ is correctly normalized, which shows that the form (174) is indeed correct.

**Fermions in a box.** This is the case of the matrix potential (83). Here $\lambda(x) = \frac{1}{2}(1 - \cos x)$. The LDA prediction is given by (169) where $V(x)$ is given in (119) (with $\beta = 2$) and $\mu$ is determined through the normalization $\int \tilde{\rho}(x)dx = 1$. In general, this expression for $\tilde{\rho}$ and the calculation of $\mu$ are very cumbersome. However, choosing $p = 1$ and $\beta = 2$ in (119) and assuming $c_1 = O(1)$, $c_2 = O(1)$, $c_3 = O(N)$, the potential is simply

$$V(x) \simeq \frac{c_3 N}{2}\cos x - \frac{c_3^2}{8}\cos^2 x + \frac{c_3^2}{16}, \quad (178)$$

to leading order for large $N$. We note that up to the additive constant $\frac{c_3^2}{16}$ this potential coincides (at $N \gg 1$) with the potential (73) with $2N$ particles, if one identifies $\tilde{g} \leftrightarrow \frac{c_3}{2N}$. Therefore, the density predicted by the LDA can be immediately deduced from Eqs. (75)-(78) with $2N$ particles, by adding a factors of 2 to $\tilde{\rho}$ due to the difference between the domains $[-\pi, \pi]$ for Gross-Witten vs. $[0, \pi]$ in the present case. The result is:

$$\tilde{\rho}(x) = \begin{cases} \frac{1}{\pi}\left(1 - \frac{c_3}{2N}\cos x\right), & 0 < \frac{c_3}{2N} < 1, \\ \frac{c_3}{N\pi}\left|\sin\left(\frac{x}{2}\right)\right|\sqrt{\left(\frac{2N}{c_3} - \cos^2\left(\frac{x}{2}\right)\right)_+}, & \frac{c_3}{2N} > 1, \end{cases} \quad (179)$$

and the associated fermi energies are

$$\mu = \begin{cases} \frac{8N^2 + c_3^2}{16}, & 0 < \frac{c_3}{2N} < 1, \\ \frac{c_3 N}{2} - \frac{c_3^2}{16}, & \frac{c_3}{2N} > 1. \end{cases} \quad (180)$$

It is assumed above that $c_3 > 0$, but flipping the sign of $c_3$ is equivalent to transforming $x \to \pi - x$.

This can be compared with the predictions of [147–149] who studied the model (6) with matrix potential (83) with (in our notations)

$$c_1 = 1 - \beta, \quad c_2 = 0, \quad c_3 = -\frac{\beta}{4}pN. \quad (181)$$

They obtained the density

$$
\sigma_{p=-\frac{4c_3}{\beta N}}(\lambda) =
\begin{cases}
\frac{p}{2\pi\sqrt{\lambda(1-\lambda)}}\left(\frac{4+p}{2p}-\lambda\right), & 0 \le \lambda \le 1, \quad -4 \le p \le 4, \\
\frac{p}{2\pi\sqrt{\lambda}}\sqrt{\frac{4}{p}-\lambda}, & 0 \le \lambda \le 4/p, \quad p \ge 4, \\
\frac{|p|}{2\pi\sqrt{1-\lambda}}\sqrt{\lambda-(1-4/|p|)}, & 1-4/|p| \le \lambda \le 1, \quad p \le -4,
\end{cases}
\tag{182}
$$

which, for $\beta = 2$, leads to

$$
\lambda'(x)\sigma(\lambda(x)) =
\begin{cases}
\frac{1}{\pi}\left(1-\frac{c_3}{2N}\cos x\right), & 0 \le \lambda \le 1, \quad -4 \le \frac{2c_3}{N} \le 4, \\
-\cos\frac{x}{2}\frac{c_3}{\pi N}\sqrt{-\frac{2N}{c_3}-\sin^2\frac{x}{2}}, & 0 \le \lambda \le 4/\left|\frac{4c_3}{\beta N}\right|, \quad -\frac{2c_3}{N} \ge 4, \\
\frac{c_3}{\pi N}\left|\sin\frac{x}{2}\right|\sqrt{\frac{2N}{c_3}-\cos^2\frac{x}{2}}, & 1-4/\frac{4c_3}{\beta N} \le \lambda \le 1, \quad -\frac{2c_3}{N} \le -4,
\end{cases}
\tag{183}
$$

in agreement with (179).

## C  Number variance for the harmonic trap

Here we calculate the number variance for interacting fermions described by the model (3), which corresponds to random matrices in the G$\beta$E, thereby obtaining Eqs. (40) and (14) of the main text. Let us first consider $\mathrm{Var}\left(\mathcal{N}_{[a,\infty)}\right)$. We aim to calculate the double integral (31) in the large-$N$ limit, by approximating $C(x,y) \simeq 0$ if either $x$ or $y$ are not in the bulk, and for $x$ and $y$ both in the bulk, plugging in $C(x,y)$ from (32) for $x$ near $y$, and (38) for $x$ far from $y$ (this procedure works because there is a joint regime of validity for both of these approximate expressions for $C(x,y)$). Thus $\mathrm{Var}\left(\mathcal{N}_{[a,\infty)}\right) \simeq I_1 + I_2$ where

$$
I_1 \equiv \left(\int_{a+\xi}^{\sqrt{\beta N}} dx \int_{-\sqrt{\beta N}}^{a} dy + \int_{a}^{a+\xi} dx \int_{-\sqrt{\beta N}}^{a-\xi} dy\right) \frac{1-\frac{xy}{\beta N}}{\beta\pi^2(x-y)^2\left(1-\frac{x^2}{\beta N}\right)^{1/2}\left(1-\frac{y^2}{\beta N}\right)^{1/2}},
\tag{184}
$$

$$
I_2 \equiv [N\rho_N(a)]^2 \int_{a}^{a+\xi} dx \int_{a-\xi}^{a} dy \, Y_{2\beta}\left(N\rho_N(a)|x-y|\right),
\tag{185}
$$

where we have chosen some cutoff $\xi$ such that $\frac{1}{\sqrt{N}} \ll \xi \ll 1$, which justifies the approximation $\rho_N(x) \simeq \rho_N(a)$ that we made in the integral $I_2$.

We now calculate (184). Rescaling $\tilde{x} = x/\sqrt{\beta N}$, $\tilde{y} = y/\sqrt{\beta N}$, this term can be written as

$$
I_1 = \frac{1}{\beta\pi^2} g\left(\frac{a}{\sqrt{\beta N}}, \frac{\xi}{\sqrt{\beta N}}\right),
\tag{186}
$$

where

$$
g(\tilde{a}, z) = \left(\int_{\tilde{a}+z}^{1} d\tilde{x} \int_{-1}^{\tilde{a}} d\tilde{y} + \int_{\tilde{a}}^{\tilde{a}+z} d\tilde{x} \int_{-1}^{\tilde{a}-z} d\tilde{y}\right) \tilde{C}(\tilde{x}, \tilde{y}),
\tag{187}
$$

$$
\tilde{C}(\tilde{x}, \tilde{y}) = \frac{1-\tilde{x}\tilde{y}}{(\tilde{x}-\tilde{y})^2(1-\tilde{x}^2)^{1/2}(1-\tilde{y}^2)^{1/2}}.
\tag{188}
$$

Now using $-\frac{1}{2}\partial_{\tilde{x}}\partial_{\tilde{y}}\sigma(\tilde{x}, \tilde{y}) = \tilde{C}(\tilde{x}, \tilde{y})$ where

$$
\sigma(\tilde{x}, \tilde{y}) = -2\log\left(\frac{|\tilde{x}-\tilde{y}|}{1-\tilde{x}\tilde{y}+\sqrt{1-\tilde{x}^2}\sqrt{1-\tilde{y}^2}}\right),
\tag{189}
$$

the integral (187) is then given in terms of $\sigma$ by

$$g(\tilde{a}, z) = \frac{1}{2}\left[\sigma(\tilde{a}+z, \tilde{a}) - \sigma(\tilde{a}+z, \tilde{a}-z) + \sigma(\tilde{a}, \tilde{a}-z)\right], \tag{190}$$

where we used $\sigma(1, \cdots) = \sigma(\cdots, -1) = 0$. In the limit $z \ll 1$ this becomes

$$g(\tilde{a}, z \ll 1) = \log\frac{4\left(1 - \tilde{a}^2\right)}{z} + o(1), \tag{191}$$

leading to

$$I_1 \simeq \frac{1}{\beta \pi^2}\left[\log 4 + \log\frac{\sqrt{\beta N}\left(1 - \tilde{a}^2\right)}{\xi}\right]. \tag{192}$$

We now turn to the integral (185), and we focus on $\beta \in \{1, 2, 4\}$. After changing integration variables $\tilde{x} = N\rho_N(a)(x-a)$, $\tilde{y} = N\rho_N(a)(y-a)$, it becomes

$$I_2 = f_\beta(N\rho_N(a)\xi), \qquad f_\beta(z) = \int_0^z d\tilde{x}\int_{-z}^0 d\tilde{y}\, Y_{2\beta}(\tilde{x}-\tilde{y}). \tag{193}$$

It is useful to note that $\eta_\beta''(z) = Y_{2\beta}(z)$ where

$$\eta_2(z) = \frac{\mathrm{Ci}(2\pi z) + 2\pi z\,\mathrm{Si}(2\pi z) - \log(2\pi z) + \cos(2\pi z)}{2\pi^2}, \tag{194}$$

$$\eta_1(z) = \frac{4\eta_2(z) + \mathrm{Is}(z) - \mathrm{Is}(z)^2}{2}, \tag{195}$$

$$\eta_4(z) = \frac{4\eta_2(2z) - \mathrm{Is}(2z)^2}{8}. \tag{196}$$

(For $\beta = 1$, note that the argument of $Y_{2\beta}$ in the integral (193) is always positive, and then we use $Y_{21}(r) = (s(r))^2 - \mathrm{Is}(r)\mathrm{Ds}(r) + \frac{1}{2}\mathrm{Ds}(r)$ for $r > 0$.) As a result,

$$-\partial_{\tilde{x}}\partial_{\tilde{y}}\eta_\beta(\tilde{x}-\tilde{y}) = Y_{2\beta}(\tilde{x}-\tilde{y}), \tag{197}$$

which leads to

$$\begin{aligned}
f_\beta(z) &= \int_0^z d\tilde{x}\int_{-z}^0 d\tilde{y}\, Y_{2\beta}(\tilde{x}-\tilde{y})\\
&= -\eta_\beta(\tilde{x}-\tilde{y})\big|_{(z,0)} + \eta_\beta(\tilde{x}-\tilde{y})\big|_{(0,0)} - \eta_\beta(\tilde{x}-\tilde{y})\big|_{(0,-z)} + \eta_\beta(\tilde{x}-\tilde{y})\big|_{(z,-z)}\\
&= \eta_\beta(2z) - 2\eta_\beta(z) + \eta_\beta(0).
\end{aligned} \tag{198}$$

For the purpose of our calculation, since $\xi \gg 1/\sqrt{N}$, we need the $z \gg 1$ behavior of $f_\beta(z)$. Using

$$\eta_2(0) = \frac{1+\gamma_E}{2\pi^2}, \qquad \eta_2(z \gg 1) \simeq \frac{\pi^2 z - \log(2\pi z)}{2\pi^2}, \tag{199}$$

$$\eta_1(0) = \frac{1+\gamma_E}{\pi^2}, \qquad \eta_1(z \gg 1) \simeq \frac{\pi^2 z - \log(2\pi z) + \frac{\pi^2}{8}}{\pi^2}, \tag{200}$$

$$\eta_4(0) = \frac{1+\gamma_E}{4\pi^2}, \qquad \eta_4(z \gg 1) \simeq \frac{2\pi^2 z - \log(4\pi z) - \frac{\pi^2}{8}}{4\pi^2}, \tag{201}$$

we find

$$f_\beta(z \gg 1) \simeq \frac{\log(\pi z) + c_\beta - \log 2}{\beta\pi^2}, \tag{202}$$

for $\beta \in \{1, 2, 4\}$, where the constants $c_\beta$ are given in (15) and (16). Finally, by using $\text{Var}\left(\mathcal{N}_{[a,\infty)}\right) \simeq I_1 + I_2$ together with Eqs. (192), (193) and (202) and plugging in the density (13), we obtain Eq. (40) of the main text.

For a finite interval $[a, b]$ it is convenient to use $\mathcal{N}_{[a,b]} = N - \mathcal{N}_{(-\infty,a]} - \mathcal{N}_{[b,\infty)}$, which together with the linearity of the covariance, yields

$$\text{Var}\left(\mathcal{N}_{[a,b]}\right) = \text{Var}\left(\mathcal{N}_{(-\infty,a]}\right) + \text{Var}\left(\mathcal{N}_{[b,\infty)}\right) + 2\text{Cov}\left(\mathcal{N}_{(-\infty,a]}, \mathcal{N}_{[b,\infty)}\right). \tag{203}$$

The covariance is calculated using (30) and then approximating $C(x, y) \simeq 0$ if $x$ or $y$ are not in the bulk, and (38) if $x$ and $y$ are both in the bulk (this approximation holds in the entire domain of integration below since we are assuming that $a$ and $b$ are well separated in the bulk)

$$
\begin{aligned}
\text{Cov}\left(\mathcal{N}_{(-\infty,a]}, \mathcal{N}_{[b,\infty)}\right) &= \int_{-\infty}^{a} dx \int_{b}^{\infty} dy\, C(x, y) \\
&\simeq -\int_{-\sqrt{\beta N}}^{a} dx \int_{b}^{\sqrt{\beta N}} dy\, \frac{1 - \frac{xy}{\beta N}}{\beta \pi^2 (x - y)^2 \left(1 - \frac{x^2}{\beta N}\right)^{1/2} \left(1 - \frac{y^2}{\beta N}\right)^{1/2}} \\
&= -\frac{1}{\beta \pi^2} \int_{-1}^{\tilde{a}} d\tilde{x} \int_{\tilde{b}}^{1} d\tilde{y}\, \tilde{C}(\tilde{x}, \tilde{y}) = -\frac{1}{2\beta \pi^2} \sigma\left(\tilde{a}, \tilde{b}\right),
\end{aligned}
\tag{204}
$$

where we rescaled $\tilde{x} = x/\sqrt{\beta N}$, $\tilde{y} = y/\sqrt{\beta N}$. Finally, plugging (40), (204) and (189) into (203), we obtain Eq. (14) of the main text.

# D Number variance for the WL$\beta$E and the J$\beta$E

In this Appendix we present a detailed derivation of Eq. (48) and we give the result for the variance of the fermion number for the models in the third and fourth line of the Table 1, which are not already given in the text.

In [52] we found the number variance for the WL$\beta$E with $\beta = 2$ for a semi-infinite interval:

$$2\pi^2 \text{Var}\mathcal{N}_{[0,a]}^{\text{LUE}} = \log(\mu) + \log\left(4\frac{a}{\sqrt{2\mu}} \frac{\left(1 - \frac{a^2}{2\mu} - \frac{\lambda^2 \mu}{2a^2}\right)^{3/2}}{(1 - \lambda^2)^{1/2}}\right) + c_2 + o(1), \tag{205}$$

where $\mu = 2N + \gamma + 1$ and $\lambda^2 = \frac{\gamma^2 - \frac{1}{4}}{\mu^2}$. Expressing this result in terms of $N$ (rather than $\mu$), we obtain to leading order for large $N$ (by replacing $\mu \to 2N + \gamma$, $\lambda^2 \to \frac{\gamma^2}{(2N+\gamma)^2}$)

$$2\pi^2 \text{Var}\mathcal{N}_{[0,a]}^{\text{LUE}} = \log\left(4\sqrt{2}\sqrt{2+\tilde{\gamma}}\tilde{a}N \frac{\left(1 - \frac{2\tilde{a}^2}{(2+\tilde{\gamma})} - \frac{\tilde{\gamma}^2}{8\tilde{a}^2(2+\tilde{\gamma})}\right)^{3/2}}{\left(1 - \frac{\tilde{\gamma}^2}{(2+\tilde{\gamma})^2}\right)^{1/2}}\right) + c_2 + o(1), \tag{206}$$

where $\tilde{\gamma} = \gamma/N$ and $\tilde{a} = a/\sqrt{4N}$. Finally, we use (206) in the conjecture $\beta \pi^2 \text{Var}\mathcal{N}_{[0,a]}^{(\beta)} - c_\beta = 2\pi^2 \text{Var}\mathcal{N}_{[0,a]}^{(\beta=2)} - c_2 + o(1)$ and this leads to Eq. (48).

For the WL$\beta$E and a general interval in the bulk, a calculation similar to that which leads to (48) yields the number variance (based on our result for $\beta = 2$ in [52], together with our

conjecture (18))

$$\frac{\beta\pi^2}{2}\mathrm{Var}\mathcal{N}_{[a,b]}^{(\beta)} = \log\left(8N\sqrt{1+\frac{\tilde{\gamma}}{2}}\sqrt{\tilde{a}\tilde{b}\kappa_{\tilde{a}}^3\kappa_{\tilde{b}}^3}\frac{|\tilde{a}^2-\tilde{b}^2|}{\tilde{a}^2+\tilde{b}^2-4\frac{\tilde{a}^2\tilde{b}^2}{2+\tilde{\gamma}}-\frac{\tilde{\gamma}^2}{4(2+\tilde{\gamma})}+2\tilde{a}\tilde{b}\kappa_{\tilde{a}}\kappa_{\tilde{b}}}\right)+c_\beta+o(1),$$

(207)

where

$$\tilde{a}=\frac{a}{\sqrt{2\beta N}},\quad \tilde{\gamma}=\frac{2\gamma}{N\beta},\quad \kappa_{\tilde{a}}=\left(1-2\frac{\tilde{a}^2}{2+\tilde{\gamma}}-\frac{\tilde{\gamma}^2}{8(2+\tilde{\gamma})\tilde{a}^2}\right)^{1/2},$$

(208)

and $\tilde{b}$ and $\kappa_{\tilde{b}}$ defined similarly.

For the fermions in the hard box potential

$$V(x)=\begin{cases}0 & x\in[0,\pi]\\ \infty & x\notin[0,\pi]\end{cases},$$

(209)

which can be obtained as the limit of the J$\beta$E for $\gamma_1=\gamma_2=1/2$, we obtained the number variance for semi-infinite and finite intervals in [52] for the case $\beta=2$. Using these results together with the conjecture (18) and its analog $\beta\pi^2\mathrm{Var}\mathcal{N}_{[0,a]}^{(\beta)}-c_\beta=2\pi^2\mathrm{Var}\mathcal{N}_{[0,a]}^{(\beta=2)}-c_2+o(1)$ for semi-infinite intervals, we find

$$\mathrm{Var}\mathcal{N}_{[0,a]} = \frac{1}{\beta\pi^2}\left(\log N+\log|\sin a|+\log 2+c_\beta+o(1)\right),$$

(210)

$$\mathrm{Var}\mathcal{N}_{[a,b]} = \mathrm{Var}\mathcal{N}_{[0,a]}+\mathrm{Var}\mathcal{N}_{[0,b]}+\frac{2}{\beta\pi^2}\log\left|\frac{\sin\frac{a-b}{2}}{\sin\frac{a+b}{2}}\right|+o(1).$$

(211)

# E Checks of the conjecture for the cumulants near the edge

In the text we have conjectured that the cumulants of order 3 and higher of the number of eigenvalues in an interval of macroscopic size in the bulk are identical for the C$\beta$E and for the G$\beta$E. This conjecture has led to predictions for fermion models. Here we provide a test of this conjecture for $\beta=1,2,4$ by showing that it matches perfectly well with the rigorous results obtained for the G$\beta$E at the edge by Bothner and Buckingham [77]. The methods used in [77] and in [74,75] being completely different, this is a quite non trivial check.

In Ref. [77] Bothner and Buckingham study the eigenvalues $\lambda_i$ of the $N\times N$ random matrices belonging to GUE,GOE and GSE ensembles near the edge, where, for large $N$, they scale as

$$\lambda_i\simeq\sqrt{2N}+\frac{1}{\sqrt{2}N^{1/6}}a_i^\beta,$$

(212)

where the $a_i^\beta$ form the Airy$_\beta$ point process. They study $\mathcal{N}_{[s,+\infty)}$ the number of points $a_i^\beta$ in the interval $[s,+\infty)$. For $v$ real, they prove that, for $\beta=1,2,4$ and in the limit $s\to-\infty$ (i.e., towards the bulk)

$$\log\left\langle e^{-v\mathcal{N}_{[s,+\infty)}}\right\rangle=-\frac{2v}{3\pi}(-s)^{3/2}+\frac{v^2}{2\beta\pi^2}\log\left(8k_\beta(-s)^{3/2}\right)+\chi_\beta(v)+o(1),$$

(213)

where $k_1=k_2=1$ and $k_4=2$ and

$$\chi_\beta(v)=\begin{cases}\frac{1}{2}\mathcal{G}(v)+\frac{1}{2}\log\frac{2}{1+e^v}, & \text{for}\quad\beta=1,\\ \mathcal{G}\left(\frac{v}{2}\right), & \text{for}\quad\beta=2,\\ \frac{1}{2}\mathcal{G}\left(\frac{v}{2}\right)+\log\left(\frac{1}{2}\left(\frac{1+\sqrt{1-e^{-v}}}{1-\sqrt{1-e^{-v}}}\right)^{1/4}+\frac{1}{2}\left(\frac{1-\sqrt{1-e^{-v}}}{1+\sqrt{1-e^{-v}}}\right)^{1/4}\right), & \text{for}\quad\beta=4,\end{cases}$$

(214)

where $\mathcal{G}(v) = \log\left(G\left(1 + \frac{iv}{\pi}\right)G\left(1 - \frac{iv}{\pi}\right)\right)$, where we recall that $G(z)$ is the Barnes G-function [89].

We now compare these results with our predictions. Let us denote $\lambda_i = e^{i\theta_i}$ with $\theta_i \in [0, 2\pi]$ the $N$ eigenvalues for the C$\beta$E. Consider the number $\mathcal{N}_{[0,\theta]}$ of eigenvalues with $\theta_i \in [0, \theta]$. The general conjecture for the FCS [74] is (as also given in the text in (49))

$$\log\left\langle e^{2\pi\sqrt{\frac{\beta}{2}}t(\mathcal{N}_{[0,\theta]} - \langle\mathcal{N}_{[0,\theta]}\rangle)}\right\rangle = 2t^2\log N + t^2\log\left(4\sin^2\frac{\theta}{2}\right) + 2\log|A_\beta(t)|^2, \qquad (215)$$

up to terms that vanish in the large $N$ limit.[13] One has [75]

$$A_\beta(t) = \begin{cases} 2^{-t^2/2}\dfrac{G\left(1 + \frac{it}{\sqrt{2}}\right)G\left(\frac{3}{2} + \frac{it}{\sqrt{2}}\right)}{G(1)G(3/2)} &, \quad \text{for} \quad \beta = 1, \\[2ex] G(1 + it) &, \qquad \text{for} \quad \beta = 2, \\[2ex] \dfrac{G\left(1 + \frac{it}{\sqrt{2}}\right)G\left(\frac{1}{2} + \frac{it}{\sqrt{2}}\right)}{G(1)G(1/2)} &, \qquad \text{for} \quad \beta = 4. \end{cases} \qquad (216)$$

Our conjecture presented in the text implies the following: consider the number $\mathcal{N}_{[\lambda,\lambda']}$ of eigenvalues in the G$\beta$E of size $N \times N$. The FCS generating function $\log\left\langle e^{2\pi\sqrt{\frac{\beta}{2}}t\mathcal{N}_{[\lambda,\lambda']}}\right\rangle$ has the same expression as (215) up to terms of order $O(t)$ and $O(t^2)$ (which correspond to first and second cumulants). This means that all cumulants or order higher than 3 coincide. The same formula holds for the semi-infinite interval, i.e., for $\mathcal{N}_{[\lambda,+\infty)}$, upon dividing the r.h.s of formula (215) by a factor of 2. It is this prediction for the G$\beta$E, valid for $\lambda$ in the bulk, that we can now compare with the result (213), valid for $\lambda$ in the edge region, i.e., as in (212). Indeed the latter result is valid asymptotically for $s \to -\infty$, which corresponds to the limit towards the bulk. We now show that the matching occurs perfectly (without any intermediate regime).

To compare (215) and (213) we note the identification

$$v = 2\pi\sqrt{\frac{\beta}{2}}t, \quad \text{equivalently} \quad t = \sqrt{\frac{2}{\beta}}\frac{v}{2\pi}. \qquad (217)$$

We now discuss the three cases separately.

*Case* $\beta = 2$. In that case $t = \frac{v}{2\pi}$. One checks that $1/2$ times the last term in (215) is equal to $\log|G(1 + it)|^2 = \log\left|G\left(1 + \frac{iv}{2\pi}\right)\right|^2$ which is the last term in (213)-(214). Hence the terms of order $v^3$ and higher exactly coincide in the two formula.

*Case* $\beta = 1$. In that case $t = \frac{v}{\sqrt{2}\pi}$. We need to compare $1/2$ times the last term in (215), which is equal to $\log\left|\dfrac{G\left(1 + \frac{it}{\sqrt{2}}\right)G\left(\frac{3}{2} + \frac{it}{\sqrt{2}}\right)}{G(1)G(3/2)}\right|^2 = \log\left|\dfrac{G\left(1 + \frac{iv}{2\pi}\right)G\left(\frac{3}{2} + \frac{iv}{2\pi}\right)}{G(1)G(3/2)}\right|^2$, with the corresponding term in (213)-(214), which reads $\frac{1}{2}\log\left(\frac{2}{1+e^v}G\left(1 + \frac{iv}{\pi}\right)G\left(1 - \frac{iv}{\pi}\right)\right)$. A priori the identification looks hopeless! However, there exists a remarkable "duplication relation" between Barnes functions, for $v$ real,

$$\left|\dfrac{G\left(1 + \frac{iv}{2\pi}\right)G\left(\frac{3}{2} + \frac{iv}{2\pi}\right)}{G(1)G(3/2)}\right|^4 = \left|G\left(1 + \frac{iv}{\pi}\right)\right|^2\frac{2}{1+e^v}e^{v/2}2^{v^2/\pi^2}, \qquad (218)$$

which we checked explicitly using Mathematica (it is presumably equivalent to the relation (3.5) in [150]). Hence, once again, the terms of order $v^3$ and higher exactly coincide in the

---

[13]Note that Eq. (49) is valid for real $t$ but can be extended for complex $t$ in the neighborhood of $t = 0$ with the replacement $|A_\beta(t)|^2 \to A_\beta(t)A_\beta(-t)$.

two formulae mentioned above.

*Case $\beta = 4$.* In that case $t = \frac{v}{2\sqrt{2}\pi}$. One checks that $1/2$ times the last term in (215) is $\log\left|\frac{G\left(1+\frac{it}{\sqrt{2}}\right)G\left(\frac{1}{2}+\frac{it}{\sqrt{2}}\right)}{G(1)G(1/2)}\right|^2 = \log\left|\frac{G\left(1+\frac{iv}{4\pi}\right)G\left(\frac{1}{2}+\frac{iv}{4\pi}\right)}{G(1)G(1/2)}\right|^2$. This must be compared with the last line (214). This looks even more hopeless than for $\beta = 1$. However, using Mathematica we have discovered the identity valid for real $v$ (where the right hand side appears to be an even function of $v$)

$$\log\frac{\left|\frac{G\left(1+\frac{iv}{4\pi}\right)G\left(\frac{1}{2}+\frac{iv}{4\pi}\right)}{G(1)G(1/2)}\right|^4}{\left|G\left(1+\frac{iv}{2\pi}\right)\right|^2} = 2\log\left(\frac{1}{2}\left(\frac{1+\sqrt{1-e^v}}{1-\sqrt{1-e^v}}\right)^{1/4} + \frac{1}{2}\left(\frac{1-\sqrt{1-e^v}}{1+\sqrt{1-e^v}}\right)^{1/4}\right)$$
$$+ \frac{v}{4} + \frac{v^2}{4\pi^2}\log 2\,, \tag{219}$$

whose derivation we leave as a challenge to the reader.[14] Thus, also for $\beta = 4$, the terms of order $v^3$ and higher exactly coincide in the two formulae mentioned above.

## F  Matching the variance near the edge for the harmonic oscillator

In this Appendix we show that our bulk result for the variance for the harmonic oscillator for general $\beta$, given in (40), matches for $\beta = 1, 2, 4$ the universal edge behavior obtained in [77].

The FCS formula (213) in Appendix E was obtained in [77] for the G$\beta$E. We now translate it in the context of the fermion model in the harmonic potential $V(x) = \frac{1}{2}x^2$. The connection is simply a scale transformation $x_i = \sqrt{\frac{\beta}{2}}\lambda_i$, see Table 1. For general $\beta$ the right edge is thus at position $x^+ = \sqrt{\beta N}$ and the width of the edge region is $w_N = \frac{\sqrt{\beta}}{2}N^{-1/6}$. To obtain the FCS for the number of fermions $\mathcal{N}_{[a,\infty)}$ in the semi-infinite interval $[a, \infty)$, we can simply replace in the formula (213)

$$s \to \hat{a} = \frac{a - x^+}{w_N}\,, \quad \mathcal{N}_{[s,+\infty)} \to \mathcal{N}_{[a,\infty)}\,, \tag{220}$$

We begin with the simplest case, $\beta = 2$. Using the expansion of the Barnes-G function [89]

$$\log G(1+z) = \frac{\log(2\pi) - 1}{2}z - \frac{(1+\gamma_E)}{2}z^2 + \sum_{k=2}^{\infty}(-1)^k\frac{\zeta(k)}{k+1}z^{k+1} \tag{221}$$

we find the leading terms in the expansion of (213) in powers of $v$:

$$\log\left\langle e^{-v\mathcal{N}_{[a,+\infty)}}\right\rangle = -\frac{2v}{3\pi}(-\hat{a})^{3/2} + \frac{\log\left[8(-\hat{a})^{3/2}\right] + 1 + \gamma_E}{2\pi^2}\frac{v^2}{2} + O(v^4)\,. \tag{222}$$

The coefficient of $v^2/2$ in the expansion of (222) corresponds to the second cumulant (the variance) and therefore it gives the asymptotic behavior of the scaling function $\mathcal{V}_2(\hat{a})$ from (58) for $\hat{a} \to -\infty$ as

$$\mathcal{V}_2(\hat{a}) \simeq 2\frac{\frac{3}{2}\log(-\hat{a}) + c_2 + 2\log 2}{2\pi^2}\,, \tag{223}$$

which, together with (58), matches exactly the bulk result (59).

---

[14]As was pointed out to us by Thomas Bothner after submission, the identities (218) and (219) can both be derived from Eq. (3.5) of Ref. [150].

Similarly, for $\beta = 1, 4$ for the harmonic oscillator it was conjectured [43, 92] that there exist universal scaling functions such that in the edge region

$$\mathrm{Var}\mathcal{N}_{[a,\infty)} \simeq \frac{1}{2}\mathcal{V}_\beta\left(\frac{a-x^+}{w_N}\right). \tag{224}$$

Applying the correspondence (220) we obtain from (213)

$$\log\left\langle e^{-\nu\mathcal{N}_{[a,+\infty)}}\right\rangle = \begin{cases} -\frac{2\nu}{3\pi}(-\hat{a})^{3/2} - \frac{\nu}{4} + \frac{\frac{3}{2}\log(-\hat{a})+1+\gamma_E+3\log 2-\frac{\pi^2}{8}}{\pi^2}\frac{\nu^2}{2} + O(\nu^4), & \beta = 1, \\ -\frac{2\nu}{3\pi}(-\hat{a})^{3/2} + \frac{\nu}{8} + \frac{\frac{3}{2}\log(-\hat{a})+1+\gamma_E+4\log 2+\frac{\pi^2}{8}}{4\pi^2}\frac{\nu^2}{2} + O(\nu^4), & \beta = 4. \end{cases} \tag{225}$$

Again, the coefficients of $\nu^2/2$ in these expansions give the asymptotic behaviours of $\mathcal{V}_1(\hat{a})$ and $\mathcal{V}_4(\hat{a})$, which, together with (223), can be summarized as

$$\mathcal{V}_\beta(\hat{a}) \simeq 2\frac{\frac{3}{2}\log(-\hat{a})+c_\beta+2\log 2}{\beta\pi^2}, \qquad -\hat{a} \gg 1, \quad \beta \in \{1, 2, 4\}, \tag{226}$$

which matches the bulk result (40). The leading order (logarithmic) term in (65) was conjectured in [43] for any $\beta$ based on the expected matching with the bulk.

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
