# Peer review of "Full counting statistics for interacting trapped fermions"

_SciPost Physics, doi:SciPost Phys. 11, 110 (2021)_

## Round 1 · Referee Report · Anonymous (Referee 1) · 2021-8-1

Report

The authors consider N spinless fermions with long-range interactions and confined by an external potential in one dimension. They focus on systems which admit a mapping to random matrix ensembles, which they use to study the full counting statistics of the particle number in the ground state.

The manuscript contains many results which, to the best of my knowledge, are new, and could be summarized as follows. First, by following an approach pioneered by Sutherland and Calogero, they identify a set of models for which a mapping to random matrix theory (RMT) exists. By construction, for these models the ground-state wave function can be written down exactly, but they are not necessarily integrable. Next, they present a thorough analysis of the full counting statistics in these models, and study: (i) the mean density; (ii) the variance of the number of fermions in a given interval; (iii) higher cumulants beyond the variance. Among the main results, they present a conjecture for the variance which they prove analytically in some special cases and show that it passes highly non-trivial tests based on previous mathematical results and numerical computations. Furthermore, they substantiate a conjecture for the higher cumulants, stating that they are determined solely from the microscopic scale, thus being independent of the size of the interval and in fact universal. This generalizes a conjecture put forward in a previous work for non-interacting fermions.

Overall, I think the paper is very strong. While some of the material is a natural generalization of earlier work, I believe the paper contains a lot non-trivial new results, and represents a significant step forward in this branch of the literature.

Furthermore, the paper is very well written. I believe the authors make a very good job in the first section, guiding the reader through what was previously known and what are the new contributions of this work. I think this part is very easy to read even by non-experts.

Finally, I believe the research presented is timely, and of potential interest for a broad audience.

For these reasons, I recommend publication of the manuscript essentially as is.

I have only one comment about a minor aspect of the work. In Sec. 1.2, after Eq. (9), the authors compare the results they obtain against LDA, and conclude that the latter is only accurate for noninteracting fermions. However, one could expect this to be true by construction, since for the LDA prediction they consider noninteracting fermions. On the other hand, when the model under consideration happens to be integrable one could perform a slightly more sophisticated version of LDA, by combining it with Bethe Ansatz. In this case, one still assumes the local chemical potential to be given by \mu-V(x), but the density is then determined by the corresponding exact Bethe Ansatz solution for the interacting model. For locally-interacting systems, this is known to work in the bulk, so one could expect that the same is true here. The authors may optionally comment on this.

---

## Round 1 · Referee Report · Anonymous (Referee 2) · 2021-8-25

Strengths

explicit results on FCS for interacting fermions

Weaknesses

some of the models considered are very fine-tuned

Report

The manuscript deals with the study of counting statistics in the
ground state of certain interacting 1D fermion systems.
In particular, they consider model systems with two-body interactions,
where a mapping onto a certain random matrix ensemble exists.
The main example is the Gaussian $\beta$ ensemble, which corresponds
to the Calogero-Sutherland model in the sense that the joint PDF
describing the random matrix eigenvalues is identical to the square
of the fermionic many-body wave function under an appropriate variable
substitution. Similar mappings can be found for the circular,
Wishart-Laguerre and Jacobi $\beta$ ensembles, which describe fermions
on various domains with generalized interaction and potential terms.

The main goal of the paper is to find, using previous RMT results
obtained via the powerful Coulomb gas technique, a formula that relates
the fermion counting statistics to the noninteracting ($\beta=2$) case.
In the Gaussian case the authors find the result (14) for the variance
in a box, which applies to $\beta=1,2,4$. This shows that, after a
proper rescaling of the lengths, the main difference w.r.t. the
free-fermion case is a factor $2/\beta$ that multiplies the leading
order logarithmic term. Furthermore, the authors also propose a more
general formula (18) that is expected to hold for each of the ensembles
and for arbitrary $\beta$. A nontrivial conjecture of the subleading
term is presented in (17) and tested numerically later on in Sec. 2.
Results for higher order cumulants are also presented in (20)-(21).
Finally, some arguments on more general RM ensembles are discussed.

I believe that the manuscript contains novel and interesting results,
extending the studies on FCS towards interacting fermionic models.
Even though some of the models discussed are, due to the mapping to
a concrete RMT ensemble, very fine tuned, these could be important
starting points for further studies. I thus recommend the publication
of the manuscript and have only a few questions.

Requested changes

1. The conjecture on higher order cumulants is not tested numerically. Is it more demanding to check, e.g. requires larger matrix statistics?

2. The statement about the variance at the end of Sec. 6 was not clear to me. Does (18) with an appropriate rescaling and the SAME constant (17) work also for the models in Table 2?

3. It is known that for non-interacting fermions the FCS becomes universal under varying the potential term in appropriate bulk and edge regimes. Could one expect a similar kind of universality when perturbing the interaction term away from the ones that correspond to random matrix ensembles?

  1. Typos
  2. sign convention in (31) and (32) seem to differ
  3. bottom of p. (18): "matches exactly with the formula (58)" seems to be the wrong reference

---

## Round 1 · Referee Report · Anonymous (Referee 3) · 2021-8-26

Strengths

  1. Many new analytical results supported by numerical evidence
  2. Puzzling behaviour of higher order cumulants of FCS is found
  3. Nice introduction to the link between RMT and Calogero-Sutherland model including free fermions
  4. Intuitive but rigorous derivations.

Weaknesses

Not so many, the authors may consider adding a paragraph or two with estimates of various energy scales when discussing universality of higher cumulants.

Report

The manuscript reports new results on Full Counting Statistics of free
fermions and Calogero-Sutherland particles in their ground state. The authors
give results for cumulants of distribution of number of particles in an
interval (finite or infinite) in the thermodynamic limit. Some results are
exact, obtained using mapping of the ground state probability to the joint
probability of eigenvalues of random matrices ensembles. There are also many
conjectured results supported by numerical evidence.

I find the manuscript very well written and useful as a reference for
calculation (at least as a zero order approximation) for strongly interacting
many body systems in one dimension. The authors do great job in explaining the
mapping between random matrices and quantum particles. The previous results
are clearly referenced.

I found the universality of higher cumulants particularly interesting. I think
it deserves more discussion, so the authors may try to add some explanation
with a back of envelope estimates of the energy scales. They could probably
speculate whether such universality will hold for interacting systems other
than Calogero-Sutherland. I also find Eq. (20) a bit puzzling as there is no
dependence on the interval size, unlike in the variance.

I summary I think this is a very good paper with excellent presentation of
many new interesting results and theoretical techniques. While being slightly
on the side of mathematical statistical physics it can benefit
experimentalists in strongly correlated systems.

---

## Round 2 · Author Response

Dear Editor,

please find the revised version of our manuscript titled

"Full counting statistics for interacting trapped fermions"

by N. S. Smith, P. Le Doussal, S. N. Majumdar and G. Schehr,

that we would like to resubmit to SciPost.

Our paper was reviewed by three referees who recommended the publication in SciPost. We have performed the changes that were suggested by the three referees (they are all in blue in this revised version). Below you will find a detailed reply to their comments.

Sincerely yours,

N. S. Smith, P. Le Doussal, S. N. Majumdar and G. Schehr.

Report 1

The authors consider N spinless fermions with long-range interactions and confined by an external potential in one dimension. They focus on systems which admit a mapping to random matrix ensembles, which they use to study the full counting statistics of the particle number in the ground state.

The manuscript contains many results which, to the best of my knowledge, are new, and could be summarized as follows. First, by following an approach pioneered by Sutherland and Calogero, they identify a set of models for which a mapping to random matrix theory (RMT) exists. By construction, for these models the ground-state wave function can be written down exactly, but they are not necessarily integrable. Next, they present a thorough analysis of the full counting statistics in these models, and study: (i) the mean density; (ii) the variance of the number of fermions in a given interval; (iii) higher cumulants beyond the variance. Among the main results, they present a conjecture for the variance which they prove analytically in some special cases and show that it passes highly non-trivial tests based on previous mathematical results and numerical computations. Furthermore, they substantiate a conjecture for the higher cumulants, stating that they are determined solely from the microscopic scale, thus being independent of the size of the interval and in fact universal. This generalizes a conjecture put forward in a previous work for non-interacting fermions.

Overall, I think the paper is very strong. While some of the material is a natural generalization of earlier work, I believe the paper contains a lot non-trivial new results, and represents a significant step forward in this branch of the literature.

Furthermore, the paper is very well written. I believe the authors make a very good job in the first section, guiding the reader through what was previously known and what are the new contributions of this work. I think this part is very easy to read even by non-experts.

Finally, I believe the research presented is timely, and of potential interest for a broad audience.

For these reasons, I recommend publication of the manuscript essentially as is.

I have only one comment about a minor aspect of the work. In Sec. 1.2, after Eq. (9), the authors compare the results they obtain against LDA, and conclude that the latter is only accurate for noninteracting fermions. However, one could expect this to be true by construction, since for the LDA prediction they consider noninteracting fermions. On the other hand, when the model under consideration happens to be integrable one could perform a slightly more sophisticated version of LDA, by combining it with Bethe Ansatz. In this case, one still assumes the local chemical potential to be given by \mu-V(x), but the density is then determined by the corresponding exact Bethe Ansatz solution for the interacting model. For locally-interacting systems, this is known to work in the bulk, so one could expect that the same is true here. The authors may optionally comment on this.

----------> Reply to referee 1: We thank the referee for their positive opinion on our work, and for the clever remark. The calculation described by the referee would of course be interesting to perform, but we believe that it is beyond the scope of this paper. We therefore added a couple of sentences to the end of the paragraph that follows Eq. (9), mentioning the possibility of such a calculation and referring the reader to Ref. [108] where a similar one was performed.

Report 2

Strengths

explicit results on FCS for interacting fermions

Weaknesses

some of the models considered are very fine-tuned

Report

The manuscript deals with the study of counting statistics in the ground state of certain interacting 1D fermion systems. In particular, they consider model systems with two-body interactions, where a mapping onto a certain random matrix ensemble exists. The main example is the Gaussian β ensemble, which corresponds to the Calogero-Sutherland model in the sense that the joint PDF describing the random matrix eigenvalues is identical to the square of the fermionic many-body wave function under an appropriate variable substitution. Similar mappings can be found for the circular, Wishart-Laguerre and Jacobi β ensembles, which describe fermions on various domains with generalized interaction and potential terms.

The main goal of the paper is to find, using previous RMT results obtained via the powerful Coulomb gas technique, a formula that relates the fermion counting statistics to the noninteracting (β=2) case. In the Gaussian case the authors find the result (14) for the variance in a box, which applies to β=1,2,4. This shows that, after a proper rescaling of the lengths, the main difference w.r.t. the free-fermion case is a factor 2/β that multiplies the leading order logarithmic term. Furthermore, the authors also propose a more general formula (18) that is expected to hold for each of the ensembles and for arbitrary β. A nontrivial conjecture of the subleading term is presented in (17) and tested numerically later on in Sec. 2. Results for higher order cumulants are also presented in (20)-(21). Finally, some arguments on more general RM ensembles are discussed.

I believe that the manuscript contains novel and interesting results, extending the studies on FCS towards interacting fermionic models. Even though some of the models discussed are, due to the mapping to a concrete RMT ensemble, very fine tuned, these could be important starting points for further studies. I thus recommend the publication of the manuscript and have only a few questions.

Requested changes

  1. The conjecture on higher order cumulants is not tested numerically. Is it more demanding to check, e.g. requires larger matrix statistics?

    -------> Indeed this is the case. We added a comment on this issue at the end of section 3.

  2. The statement about the variance at the end of Sec. 6 was not clear to me. Does (18) with an appropriate rescaling and the SAME constant (17) work also for the models in Table 2?

    -------> Yes, it is the same constants given in (17). We clarified our statement at the end of section 6.

  3. It is known that for non-interacting fermions the FCS becomes universal under varying the potential term in appropriate bulk and edge regimes. Could one expect a similar kind of universality when perturbing the interaction term away from the ones that correspond to random matrix ensembles?

    -------> This is indeed an interesting question that we leave for future work. We discuss this question in the 4th paragraph of the discussion section (we extended this part of the discussion a little following the referee's comment).

  4. Typos

    • sign convention in (31) and (32) seem to differ

      --------> We thank the referee for this important correction. We have corrected the sign in Eq. (32).

    • bottom of p. (18): "matches exactly with the formula (58)" seems to be the wrong reference

      --------> This reference is in fact correct, the matching is with the asymptotic behavior of {\cal V}_2 which is given in Eq. (222) in Appendix F.

Report 3

Strengths

  1. Many new analytical results supported by numerical evidence

  2. Puzzling behaviour of higher order cumulants of FCS is found

  3. Nice introduction to the link between RMT and Calogero-Sutherland model including free fermions

  4. Intuitive but rigorous derivations.

Weaknesses

Not so many, the authors may consider adding a paragraph or two with estimates of various energy scales when discussing universality of higher cumulants.

Report

The manuscript reports new results on Full Counting Statistics of free fermions and Calogero-Sutherland particles in their ground state. The authors give results for cumulants of distribution of number of particles in an interval (finite or infinite) in the thermodynamic limit. Some results are exact, obtained using mapping of the ground state probability to the joint probability of eigenvalues of random matrices ensembles. There are also many conjectured results supported by numerical evidence.

I find the manuscript very well written and useful as a reference for calculation (at least as a zero order approximation) for strongly interacting many body systems in one dimension. The authors do great job in explaining the mapping between random matrices and quantum particles. The previous results are clearly referenced.

I found the universality of higher cumulants particularly interesting. I think it deserves more discussion, so the authors may try to add some explanation with a back of envelope estimates of the energy scales. They could probably speculate whether such universality will hold for interacting systems other than Calogero-Sutherland. I also find Eq. (20) a bit puzzling as there is no dependence on the interval size, unlike in the variance.

I summary I think this is a very good paper with excellent presentation of many new interesting results and theoretical techniques. While being slightly on the side of mathematical statistical physics it can benefit experimentalists in strongly correlated systems.

----------> Reply to referee 3: We thank the referee for their positive opinion on our work. Regarding the universality of higher cumulants (which we too find quite interesting), we gave it much attention in the abstract, introduction and discussion. Nevertheless, this universality was observed in some particular cases already in [49,72], including the different behavior of the variance. Therefore, we did not discuss in great detail some of the aspects of this universality, including in particular the difference between the behavior of the variance and the higher cumulants. The question that the referee raises, regarding other types of interactions, is indeed an interesting question that we leave for future work. We discuss this question in the 4th paragraph of the discussion section (we extended this part of the discussion a little following the referee's comment).

---

## Round 2 · List of Changes

Warnings issued while processing user-supplied markup:

  • Inconsistency: Markdown and reStructuredText syntaxes are mixed. Markdown will be used.
    Add "#coerce:reST" or "#coerce:plain" as the first line of your text to force reStructuredText or no markup.
    You may also contact the helpdesk if the formatting is incorrect and you are unable to edit your text.

Report 1

The authors consider N spinless fermions with long-range interactions and confined by an external potential in one dimension. They focus on systems which admit a mapping to random matrix ensembles, which they use to study the full counting statistics of the particle number in the ground state.

The manuscript contains many results which, to the best of my knowledge, are new, and could be summarized as follows. First, by following an approach pioneered by Sutherland and Calogero, they identify a set of models for which a mapping to random matrix theory (RMT) exists. By construction, for these models the ground-state wave function can be written down exactly, but they are not necessarily integrable. Next, they present a thorough analysis of the full counting statistics in these models, and study: (i) the mean density; (ii) the variance of the number of fermions in a given interval; (iii) higher cumulants beyond the variance. Among the main results, they present a conjecture for the variance which they prove analytically in some special cases and show that it passes highly non-trivial tests based on previous mathematical results and numerical computations. Furthermore, they substantiate a conjecture for the higher cumulants, stating that they are determined solely from the microscopic scale, thus being independent of the size of the interval and in fact universal. This generalizes a conjecture put forward in a previous work for non-interacting fermions.

Overall, I think the paper is very strong. While some of the material is a natural generalization of earlier work, I believe the paper contains a lot non-trivial new results, and represents a significant step forward in this branch of the literature.

Furthermore, the paper is very well written. I believe the authors make a very good job in the first section, guiding the reader through what was previously known and what are the new contributions of this work. I think this part is very easy to read even by non-experts.

Finally, I believe the research presented is timely, and of potential interest for a broad audience.

For these reasons, I recommend publication of the manuscript essentially as is.

I have only one comment about a minor aspect of the work. In Sec. 1.2, after Eq. (9), the authors compare the results they obtain against LDA, and conclude that the latter is only accurate for noninteracting fermions. However, one could expect this to be true by construction, since for the LDA prediction they consider noninteracting fermions. On the other hand, when the model under consideration happens to be integrable one could perform a slightly more sophisticated version of LDA, by combining it with Bethe Ansatz. In this case, one still assumes the local chemical potential to be given by \mu-V(x), but the density is then determined by the corresponding exact Bethe Ansatz solution for the interacting model. For locally-interacting systems, this is known to work in the bulk, so one could expect that the same is true here. The authors may optionally comment on this.

----------> Reply to referee 1: We thank the referee for their positive opinion on our work, and for the clever remark. The calculation described by the referee would of course be interesting to perform, but we believe that it is beyond the scope of this paper. We therefore added a couple of sentences to the end of the paragraph that follows Eq. (9), mentioning the possibility of such a calculation and referring the reader to Ref. [108] where a similar one was performed.

Report 2

Strengths explicit results on FCS for interacting fermions

Weaknesses some of the models considered are very fine-tuned

Report The manuscript deals with the study of counting statistics in the ground state of certain interacting 1D fermion systems. In particular, they consider model systems with two-body interactions, where a mapping onto a certain random matrix ensemble exists. The main example is the Gaussian β ensemble, which corresponds to the Calogero-Sutherland model in the sense that the joint PDF describing the random matrix eigenvalues is identical to the square of the fermionic many-body wave function under an appropriate variable substitution. Similar mappings can be found for the circular, Wishart-Laguerre and Jacobi β ensembles, which describe fermions on various domains with generalized interaction and potential terms.

The main goal of the paper is to find, using previous RMT results obtained via the powerful Coulomb gas technique, a formula that relates the fermion counting statistics to the noninteracting (β=2) case. In the Gaussian case the authors find the result (14) for the variance in a box, which applies to β=1,2,4. This shows that, after a proper rescaling of the lengths, the main difference w.r.t. the free-fermion case is a factor 2/β that multiplies the leading order logarithmic term. Furthermore, the authors also propose a more general formula (18) that is expected to hold for each of the ensembles and for arbitrary β. A nontrivial conjecture of the subleading term is presented in (17) and tested numerically later on in Sec. 2. Results for higher order cumulants are also presented in (20)-(21). Finally, some arguments on more general RM ensembles are discussed.

I believe that the manuscript contains novel and interesting results, extending the studies on FCS towards interacting fermionic models. Even though some of the models discussed are, due to the mapping to a concrete RMT ensemble, very fine tuned, these could be important starting points for further studies. I thus recommend the publication of the manuscript and have only a few questions.

Requested changes 1. The conjecture on higher order cumulants is not tested numerically. Is it more demanding to check, e.g. requires larger matrix statistics?

-------> Indeed this is the case. We added a comment on this issue at the end of section 3.

  1. The statement about the variance at the end of Sec. 6 was not clear to me. Does (18) with an appropriate rescaling and the SAME constant (17) work also for the models in Table 2?

-------> Yes, it is the same constants given in (17). We clarified our statement at the end of section 6.

  1. It is known that for non-interacting fermions the FCS becomes universal under varying the potential term in appropriate bulk and edge regimes. Could one expect a similar kind of universality when perturbing the interaction term away from the ones that correspond to random matrix ensembles?

-------> This is indeed an interesting question that we leave for future work. We discuss this question in the 4th paragraph of the discussion section (we extended this part of the discussion a little following the referee's comment).

  1. Typos
  2. sign convention in (31) and (32) seem to differ --------> We thank the referee for this important correction. We have corrected the sign in Eq. (32).
  3. bottom of p. (18): "matches exactly with the formula (58)" seems to be the wrong reference --------> This reference is in fact correct, the matching is with the asymptotic behavior of {\cal V}_2 which is given in Eq. (222) in Appendix F.

Report 3

Strengths 1. Many new analytical results supported by numerical evidence 2. Puzzling behaviour of higher order cumulants of FCS is found 3. Nice introduction to the link between RMT and Calogero-Sutherland model including free fermions 4. Intuitive but rigorous derivations.

Weaknesses Not so many, the authors may consider adding a paragraph or two with estimates of various energy scales when discussing universality of higher cumulants.

Report The manuscript reports new results on Full Counting Statistics of free fermions and Calogero-Sutherland particles in their ground state. The authors give results for cumulants of distribution of number of particles in an interval (finite or infinite) in the thermodynamic limit. Some results are exact, obtained using mapping of the ground state probability to the joint probability of eigenvalues of random matrices ensembles. There are also many conjectured results supported by numerical evidence.

I find the manuscript very well written and useful as a reference for calculation (at least as a zero order approximation) for strongly interacting many body systems in one dimension. The authors do great job in explaining the mapping between random matrices and quantum particles. The previous results are clearly referenced.

I found the universality of higher cumulants particularly interesting. I think it deserves more discussion, so the authors may try to add some explanation with a back of envelope estimates of the energy scales. They could probably speculate whether such universality will hold for interacting systems other than Calogero-Sutherland. I also find Eq. (20) a bit puzzling as there is no dependence on the interval size, unlike in the variance.

I summary I think this is a very good paper with excellent presentation of many new interesting results and theoretical techniques. While being slightly on the side of mathematical statistical physics it can benefit experimentalists in strongly correlated systems.

----------> Reply to referee 3: We thank the referee for their positive opinion on our work. Regarding the universality of higher cumulants (which we too find quite interesting), we gave it much attention in the abstract, introduction and discussion. Nevertheless, this universality was observed in some particular cases already in [49,72], including the different behavior of the variance. Therefore, we did not discuss in great detail some of the aspects of this universality, including in particular the difference between the behavior of the variance and the higher cumulants. The question that the referee raises, regarding other types of interactions, is indeed an interesting question that we leave for future work. We discuss this question in the 4th paragraph of the discussion section (we extended this part of the discussion a little following the referee's comment).

---

## Editorial Decision

published